# Nonlinear transient amplification in recurrent neural networks with short-term plasticity

Yue Kris Wu[1,2,3,4], Friedemann Zenke[1,2]*

[1]Friedrich Miescher Institute for Biomedical Research, Basel, Switzerland; [2]Faculty of Natural Sciences, University of Basel, Basel, Switzerland; [3]Max Planck Institute for Brain Research, Frankfurt, Germany; [4]School of Life Sciences, Technical University of Munich, Freising, Germany

**Abstract** To rapidly process information, neural circuits have to amplify specific activity patterns transiently. How the brain performs this nonlinear operation remains elusive. Hebbian assemblies are one possibility whereby strong recurrent excitatory connections boost neuronal activity. However, such Hebbian amplification is often associated with dynamical slowing of network dynamics, non-transient attractor states, and pathological run-away activity. Feedback inhibition can alleviate these effects but typically linearizes responses and reduces amplification gain. Here, we study nonlinear transient amplification (NTA), a plausible alternative mechanism that reconciles strong recurrent excitation with rapid amplification while avoiding the above issues. NTA has two distinct temporal phases. Initially, positive feedback excitation selectively amplifies inputs that exceed a critical threshold. Subsequently, short-term plasticity quenches the run-away dynamics into an inhibition-stabilized network state. By characterizing NTA in supralinear network models, we establish that the resulting onset transients are stimulus selective and well-suited for speedy information processing. Further, we find that excitatory-inhibitory co-tuning widens the parameter regime in which NTA is possible in the absence of persistent activity. In summary, NTA provides a parsimonious explanation for how excitatory-inhibitory co-tuning and short-term plasticity collaborate in recurrent networks to achieve transient amplification.

*For correspondence:
friedemann.zenke@fmi.ch

**Competing interest:** The authors declare that no competing interests exist.

## Editor's evaluation

Many brain circuits, particularly those found in mammalian sensory cortices, need to respond rapidly to stimuli while at the same time avoiding pathological, runaway excitation. Over several years, many theoretical studies have attempted to explain how cortical circuits achieve these goals through interactions between inhibitory and excitatory cells. This study adds to this literature by showing how synaptic short-term depression can stabilise strong positive feedback in a circuit under a variety of plausible scenarios, allowing strong, rapid and stimulus-specific responses.

## Introduction

Perception in the brain is reliable and strikingly fast. Recognizing a familiar face or locating an animal in a picture only takes a split second (*Thorpe et al., 1996*). This pace of processing is truly remarkable since it involves several recurrently connected brain areas each of which has to selectively amplify or suppress specific signals before propagating them further. This processing is mediated through circuits with several intriguing properties. First, excitatory-inhibitory (EI) currents into individual neurons are commonly correlated in time and co-tuned in stimulus space (*Wehr and Zador, 2003*; *Froemke et al.,*

*2007*; *Okun and Lampl, 2008*; *Hennequin et al., 2017*; *Rupprecht and Friedrich, 2018*; *Znamenskiy et al., 2018*). Second, neural responses to stimulation are shaped through diverse forms of short-term plasticity (STP) (*Tsodyks and Markram, 1997*; *Markram et al., 1998*; *Zucker and Regehr, 2002*; *Pala and Petersen, 2015*). Finally, mounting evidence suggests that amplification rests on neuronal ensembles with strong recurrent excitation (*Marshel et al., 2019*; *Peron et al., 2020*), whereby excitatory neurons with similar tuning preferentially form reciprocal connections (*Ko et al., 2011*; *Cossell et al., 2015*). Such predominantly symmetric connectivity between excitatory cells is consistent with the notion of Hebbian cell assemblies (*Hebb, 1949*), which are considered an essential component of neural circuits and the putative basis of associative memory (*Harris, 2005*; *Josselyn and Tonegawa, 2020*). Computationally, Hebbian cell assemblies can amplify specific activity patterns through positive feedback, also referred to as Hebbian amplification. Based on these principles, several studies have shown that Hebbian amplification can drive persistent activity that outlasts a preceding stimulus (*Hopfield, 1982*; *Amit and Brunel, 1997*; *Yakovlev et al., 1998*; *Wong and Wang, 2006*; *Zenke et al., 2015*; *Gillary et al., 2017*), comparable to selective delay activity observed in the prefrontal cortex when animals are engaged in working memory tasks (*Funahashi et al., 1989*; *Romo et al., 1999*).

However, in most brain areas, evoked responses are transient and sensory neurons typically exhibit pronounced stimulus *onset* responses, after which the circuit dynamics settle into a low-activity steady-state even when the stimulus is still present (*DeWeese et al., 2003*; *Mazor and Laurent, 2005*; *Bolding and Franks, 2018*). Preventing run-away excitation and multi-stable attractor dynamics in recurrent networks requires powerful and often finely tuned feedback inhibition resulting in EI balance (*Amit and Brunel, 1997*; *Compte et al., 2000*; *Litwin-Kumar and Doiron, 2012*; *Ponce-Alvarez et al., 2013*; *Mazzucato et al., 2019*), However, strong feedback inhibition tends to linearize steady-state activity (*van Vreeswijk and Sompolinsky, 1996*; *Baker et al., 2020*). *Murphy and Miller, 2009* proposed *balanced amplification* which reconciles transient amplification with strong recurrent excitation by tightly balancing recurrent excitation with strong feedback inhibition (*Goldman, 2009*; *Hennequin et al., 2012*; *Hennequin et al., 2014*; *Bondanelli and Ostojic, 2020*; *Gillett et al., 2020*). Importantly, balanced amplification was formulated for linear network models of excitatory and inhibitory neurons. Due to linearity, it intrinsically lacks the ability to nonlinearly amplify stimuli which limits its capabilities for pattern completion and pattern separation. Further, how balanced amplification relates to nonlinear neuronal activation functions and nonlinear synaptic transmission as, for instance, mediated by STP (*Tsodyks and Markram, 1997*; *Markram et al., 1998*; *Zucker and Regehr, 2002*; *Pala and Petersen, 2015*), remains elusive. This begs the question of whether there are alternative nonlinear amplification mechanisms and how they relate to existing theories of recurrent neural network processing.

Here, we address this question by studying an alternative mechanism for the emergence of transient dynamics that relies on recurrent excitation, supralinear neuronal activation functions, and STP. Specifically, we build on the notion of ensemble synchronization in recurrent networks with STP (*Loebel and Tsodyks, 2002*; *Loebel et al., 2007*) and study this phenomenon in analytically tractable network models with rectified quadratic activation functions (*Ahmadian et al., 2013*; *Rubin et al., 2015*; *Hennequin et al., 2018*; *Kraynyukova and Tchumatchenko, 2018*) and STP. We first characterize the conditions under which individual neuronal ensembles with supralinear activation functions and recurrent excitatory connectivity succumb to explosive run-away activity in response to external stimulation. We then show how STP effectively mitigates this instability by re-stabilizing ensemble dynamics in an inhibition-stabilized network (ISN) state, but only after generating a pronounced stimulus-triggered onset transient. We call this mechanism NTA and show that it yields selective onset responses that carry more relevant stimulus information than the subsequent steady-state. Finally, we characterize the functional benefits of inhibitory co-tuning, a feature that is widely observed in the brain (*Wehr and Zador, 2003*; *Froemke et al., 2007*; *Okun and Lampl, 2008*; *Rupprecht and Friedrich, 2018*) and readily emerges in computational models endowed with activity-dependent plasticity of inhibitory synapses (*Vogels et al., 2011*). We find that co-tuning prevents persistent attractor states but does not preclude NTA from occurring. Importantly, NTA purports that, following transient amplification, neuronal ensembles settle into a stable ISN state, consistent with recent studies suggesting that inhibition stabilization is a ubiquitous feature of cortical networks (*Sanzeni et al., 2020*). In summary, our work indicates that NTA is ideally suited to amplify stimuli rapidly through the interaction of strong recurrent excitation with STP.

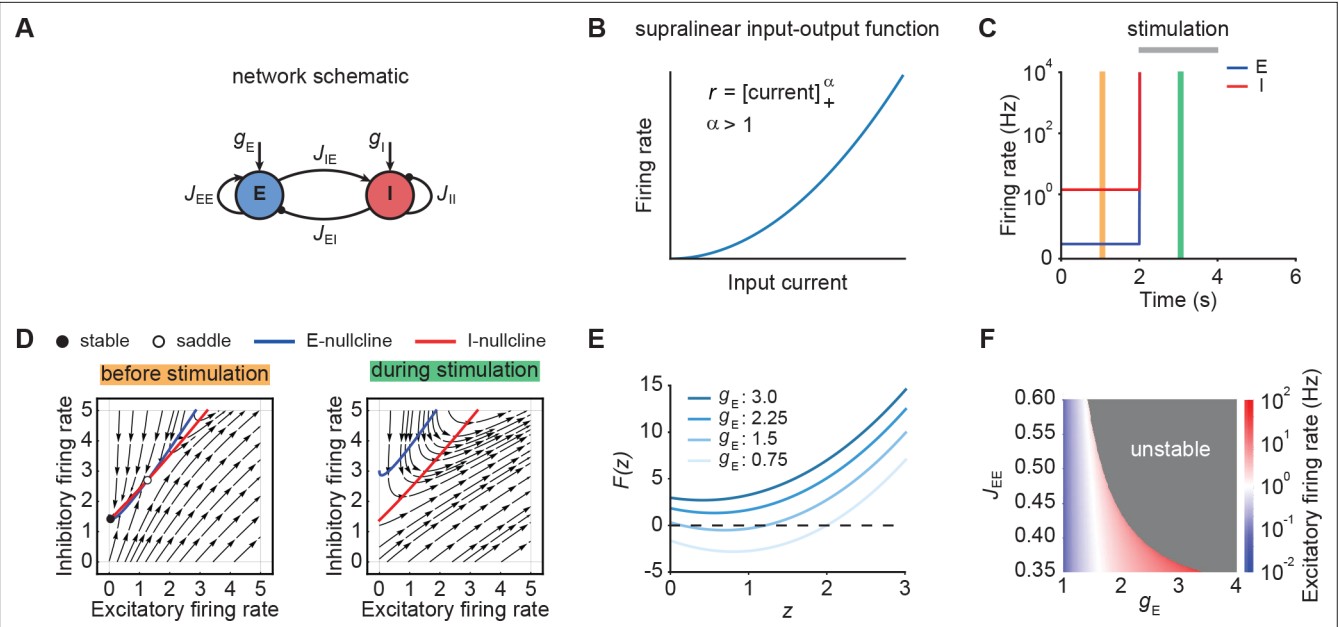

**Figure 1.** Neuronal ensembles nonlinearly amplify inputs above a critical threshold. (**A**) Schematic of the recurrent ensemble model consisting of an excitatory (blue) and an inhibitory population (red). (**B**) Supralinear input-output function given by a rectified power law with exponent $\alpha = 2$. (**C**) Firing rates of the excitatory (blue) and inhibitory population (red) in response to external stimulation during the interval from 2 to 4 s (gray bar). The stimulation was implemented by temporarily increasing the input $g_E$. (**D**) Phase portrait of the system before stimulation (left; C orange) and during stimulation (right; C green). (**E**) Characteristic function $F(z)$ for varying input strength $g_E$. Note that the function loses its zero crossings, which correspond to fixed points of the system for increasing external input. (**F**) Heat map showing the evoked firing rate of the excitatory population for different parameter combinations $J_{EE}$ and $g_E$. The gray region corresponds to the parameter regime with unstable dynamics.

The online version of this article includes the following figure supplement(s) for figure 1:

**Figure supplement 1.** Unstable ensemble dynamics can be triggered by additional stimulation in supralinear networks with negative determinant even in the presence of substantial feedforward inhibition.

## Results

To understand the emergence of transient responses in recurrent neural networks, we studied rate-based population models with a supralinear, power law input-output function (**Figure 1A and B**; **Ahmadian et al., 2013**; **Hennequin et al., 2018**), which captures essential aspects of neuronal activation (**Priebe et al., 2004**), while also being analytically tractable. We first considered an isolated neuronal ensemble consisting of one excitatory (E) and one inhibitory (I) population (**Figure 1A**).

The dynamics of this network are given by

$$\tau_E \frac{dr_E}{dt} = -r_E + \left[ J_{EE} r_E - J_{EI} r_I + g_E \right]_+^{\alpha_E} \quad , \tag{1}$$

$$\tau_I \frac{dr_I}{dt} = -r_I + \left[ J_{IE} r_E - J_{II} r_I + g_I \right]_+^{\alpha_I} \quad , \tag{2}$$

where $r_E$ and $r_I$ are the firing rates of the excitatory and inhibitory population, $\tau_E$ and $\tau_I$ represent the corresponding time constants, $J_{XY}$ denotes the synaptic strength from the population $Y$ to the population $X$, where $X, Y \in \{E, I\}$, $g_E$ and $g_I$ are the external inputs to the respective populations. Finally, $\alpha_E$ and $\alpha_I$, the exponents of the respective input-output functions, are fixed at two unless mentioned otherwise. For ease of notation, we further define the weight matrix $\mathbf{J}$ of the compound system as follows:

$$\mathbf{J} = \begin{bmatrix} J_{EE} & -J_{EI} \\ J_{IE} & -J_{II} \end{bmatrix} \quad . \tag{3}$$

We were specifically interested in networks with strong recurrent excitation that can generate positive feedback dynamics in response to external inputs $g_E$. Therefore, we studied networks with

$$\det(\mathbf{J}) = -J_{EE}J_{II} + J_{IE}J_{EI} < 0 \quad . \tag{4}$$

In contrast, networks in which recurrent excitation is met by strong feedback inhibition such that $\det(\mathbf{J}) > 0$ are unable to generate positive feedback dynamics provided that inhibition is fast enough (*Ahmadian et al., 2013*). Importantly, we assumed that most inhibition originates from recurrent connections (*Franks et al., 2011*; *Large et al., 2016*) and, hence, we kept the input to the inhibitory population $g_I$ fixed unless mentioned otherwise.

## Nonlinear amplification of inputs above a critical threshold

We initialized the network in a stable low-activity state in the absence of external stimulation, consistent with spontaneous activity in cortical networks (*Figure 1C*). However, an input $g_E$ of sufficient strength, destabilized the network (*Figure 1C*). Importantly, this behavior is distinct from linear network models in which the network stability is independent of inputs (Materials and methods). The transition from stable to unstable dynamics can be understood by examining the phase portrait of the system (*Figure 1D*). Before stimulation, the system has a stable and an unstable fixed point (*Figure 1D*, left). However, both fixed points disappear for an input $g_E$ above a critical stimulus strength (*Figure 1D*, right).

To further understand the system's bifurcation structure, we consider the characteristic function

$$F(z) = J_{EE}\left[z\right]_+^{\alpha_E} - J_{EI}\left[\det(\mathbf{J}) \cdot J_{EI}^{-1}\left[z\right]_+^{\alpha_E} + J_{EI}^{-1}J_{II}z - J_{EI}^{-1}J_{II}g_E + g_I\right]_+^{\alpha_I} - z + g_E \quad , \tag{5}$$

where $z$ denotes the total current into the excitatory population and $\det(\mathbf{J})$ represents the determinant of the weight matrix (*Kraynyukova and Tchumatchenko, 2018*; Materials and methods). The characteristic function reduces the original two-dimensional system to one dimension, whereby the zero crossings of the characteristic function correspond to the fixed points of the original system (*Eq. (1)-(2)*). We use this correspondence to visualize how the fixed points of the system change with the input $g_E$. Increasing $g_E$ shifts $F(z)$ upwards, which eventually leads to all zero crossings disappearing and the ensuing unstable dynamics (*Figure 1E*; Materials and methods). Importantly, for any weight matrix $\mathbf{J}$ with negative determinant, there exists a critical input $g_E$ at which all fixed points disappear (Materials and methods). While for weak recurrent E-to-E connection strength $J_{EE}$, the transition from stable dynamics to unstable is gradual, in that it happens at higher firing rates (*Figure 1F*), it becomes more abrupt for stronger $J_{EE}$. Thus, our analysis demonstrates that individual neuronal ensembles with negative determinant $\det(\mathbf{J})$ nonlinearly amplify inputs above a critical threshold by switching from initially stable to unstable dynamics.

## Short-term plasticity, but not spike-frequency adaptation, can re-stabilize ensemble dynamics

Since unstable dynamics are not observed in neurobiology, we wondered whether neuronal spike frequency adaptation (SFA) or STP could re-stabilize the ensemble dynamics while keeping the nonlinear amplification character of the system. Specifically, we considered SFA of excitatory neurons, E-to-E short-term depression (STD), and E-to-I short-term facilitation (STF). We focused on these particular mechanisms because they are ubiquitously observed in the brain. Most pyramidal cells exhibit SFA (*Barkai and Hasselmo, 1994*) and most synapses show some form of STP (*Markram et al., 1998*; *Zucker and Regehr, 2002*; *Pala and Petersen, 2015*). Moreover, the time scales of these mechanisms are well-matched to typical timescales of perception, ranging from milliseconds to seconds (*Tsodyks and Markram, 1997*; *Fairhall et al., 2001*; *Pozzorini et al., 2013*).

When we simulated our model with SFA (*Eqs. (21)–(23)*), we observed different network behaviors depending on the adaptation strength. When adaptation strength was weak, SFA was unable to stabilize run-away excitation (*Figure 2A*; Materials and methods). Increasing the adaptation strength eventually prevented run-away excitation, but to give way to oscillatory ensemble activity (*Figure 2—figure supplement 1*). Finally, we confirmed analytically that SFA cannot stabilize excitatory run-away dynamics at a stable fixed point (Materials and methods). In particular, while the input is present, strong SFA creates a stable limit cycle with associated oscillatory ensemble activity (*Figure 2—figure*

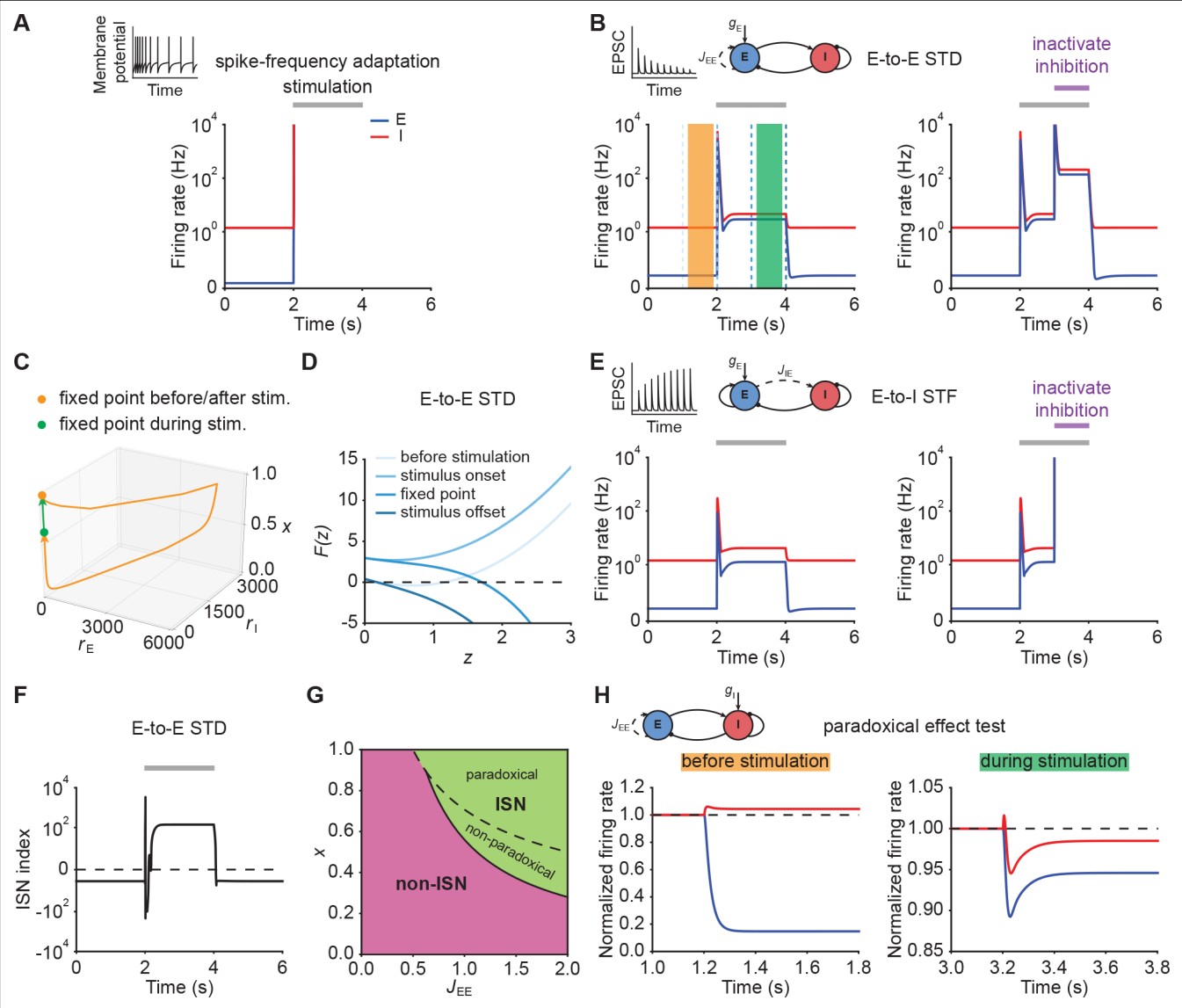

**Figure 2.** Short-term plasticity, but not spike-frequency adaptation, re-stabilizes ensemble dynamics. (**A**) Firing rates of the excitatory (blue) and inhibitory population (red) in the presence of spike-frequency adaptation (SFA). During stimulation (gray bar) additional input is injected into the excitatory population. The inset shows a cartoon of how SFA affects spiking neuronal dynamics in response to a step current input. (**B**) Left: Same as (**A**) but in the presence of E-to-E short-term depression (STD). Right: Same as left but inactivating inhibition in the period marked in purple. (**C**) 3D plot of the excitatory activity $r_E$, inhibitory activity $r_I$, and the STD variable $x$ of the network in B left. The orange and green points mark the fixed points before/after and during stimulation. (**D**) Characteristic function $F(z)$ in networks with E-to-E STD. Different brightness levels correspond to different time points in B left. (**E**) Same as (**B**) but in the presence of E-to-I short-term facilitation (STF). (**F**) Inhibition-stabilized network (ISN) index, which corresponds to the largest real part of the eigenvalues of the Jacobian matrix of the E-E subnetwork with STD, as a function of time for the network with E-to-E STD in B left. For values above zero (dashed line), the ensemble is an ISN. (**G**) Analytical solution of non-ISN (magenta), ISN (green), paradoxical, and non-paradoxical regions for different parameter combinations $J_{EE}$ and the STD variable $x$. The solid line separates the non-ISN and ISN regions, whereas the dashed line separates the non-paradoxical and paradoxical regions. (**H**) The normalized firing rates of the excitatory (blue) and inhibitory population (red) when injecting additional excitatory current into the inhibitory population before stimulation (left; orange bar in B), and during stimulation (right; green bar in B). Initially, the ensemble is in the non-ISN regime and injecting excitatory current into the inhibitory population increases its firing rate. During stimulation, however, the ensemble is an ISN. In this case, excitatory current injection into the inhibitory population results in a reduction of its firing rate, also known as the *paradoxical effect*.

The online version of this article includes the following figure supplement(s) for figure 2:

**Figure supplement 1.** Ensemble dynamics in supralinear networks with strong SFA.

**Figure supplement 2.** Dependence of peak amplitude and fixed point activity on input $g_E$ and E-to-E connection strength $J_{EE}$.

*Figure 2 continued on next page*

*Figure 2 continued*

**Figure supplement 3.** Comparisons of amplification ability between NTA and linear networks, and between NTA and SSNs.

**Figure supplement 4.** Dependence of peak amplitude and fixed point activity on STP parameters.

**Figure supplement 5.** Networks initially in the ISN regime can exhibit strong NTA.

**Figure supplement 6.** ISN index and paradoxical effect test for networks with E-to-I STF.

**Figure supplement 7.** Inhibition stabilization does not imply paradoxical response in networks with E-to-E STD.

**Figure supplement 8.** Transition from non-ISN to ISN indicating by frozen inhibition test.

**Figure supplement 9.** Similar qualitative behavior in rate-based models with maximal firing rate capped at 300 Hz.

**Figure supplement 10.** Similar qualitative behavior in spiking neural networks.

**Figure supplement 11.** Unstable dynamics can emerge in supralinear networks with positive determinant and slow inhibition.

**Figure supplement 12.** Networks with substantial feedforward inhibition can exhibit strong NTA.

*supplement 1*; Materials and methods), which was also shown in previous modeling studies (*van Vreeswijk and Hansel, 2001*), but is not typically observed in sensory systems (*DeWeese et al., 2003*; *Rupprecht and Friedrich, 2018*).

Next, we considered STP, which is capable of saturating the effective neuronal input-output function (*Mongillo et al., 2012*; *Zenke et al., 2015*; *Eqs. (37)–(39)*, *Eqs. (41)–(43)*). We first analyzed the stimulus-evoked network dynamics when we added STD to the recurrent E-to-E connections. Strong depression of synaptic efficacy resulted in a brief onset transient after which the ensemble dynamics quickly settled into a stimulus-evoked steady-state with slightly higher activity than the baseline (*Figure 2B*, left). After stimulus removal, the ensemble activity returned back to its baseline level (*Figure 2B*, left; *Figure 2C*). Notably, the ensemble dynamics settled at a stable steady state with a much higher firing rate, when inhibition was inactivated during stimulus presentation (*Figure 2B*, right). This shows that STP is capable of creating a stable high-activity fixed point, which is fundamentally different from the SFA dynamics discussed above. This difference in ensemble dynamics can be readily understood by analyzing the stability of the three-dimensional dynamical system (Materials and methods). We can gain a more intuitive understanding by considering self-consistent solutions of the characteristic function $F(z)$. Initially, the ensemble is at the stable low activity fixed point. But the stimulus causes this fixed point to disappear, thus giving way to positive feedback which creates the leading edge of the onset transient (*Figure 2B*). However, because E-to-E synaptic transmission is rapidly reduced by STD, the curvature of $F(z)$ changes and a stable fixed point is created, thereby allowing excitatory run-away dynamics to terminate and the ensemble dynamics settle into a steady-state at low activity levels (*Figure 2D*). We found that E-to-I STF leads to similar dynamics (*Figure 2E*, left; Appendix 1) with the only difference that this configuration requires inhibition for network stability (*Figure 2E*, right), whereas E-to-E STD stabilizes activity even without inhibition, albeit at physiologically implausibly high activity levels. Importantly, the re-stabilization through either form of STP did not impair an ensemble's ability to amplify stimuli during the initial onset phase.

Crucially, transient amplification in supralinear networks with STP occurs above a critical threshold (*Figure 2—figure supplement 2*), and requires recurrent excitation $J_{EE}$ to be sufficiently strong (*Figure 2—figure supplement 2C, D*). To quantify the amplification ability of these networks, we calculated the ratio of the evoked peak firing rate to the input strength, henceforth called the 'Amplification index'. We found that amplification in STP-stabilized supralinear networks can be orders of magnitude larger than in linear networks with equivalent weights and comparable stabilized supralinear networks (SSNs) without STP (*Figure 2—figure supplement 3*). We stress that the resulting firing rates are parameter-dependent (*Figure 2—figure supplement 4*) and their absolute value can be high due to the high temporal precision of the onset peak and its short duration. In experiments, such high rates manifest themselves as precisely time-locked spikes with millisecond resolution (*DeWeese et al., 2003*; *Wehr and Zador, 2003*; *Bolding and Franks, 2018*; *Gjoni et al., 2018*).

Recent studies suggest that cortical networks operate as inhibition-stabilized networks (ISNs) (*Sanzeni et al., 2020*; *Sadeh and Clopath, 2021*), in which the excitatory network is unstable in the absence of feedback inhibition (*Tsodyks et al., 1997*; *Ozeki et al., 2009*). To that end, we investigated how ensemble re-stabilization relates to the network operating regime at baseline and during stimulation. Whether a network is an ISN or not is mathematically determined by the real part of the

leading eigenvalue of the Jacobian of the excitatory-to-excitatory subnetwork (*Tsodyks et al., 1997*). We computed the leading eigenvalue in our model incorporating STP and referred to it as 'ISN index' (Materials and methods; Appendix 2). We found that in networks with STP the ISN index can switch sign from negative to positive during external stimulation, indicating that the ensemble can transition from a non-ISN to an ISN (*Figure 2F*). Notably, this behavior is distinct from linear network models in which the network operating regime is independent of the input (Materials and methods). Whether this switch between non-ISN to ISN occurred, however, was parameter dependent and we also found network configurations that were already in the ISN regime at baseline and remained ISNs during stimulation (*Figure 2—figure supplement 5*). Thus, re-stabilization was largely unaffected by the network state and consistent with experimentally observed ISN states (*Sanzeni et al., 2020*).

Theoretical studies have shown that one defining characteristic of ISNs in static excitatory and inhibitory networks is that injecting excitatory (inhibitory) current into inhibitory neurons decreases (increases) inhibitory firing rates, which is also known as the paradoxical effect (*Tsodyks et al., 1997*; *Miller and Palmigiano, 2020*). Yet, it is unclear whether in networks with STP, inhibitory stabilization implies paradoxical response and vice versa. We therefore analyzed the condition of being an ISN and the condition of having paradoxical response in networks with STP (Materials and methods; Appendix 2; Appendix 3). Interestingly, we found that in networks with E-to-E STD, the paradoxical effect implies inhibitory stabilization, whereas inhibitory stabilization does not necessarily imply paradoxical response (*Figure 2G*; Materials and methods), suggesting that having paradoxical effect is a sufficient but not necessary condition for being an ISN. In contrast, in networks with E-to-I STF, inhibitory stabilization and paradoxical effect imply each other (Appendix 2; Appendix 3). Therefore, paradoxical effect can be exploited as a proxy for inhibition stabilization for networks with STP we considered here. By injecting excitatory current into the inhibitory population, we found that the network did not exhibit the paradoxical effect before stimulation (*Figure 2H*, left; *Figure 2—figure supplement 6*). In contrast, injecting excitatory inputs into the inhibitory population during stimulation reduced their activity (*Figure 2H*, right; *Figure 2—figure supplement 6*). As demonstrated in our analysis, non-paradoxical response does not imply non-ISN (*Figure 2—figure supplement 7*; Materials and methods). We therefore examined the inhibition stabilization property of the ensemble by probing the ensemble behavior when a small transient perturbation to excitatory population activity is introduced while inhibition is frozen before stimulation and during stimulation. Before stimulation, the firing rate of the excitatory population slightly increases and then returns to its baseline after the transient perturbation (*Figure 2—figure supplement 8*). During stimulation, however, the transient perturbation leads to a transient explosion of the excitatory firing rate (*Figure 2—figure supplement 8*). These results further confirm that the ensemble shown in our example is initially a non-ISN before stimulation and can transition to an ISN with stimulation. By elevating the input level at the baseline in the model, the ensemble can be initially an ISN (*Figure 2—figure supplement 5*), resembling recent studies revealing that cortical circuits in the mouse V1 operate as ISNs in the absence of sensory stimulation (*Sanzeni et al., 2020*).

Despite the fact that the supralinear input-output function of our framework captures some aspects of intracellular recordings (*Priebe et al., 2004*), it is unbounded and thus allows infinitely high firing rates. This is in contrast to neurobiology where firing rates are bounded due to neuronal refractory effects. While this assumption permitted us to analytically study the system and therefore to gain a deeper understanding of the underlying ensemble dynamics, we wondered whether our main conclusions were also valid when we limited the maximum firing rates. To that end, we carried out the same simulations while capping the firing rate at 300 Hz. In the absence of additional SFA or STP mechanisms, the firing rate saturation introduced a stable high-activity state in the ensemble dynamics which replaced the unstable dynamics in the uncapped model. As above, the ensemble entered this high-activity steady-state when stimulated with an external input above a critical threshold and exhibited persistent activity after stimulus removal (*Figure 2—figure supplement 9*). While weak SFA did not change this behavior, strong SFA resulted in oscillatory behavior during stimulation consistent with previous analytical work (*Figure 2—figure supplement 9*, *van Vreeswijk and Hansel, 2001*), but did not in stable steady-states commonly observed in biological circuits. In the presence of E-to-E STD or E-to-I STF, however, the ensemble exhibited transient evoked activity at stimulation onset that was comparable to the uncapped case. Importantly, the ensemble did not show persistent activity after the stimulation (*Figure 2—figure supplement 9*). Finally, we confirmed that all of these findings were

qualitatively similar in a realistic spiking neural network model (*Figure 2—figure supplement 10*; Materials and methods).

In summary, we found that neuronal ensembles can rapidly, nonlinearly, and transiently amplify inputs by briefly switching from stable to unstable dynamics before being re-stabilized through STP mechanisms. We call this mechanism nonlinear transient amplification (NTA) which, in contrast to balanced amplification (*Murphy and Miller, 2009*; *Hennequin et al., 2012*), arises from population dynamics with supralinear neuronal activation functions interacting with STP. While we acknowledge that there may be other nonlinear transient amplification mechanisms, in this article we restrict our analysis to the definition above. NTA is characterized by a large onset response, a subsequent ISN steady-state while the stimulus persists, and a return to a unique baseline activity state after the stimulus is removed. Thus, NTA is ideally suited to rapidly and nonlinearly amplify sensory inputs through recurrent excitation, like reported experimentally (*Ko et al., 2011*; *Cossell et al., 2015*), while avoiding persistent activity.

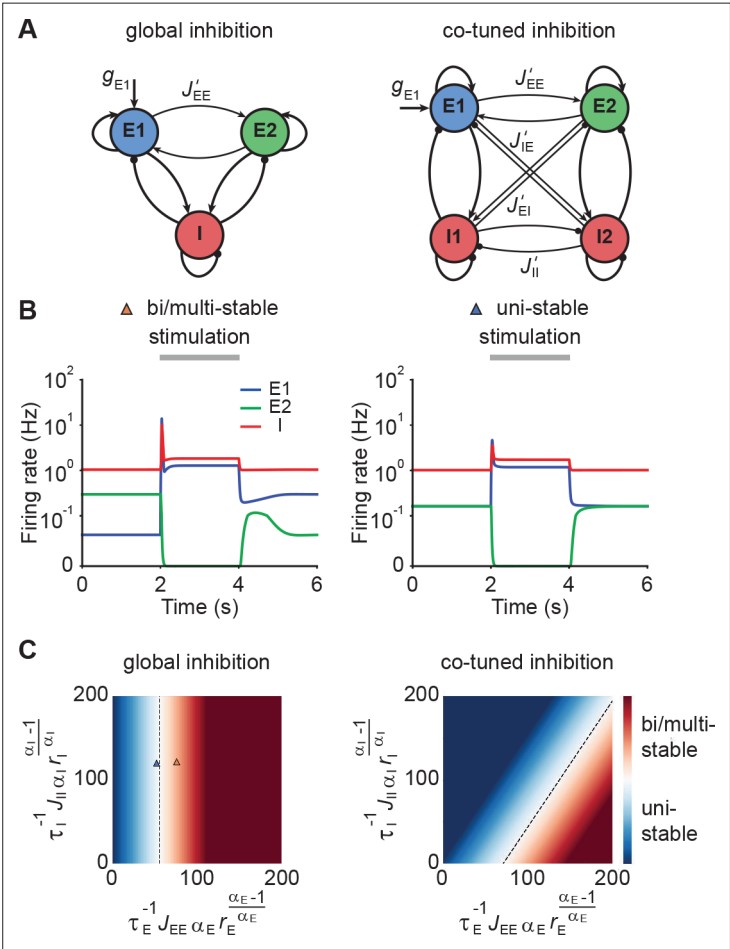

**Figure 3.** Co-tuned inhibition broadens the parameter regime of NTA in the absence of persistent activity. (**A**) Schematic of two neuronal ensembles with global inhibition (left) and with co-tuned inhibition (right). (**B**) Firing rate dynamics of bi/multi-stable ensemble dynamics (left) and uni-stable (right). In both cases, additional excitatory inputs are injected into excitatory ensemble E1 during the period marked in gray. (**C**) Analytical solution of uni- and bi/multi-stability regions for global inhibition (left) and co-tuned inhibition (right). Co-tuning results in a larger parameter regime of uni-stability. The triangles correspond to the two examples in B.

The online version of this article includes the following figure supplement(s) for figure 3:

**Figure supplement 1.** Ensembles with co-tuned inhibition exhibit weaker — but still strong — NTA in comparison to ensembles with global inhibition.

## Co-tuned inhibition broadens the parameter regime of NTA in the absence of persistent activity

Up to now, we have focused on a single neuronal ensemble. However, to process information in the brain, several ensembles with different stimulus selectivity presumably coexist and interact in the same circuit. This coexistence creates potential problems. It can lead to multi-stable persistent attractor dynamics, which are not commonly observed and could have adverse effects on the processing of subsequent stimuli. One solution to this issue could be EI co-tuning, which arises in network models with plastic inhibitory synapses (*Vogels et al., 2011*) and has been observed experimentally in several sensory systems (*Wehr and Zador, 2003*; *Froemke et al., 2007*; *Okun and Lampl, 2008*; *Rupprecht and Friedrich, 2018*).

To characterize the conditions under which neuronal ensembles nonlinearly amplify stimuli without persistent activity, we analyzed the case of two interacting ensembles. More specifically, we considered networks with two excitatory ensembles and distinguished between global and co-tuned inhibition (*Figure 3A*). In the case of global inhibition, one inhibitory population non-specifically inhibits both excitatory populations (*Figure 3A*, left). In contrast, in networks with co-tuned inhibition, each ensemble is formed by a dedicated pair of an excitatory and an inhibitory population which can have cross-over connections, for instance, due to overlapping ensembles (*Figure 3A*, right).

Global inhibition supports winner-take-all competition and is therefore often associated with multi-stable attractor dynamics (*Wong and Wang, 2006*; *Mongillo et al., 2008*). We first illustrated this effect in a network model with global inhibition. When the recurrent excitatory connections within each ensemble were sufficiently strong, small amounts of noise in the initial condition led to one of the ensembles spontaneously activating at elevated firing rates, while the other ensemble's activity remained low (*Figure 3B*, left). A specific external stimulation could trigger a switch from one state to the other in which the other ensemble was active at a high firing rate. Importantly, this change persisted even after the stimulus had been removed, a hallmark of multi-stable dynamics. In contrast, uni-stable systems have a global symmetric state in which both ensembles have the same activity in the absence of stimulation. While the stimulated ensemble showed elevated firing rates in response to the stimulus, its activity returned to the baseline level after the stimulus is removed (*Figure 3B*, right), consistent with experimental observations (*DeWeese et al., 2003*; *Rupprecht and Friedrich, 2018*; *Bolding and Franks, 2018*). Note that the only difference between these two models is that $J_{EE}$ is larger in the multi-stable example than in the uni-stable one.

Symmetric baseline activity is most consistent with activity observed in sensory areas. Hence, we sought to understand which inhibitory connectivity would be most conducive to maintain it. To that end, we analytically identified the uni-stability conditions, which are determined by the leading eigenvalue of the Jacobian matrix of the system, for networks with varying degrees of EI co-tuning (Materials and methods). We found that a broader parameter regime underlies uni-stability in networks with co-tuned inhibition than global inhibition (*Figure 3C*). Notably, this conclusion is general and extends to networks with an arbitrary number of ensembles (Materials and methods). In comparison to the ensemble with global inhibition, the ensemble with co-tuned inhibition exhibits weaker — but still strong — NTA (*Figure 3—figure supplement 1*). Thus, co-tuned inhibition broadens the parameter regime in which NTA is possible while simultaneously avoiding persistent attractor dynamics.

## NTA provides better pattern completion than fixed points while retaining stimulus selectivity

Neural circuits are capable of generating stereotypical activity patterns in response to partial cues and forming distinct representations in response to different stimuli (*Carrillo-Reid et al., 2016*; *Marshel et al., 2019*; *Bolding et al., 2020*; *Vinje and Gallant, 2000*; *Cayco-Gajic and Silver, 2019*). To test whether NTA achieves pattern completion while retaining stimulus selectivity, we analyzed the transient onset activity in our models and compared it to the fixed point activity.

To investigate pattern completion and stimulus selectivity in our model, we considered a co-tuned network with E-to-E STD and two distinct excitatory ensembles $E1$ and $E2$. We gave additional input $g_{E1}$ to a Subset 1, consisting of 75% of the neurons in ensemble $E1$ (*Figure 4A*). We then measured the evoked activity in the remaining 25% of the excitatory neurons in $E1$ to quantify pattern completion. To assess stimulus selectivity, we injected additional input $g_{E1}$ into the entire $E1$ ensemble during the second stimulation phase (*Figure 4A*) while measuring the activity of $E2$. We found that neurons

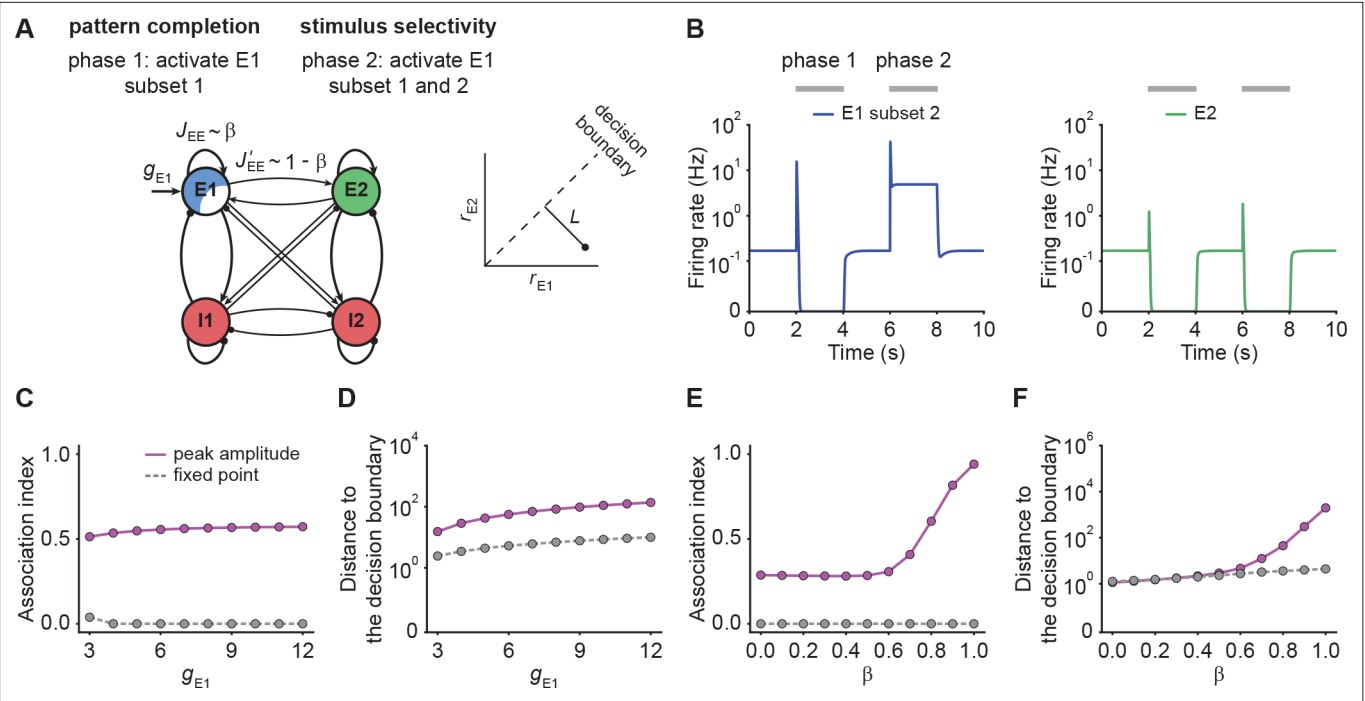

**Figure 4.** NTA yields stronger pattern completion than fixed points while retaining stimulus selectivity. (**A**) Schematic of the network setup used to probe pattern completion and stimulus selectivity. To assess the effect on pattern completion, 75% of the neurons (Subset 1) in ensemble E1 received additional input $g_{E1}$ during Phase one (2–4 s), while we recorded the firing rate of the remaining 25% (Subset 2) in the excitatory ensemble E1. To evaluate the impact on stimulus selectivity, all neurons in E1 received additional inputs $g_{E1}$ in Phase two (6–8 s) while the firing rate of E2 was measured. A downstream neuron's ability to discriminate between E1 or E2 being active depends on whether their activity is well separated by a symmetric decision boundary (inset). (**B**) Examples of firing rates of Subset 2 of E1 (left, blue) and E2 (right, green) with E-to-E STD. (**C**) Association index as a function of input $g_{E1}$ for the onset peak amplitude (magenta solid line) and fixed point activity (gray dashed line) for E-to-E STD. (**D**) Distance to the decision boundary (see panel A, inset) as a function of input $g_{E1}$ for the onset peak amplitude (magenta solid line) and fixed point activity (gray dashed line) for E-to-E STD. (**E and F**) Same as C and D but as a function of β, which controls the inner- and inter-ensemble connection strength.

The online version of this article includes the following figure supplement(s) for figure 4:

**Figure supplement 1.** Change in steady state activity for unstimulated co-tuned neurons in the rate-based model.

**Figure supplement 2.** Quantification of pattern completion and stimulus selectivity in networks with E-to-I STF.

in Subset 2, which did not receive additional input, showed large onset responses, their steady-state activity was largely suppressed (**Figure 4B**). Despite the fact that inputs to $E1$ caused increased transient onset responses in $E2$, the amount of increase was orders of magnitude smaller than in $E1$ (**Figure 4B**). To quantify pattern completion, we defined the

$$\text{Association index} = 1 + \frac{r_{E1_2} - r_{E1_1}}{r_{E1_2} + r_{E1_1}} \quad . \tag{6}$$

Here, $r_{E1_1}$ and $r_{E1_2}$ correspond to the subpopulation activities of $E1$, respectively. By definition, the Association index ranges from zero to one, with larger values indicating stronger associativity. In addition, to quantify the selectivity between $E1$ and $E2$, we considered a symmetric binary classifier (**Figure 4A**, inset) and measured the distance to the decision boundary (Materials and methods). Note that the Association index was computed during Phase one and the distance to the decision boundary during Phase two in this simulation paradigm (**Figure 4B**).

With these definitions, we ran simulations with different input strengths $g_{E1}$. We found that the onset peaks showed stronger association than the fixed point activity (**Figure 4C**). Note that the Association index at the fixed point remained zero, a direct consequence of $r_{E1_2}$ being suppressed to zero. Furthermore, we found that the distance between the transient onset response and the decision boundary was always greater than for the fixed point activity (**Figure 4D**) showing that onset responses retain stimulus selectivity. While the fixed point activity of the unstimulated co-tuned neurons is zero in the given example, stimulating a subset of neurons in one ensemble can lead to an increase in

the fixed point activity of the unstimulated neurons in the same ensemble under certain conditions (*Figure 4—figure supplement 1*; Appendix 4), which is consistent with pattern completion experiments (*Carrillo-Reid et al., 2016*; *Marshel et al., 2019*) showing that unstimulated neurons from the same ensemble can remain active throughout the whole stimulation period.

To investigate how the recurrent excitatory connectivity affects both pattern completion and stimulus selectivity, we introduced the parameter $\beta$ which controls recurrent excitatory tuning by trading off within-ensemble E-to-E strength $J_{EE}$ relative to the inter-ensemble strength $J'_{EE}$ (*Figure 4A*) such that $J_{EE} = \beta J_{tot}$ and $J'_{EE} = (1 - \beta)J_{tot}$. These definitions ensure that the total weight $J_{tot} = J_{EE} + J'_{EE}$ remains constant for any choice of $\beta$. Notably, the overall recurrent excitation strength within an ensemble $J_{EE}$ increases with increasing $\beta$. When $\beta$ is larger than 0.5, the excitatory connection strength within the ensemble $J_{EE}$ exceeds the one between ensembles $J'_{EE}$.

We found that pattern completion ability monotonically increases with $\beta$ with a pronounced onset for $\beta > 0.6$ where NTA takes hold (*Figure 4E*). Moreover, in this regime the two stimulus representations are well separated (*Figure 4F*) which ensures stimulus selectivity also during onset transients. Together, these findings recapitulate the point that recurrent excitatory tuning is a key determinant of network dynamics. Finally, we confirmed that our findings were also valid in networks with E-to-I STF (*Figure 4—figure supplement 2*), which is commonly observed in the brain (*Markram et al., 1998*; *Zucker and Regehr, 2002*; *Pala and Petersen, 2015*). In summary, NTA's transient onset responses maintain stimulus selectivity and result in overall better pattern completion than fixed point activity.

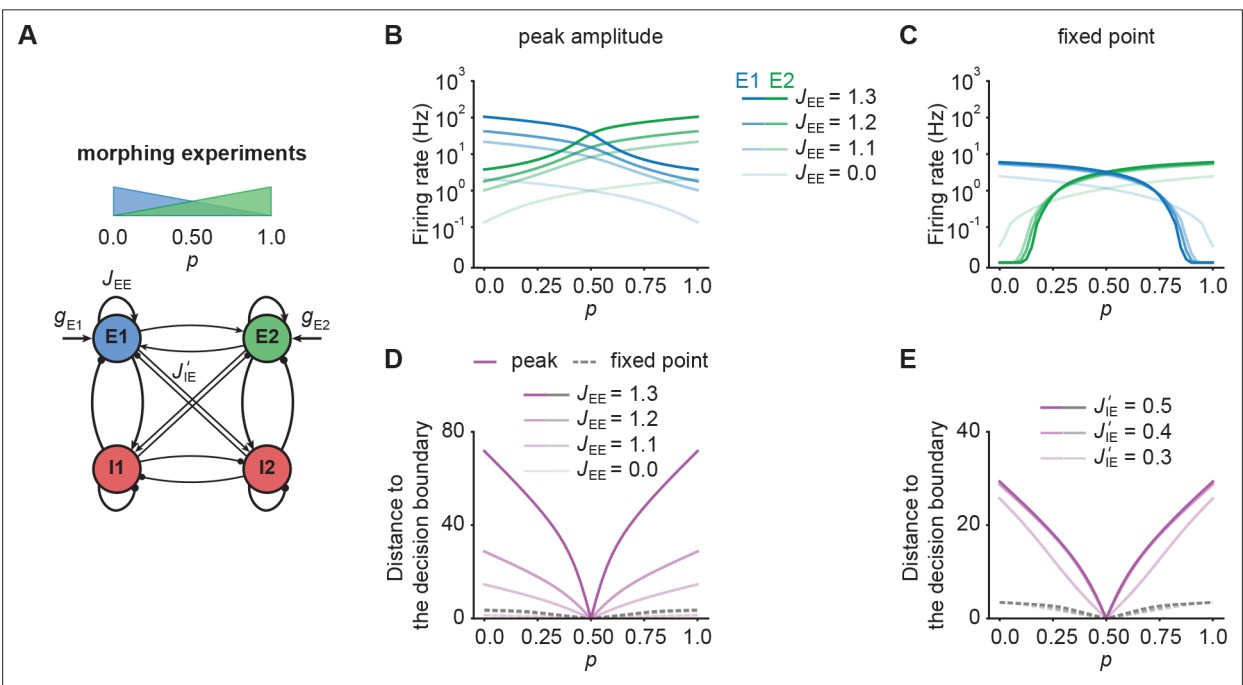

**Figure 5.** NTA provides stronger amplification and pattern separation in morphing experiments than fixed point activity. (**A**) Schematic of the morphing stimulation paradigm. The fraction of the additional inputs into the two excitatory ensembles is controlled by the parameter p. (**B**) Peak amplitude of E1 (blue) and E2 (green) as a function of p for E-to-E STD. Brightness levels represent different recurrent E-to-E connection strengths $J_{EE}$. (**C**) Same as in B but for fixed point activity. (**D**) Distance to the decision boundary as a function of p for the peak onset response (magenta solid line) and fixed point activity (gray dashed line) for E-to-E STD in a network with $J'_{IE} = 0.4$. (**E**) Same as D but with different E-to-I connection strengths $J'_{IE}$ across ensembles for a network with $J_{EE} = 1.2$.

The online version of this article includes the following figure supplement(s) for figure 5:

**Figure supplement 1.** Quantification of pattern separation in morphing experiments using a normalized measure.

**Figure supplement 2.** Quantification of pattern separation in morphing experiments for networks with E-to-I STF.

## NTA provides higher amplification and pattern separation in morphing experiments

So far, we only considered input to one ensemble. To examine how representations in our model are affected by ambiguous inputs to several ensembles, we performed additional morphing experiments (*Freedman et al., 2001*; *Niessing and Friedrich, 2010*). To that end, we introduced the parameter $p$ which interpolates between two input stimuli which target $E1$ and $E2$ respectively. When $p$ is zero, all additional input is injected into $E1$. For $p$ equal to one, all additional input is injected into $E2$. Finally, $p$ equal to 0.5 corresponds to the symmetric case in which $E1$ and $E2$ receive the same amount of additional input (*Figure 5A*).

First, we investigated how the recurrent excitatory connection strength within each ensemble $J_{EE}$ affects the onset peak amplitude and fixed point activity. We found that the peak amplitudes depend strongly on $J_{EE}$, whereas the fixed point activity was only weakly dependent on $J_{EE}$ (*Figure 5B and C*). When we disconnected the ensembles by completely eliminating all recurrent excitatory connections, activity was noticeably decreased (*Figure 5B and C*). This illustrates, that recurrent excitation does play an important role in selectively amplifying specific stimuli similar to experimental observations (*Marshel et al., 2019*; *Peron et al., 2020*), but that amplification is highest at the onset.

Further, we examined the impact of competition through lateral inhibition as a function of the E-to-I inter-ensemble strength $J'_{IE}$ (Materials and methods). As above, we quantified its impact by measuring the representational distance to the decision boundary for the transient onset responses and fixed point activity. We found that regardless of the specific STP mechanism, the distance was larger for the onset responses than for the fixed point activity, consistent with the notion that the onset dynamics separate stimulus identity reliably (*Figure 5D and E*). Since the absolute activity levels between onset and fixed point differed substantially, we further computed the relative pattern Separation index $(r_{E2} - r_{E1})/(r_{E1} + r_{E2})$ and found that the onset transient provides better pattern separation ability for ambiguous stimuli with $p$ close to 0.5 (*Figure 5—figure supplement 1*) provided that the E-to-I connection strength across ensembles $J'_{IE}$ is strong enough. All the while separability for the onset transient was slightly decreased for distinct inputs with $p \in \{0, 1\}$ in comparison to the fixed point. In contrast, fixed points clearly separated such pure stimuli while providing weaker pattern separation for ambiguous input combinations. Importantly, these findings qualitatively held for networks with NTA mediated by E-to-I STF (*Figure 5—figure supplement 2*). Thus, NTA provides stronger amplification and pattern separation than fixed point activity in response to ambiguous stimuli.

## NTA in spiking neural networks

Thus far, our analysis relied on power law neuronal input-output functions in the interest of analytical tractability. To test whether our findings also qualitatively apply to more realistic network models, we built a spiking neural network consisting of randomly connected 800 excitatory and 200 inhibitory neurons, in which the E-to-E synaptic connections were subject to STD (Materials and methods). Here, we defined five overlapping ensembles, each corresponding to 200 randomly selected excitatory neurons. During an initial simulation phase (0–22 s), we consecutively stimulated each ensemble by giving additional input to their excitatory neurons, whereas the input to other neurons remained unchanged (*Figure 6A*). In addition, we also tested pattern completion by stimulating only 75% (Subset 1) of the neurons belonging to Ensemble 5 (22–24 s; *Figure 6A*). We quantified each ensemble's activity by calculating the population firing rate of the ensemble (Materials and methods). As in the case of the rate-based model, the neuronal ensembles in the spiking model generated pronounced transient onset responses. We then measured the difference of peak ensemble activity and fixed point activity between the stimulated ensemble and the remaining unstimulated ensembles (Materials and methods). As for the rate-based networks, this difference was consistently larger for the onset peak than for the fixed point (*Figure 6B and C*). Thus, transient onset responses allow better stimulus separation than fixed points also in spiking neural network models.

Finally, to visualize the neural activity, we projected the binned spiking activity during the first 10 s of our simulation onto its first two principal components. Notably, the PC trajectory does not exhibit a pronounced rotational component (*Figure 6D*) as activity is confined to one specific ensemble, consistent with experiments (*Marshel et al., 2019*). Furthermore, we computed the fifth ensemble's activity for Subset 1 and 2 during the time interval 16–26 s. In agreement with our rate models, neurons in Subset 2 which did not receive additional inputs showed a strong response at the onset (*Figure 6E*),

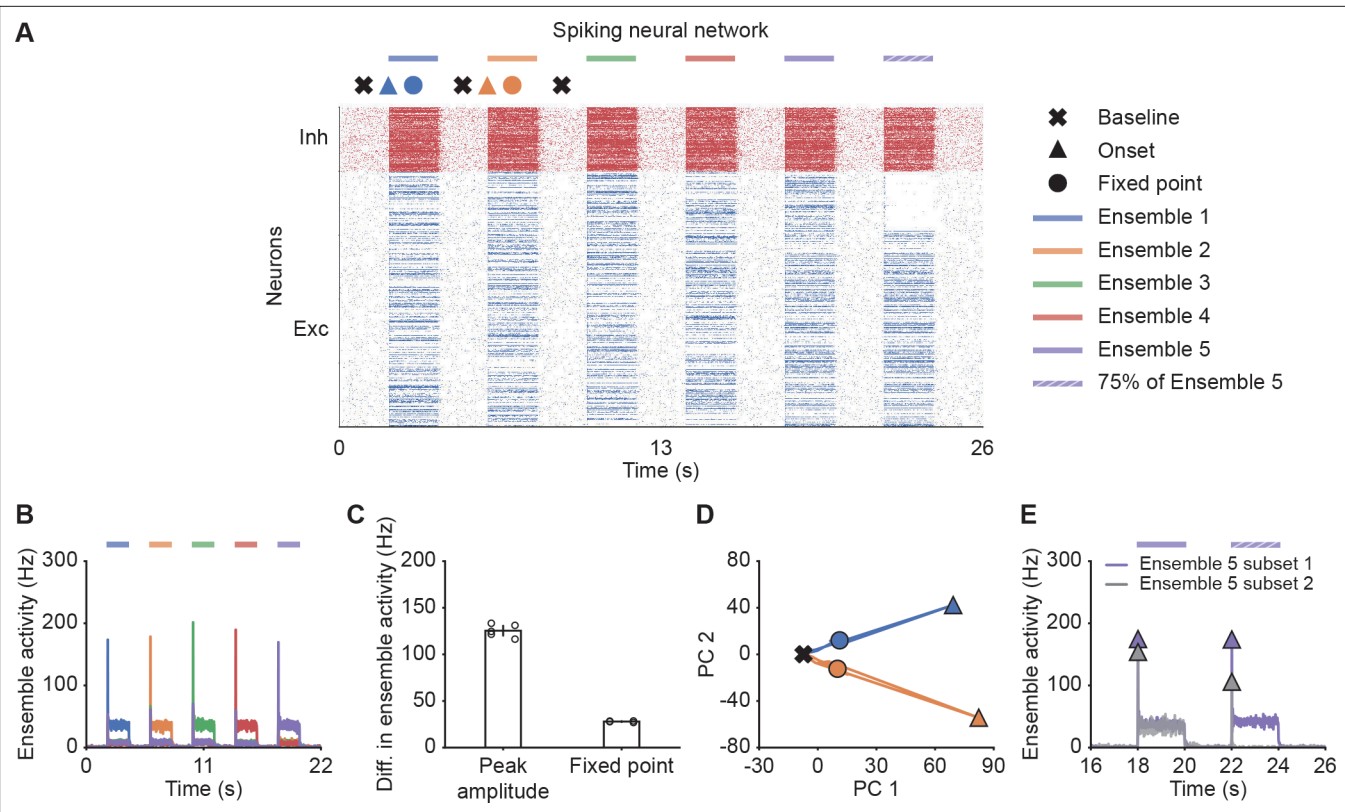

**Figure 6.** Spiking neural network simulations qualitatively reproduce NTA dynamics of rate models. (**A**) Spiking activity of excitatory (blue) and inhibitory (red) neurons in a spiking neural network. From 2 to 20 s, Ensembles 1–5 individually received additional input for 2 s each (colored bars). From 22 to 24 s, 75% of Ensemble 5 neurons (Subset 1) received additional input, whereas the rest 25% of Ensemble 5 neurons (Subset 2) did not receive additional input. The symbols at the top designate the different simulation phases of baseline activity, the onset transients, and the fixed point activity. Different colors correspond to the distinct stimulation periods. (**B**) Ensemble activity (colors). (**C**) Difference in ensemble activity between the stimulated ensemble with the remaining ensembles for the transient onset peak and the fixed point. Points correspond to the different stimulation periods. (**D**) Spiking activity during the interval 0–10 s represented in the PCA basis spanned by the first two principal components which captured approximately 40% of the total variance. The colored lines represent the PC trajectories of the first two stimuli shown in A and B. Triangles, points and crosses correspond to the onset peak, fixed point, and baseline activity, respectively. (**E**) Ensemble activity of Subset 1 (purple) and Subset 2 (gray) of Ensemble 5 from 16 to 26 s. Onset peaks are marked by triangles.

The online version of this article includes the following figure supplement(s) for figure 6:

**Figure supplement 1.** Change in steady state activity for unstimulated co-tuned neurons in spiking neural networks.

but not at the fixed point, suggesting that the strongest pattern completion occurs during the initial amplification phase. Finally, we also observed higher-than-baseline fixed point activity in unstimulated neurons of Subset 2 in spiking neural networks (*Figure 6—figure supplement 1*). Thus, the key characteristics of NTA are preserved across rate-based and more realistic spiking neural network models.

## Discussion

In this study, we demonstrated that neuronal ensemble models with recurrent excitation and suitable forms of STP exhibit nonlinear transient amplification (NTA), a putative mechanism underlying selective amplification in recurrent circuits. NTA combines a supralinear neuronal transfer function, recurrent excitation between neurons with similar tuning, and pronounced STP. Using analytical and numerical methods, we showed that NTA generates rapid transient onset responses during which optimal stimulus separation occurs rather than at steady-states. Additionally, we showed that co-tuned inhibition is conducive to prevent the emergence of persistent activity, which could otherwise interfere with processing subsequent stimuli. In contrast to balanced amplification (*Murphy and Miller, 2009*), NTA is an intrinsically nonlinear mechanism for which only stimuli above a critical threshold are

amplified effectively. While the precise threshold value is parameter-dependent, it can be arbitrarily low provided the excitatory recurrent connections are sufficiently strong (*Figure 1F*). Importantly, such a critical activation threshold offers a possible explanation for sensory perception experiments which show similar threshold behavior (*Marshel et al., 2019*; *Peron et al., 2020*). Following transient amplification, ensemble dynamics are inhibition-stabilized, which renders our model compatible with existing work on SSNs (*Ahmadian et al., 2013*; *Rubin et al., 2015*; *Hennequin et al., 2018*; *Kraynyukova and Tchumatchenko, 2018*; *Echeveste et al., 2020*). Thus, NTA provides a parsimonious explanation for why sensory systems may rely upon neuronal ensembles with recurrent excitation in combination with EI co-tuning, and pronounced STP dynamics.

Several theoretical studies approached the problem of transient amplification in recurrent neural network models. *Loebel and Tsodyks, 2002* have described an NTA-like mechanism as a driver for powerful ensemble synchronization in rate-based networks and in spiking neural network models of auditory cortex (*Loebel et al., 2007*). Here, we generalized this work to both E-to-E STD and E-to-I STF and provide an in-depth characterization of its amplification capabilities, pattern completion properties, and the resulting network states with regard to their inhibition-stabilization properties. Moreover, we showed that SFA cannot provide similar network stabilization and explored how EI co-tuning interacts with NTA. Finally, we contrasted NTA to alternative transient amplification mechanisms. Balanced amplification is a particularly well-studied transient amplification mechanism (*Murphy and Miller, 2009*; *Goldman, 2009*; *Hennequin et al., 2014*; *Bondanelli and Ostojic, 2020*; *Gillett et al., 2020*; *Christodoulou et al., 2021*) that relies on non-normality of the connectivity matrix to selectively and rapidly amplify stimuli. Importantly, balanced amplification occurs in networks in which strong recurrent excitation is appropriately balanced by strong recurrent inhibition. It is capable of generating rich transient activity in linear network models (*Hennequin et al., 2014*), and selectively amplifies specific activity patterns, but without a specific activation threshold. In addition, in spiking neural networks, strong input can induce synchronous firing at the population level which is subsequently stabilized by strong feedback inhibition without the requirement for STP mechanisms (*Stern et al., 2018*). These properties contrast with NTA, which has a nonlinear activation threshold and intrinsically relies on STP to stabilize otherwise unstable run-away dynamics. Due to the switch of the network's dynamical state, NTA's amplification can be orders of magnitudes larger than balanced amplification (*Figure 2—figure supplement 3*). Interestingly, after the transient amplification phase, ensemble dynamics settle in an inhibitory-stabilized state, which renders NTA compatible with previous work on SSNs but in the presence of STP. Finally, although NTA and balanced amplification rely on different amplification mechanisms, they are not mutually exclusive and could, in principle, co-exist in biological networks.

NTA's requirement to generate positive feedback dynamics through recurrent excitation, motivated our focus on networks with $\det(\mathbf{J}) < 0$. As demonstrated in previous work (*Ahmadian et al., 2013*), supralinear networks with $\det(\mathbf{J}) > 0$ and instantaneous inhibition ($\tau_I/\tau_E \to 0$) are always stable for any given input, they are thus unable to generate positive feedback dynamics. In addition, networks with $\det(\mathbf{J}) > 0$ can exhibit a range of interesting behaviors, for example, oscillatory dynamics and persistent activity (*Kraynyukova and Tchumatchenko, 2018*). It is worth noting, however, that for delayed or slow inhibition, stimulation can still lead to unstable network dynamics in networks with $\det(\mathbf{J}) > 0$. Nevertheless, our simulations suggest that our main conclusions about the stabilization mechanisms still hold (*Figure 2—figure supplement 11*).

NTA shares some properties with the notion of network criticality in the brain, like synchronous activation of cell ensembles (*Plenz and Thiagarajan, 2007*) and STP which can tune networks to a critical state (*Levina et al., 2007*). However, in contrast to most models of criticality, in NTA an ensemble briefly transitions to supercritical dynamics in a controlled, stimulus-dependent manner rather than spontaneously. Yet, how the two paradigms are connected at a more fundamental level, is an intriguing question left for future work. Furthermore, recurrent co-tuned inhibition is essential for NTA to ensure uni-stability and selectivity through the suppression of ensembles with different tuning. This requirement is similar in flavor to semi-balanced networks characterized by excess inhibition to some excitatory ensembles while others are balanced (*Baker et al., 2020*). However, the theory of semi-balanced networks has, so far, only been applied to steady-state dynamics while ignoring transients and STP. EI co-tuning prominently features in several models and was shown to support network stability (*Vogels et al., 2011*; *Hennequin et al., 2017*; *Znamenskiy et al., 2018*), efficient coding (*Denève and Machens, 2016*), novelty detection (*Schulz et al., 2021*), changes in neuronal variability

(*Hennequin et al., 2018*; *Rost et al., 2018*), and correlation structure (*Wu et al., 2020*). Moreover, some studies have argued that EI balance and co-tuning could increase robustness to noise in the brain (*Rubin et al., 2017*). The present work mainly highlights its importance for preventing multistability and delay activity in circuits not requiring such long-timescale dynamics.

NTA is consistent with several experimental findings. First, our model recapitulates the key findings of *Shew et al., 2015* who showed ex vivo that strong sensory inputs cause a transient shift to a supercritical state, after which adaptive changes rapidly tune the network to criticality. Second, NTA requires strong recurrent excitatory connectivity between neurons with similar tuning, which has been reported in experiments (*Ko et al., 2011*; *Cossell et al., 2015*; *Peron et al., 2020*). Third, ensemble activation in our model depends on a critical stimulus strength in line with recent all-optical experiments in the visual cortex, which further link ensemble activation with a perceptual threshold (*Marshel et al., 2019*). Fourth, sensory networks are uni-stable in that they return to a non-selective activity state after the removal of the stimulus and usually do not show persistent activity (*DeWeese et al., 2003*; *Mazor and Laurent, 2005*; *Rupprecht and Friedrich, 2018*). Fifth, our work shows that NTA's onset responses encode stimulus identity better than the fixed point activity, consistent with experiments in the locust antennal lobe (*Mazor and Laurent, 2005*) and research supporting that the brain relies on coactivity on short timescales to represent information (*Stopfer et al., 1997*; *Engel et al., 2001*; *Harris et al., 2003*; *El-Gaby et al., 2021*). Yet, it remains to be seen whether these findings are also coherent with data on the temporal evolution in other sensory systems. Finally, EI co-tuning, which is conducive for NTA, has been found ubiquitously in different sensory circuits (*Wehr and Zador, 2003*; *Froemke et al., 2007*; *Okun and Lampl, 2008*; *Rupprecht and Friedrich, 2018*; *Znamenskiy et al., 2018*).

In our model, we made several simplifying assumptions. For instance, we kept the input to inhibitory neurons fixed and only varied the input to the excitatory population. This step was motivated by experiments in the piriform cortex where the total inhibition is dominated by feedback inhibition (*Franks et al., 2011*). Nevertheless, significant feedforward inhibition was observed in other areas (*Bissière et al., 2003*; *Cruikshank et al., 2007*; *Ji et al., 2016*; *Miska et al., 2018*). While an in-depth comparison for different origins of inhibition was beyond the scope of the present study, we found that increasing the inputs to the excitatory population and inhibitory population by the same amount can still lead to NTA (*Figure 1—figure supplement 1*; *Figure 2—figure supplement 12*; Materials and methods), suggesting that our main findings can remain unaffected in the presence of substantial feedforward inhibition. In addition, we limited our analysis to only a few overlapping ensembles. It will be interesting future work to study NTA in the case of many interacting and potentially overlapping ensembles and to determine the maximum storage capacity above which performance degrades. Finally, we anticipate that temporal differences in excitatory and inhibitory synaptic transmission may be important to preserve NTA's stimuli selectivity.

Our model makes several predictions. In contrast to balanced amplification, in which the network operating regime depends solely on the connectivity, an ensemble involved in NTA can transition from a non-ISN to an ISN state. Such a transition is consistent with noise variability observed in sensory cortices (*Hennequin et al., 2018*) and could be tested experimentally by probing the paradoxical effect under different stimulation conditions (*Figure 2G–H*; *Figure 2—figure supplement 6*). Moreover, NTA predicts that onset activity provides a better stimulus encoding and its activity is correlated with the fixed point activity. This signature is different from purely non-normal amplification mechanisms which would involve a wave of neuronal activity across several distinct ensembles similar to a synfire chain (*Abeles, 1991*). The difference should be clearly discernible in data. Since NTA relies on recurrent excitation between ensemble neurons, it suggests normal dynamics in which distinct ensembles first activate and then inactivate. The resulting dynamics have weak rotational components (*Figure 6D*) as seen in some experiments (*Marshel et al., 2019*). Strong non-normal amplification, on the other hand, relies on sequential activation associated with pronounced rotational dynamics (*Hennequin et al., 2014*; *Gillett et al., 2020*), as for instance observed in motor areas (*Churchland et al., 2012*). Although both non-normal mechanisms and NTA are likely to co-exist in the brain, we speculate that strong NTA is best suited for, and thus most like to be found in, sensory systems.

In summary, we introduced a general theoretical framework of selective transient signal amplification in recurrent networks. Our approach derives from the minimal assumptions of a nonlinear neuronal transfer function, recurrent excitation within neuronal ensembles, and STP. Importantly, our

analysis revealed the functional benefits of STP and EI co-tuning, both pervasively found in sensory circuits. Finally, our work suggests that transient onset responses rather than steady-state activity are ideally suited for coactivity-based stimulus encoding and provides several testable predictions.

## Materials and methods
### Stability conditions for supralinear networks

The dynamics of a neuronal ensemble consisting of one excitatory and one inhibitory population with a supralinear, power law input-output function can be described as follows:

$$\tau_E \frac{dr_E}{dt} = -r_E + \left[ J_{EE}r_E - J_{EI}r_I + g_E \right]_+^{\alpha_E} \tag{7}$$

$$\tau_I \frac{dr_I}{dt} = -r_I + \left[ J_{IE}r_E - J_{II}r_I + g_I \right]_+^{\alpha_I} \tag{8}$$

The Jacobian $\mathbf{M}$ of the system is given by

$$\mathbf{M} = \begin{bmatrix} \tau_E^{-1}(J_{EE}\alpha_E r_E^{\frac{\alpha_E-1}{\alpha_E}} - 1) & -\tau_E^{-1}J_{EI}\alpha_E r_E^{\frac{\alpha_E-1}{\alpha_E}} \\ \tau_I^{-1}J_{IE}\alpha_I r_I^{\frac{\alpha_I-1}{\alpha_I}} & -\tau_I^{-1}(1 + J_{II}\alpha_I r_I^{\frac{\alpha_I-1}{\alpha_I}}) \end{bmatrix} \tag{9}$$

To ensure that the system is stable, the product of $\mathbf{M}$'s eigenvalues $\lambda_1\lambda_2$, which is equivalent to the determinant of $\mathbf{M}$, has to be positive. In addition, the sum of the two eigenvalues $\lambda_1 + \lambda_2$, which corresponds to tr($\mathbf{M}$), has to be negative. We therefore obtained the following two stability conditions

$$\lambda_1\lambda_2 = -\tau_E^{-1}\tau_I^{-1}(J_{EE}\alpha_E r_E^{\frac{\alpha_E-1}{\alpha_E}} - 1)(1 + J_{II}\alpha_I r_I^{\frac{\alpha_I-1}{\alpha_I}}) + \tau_E^{-1}\tau_I^{-1}J_{EI}\alpha_E r_E^{\frac{\alpha_E-1}{\alpha_E}} J_{IE}\alpha_I r_I^{\frac{\alpha_I-1}{\alpha_I}} > 0 \tag{10}$$

$$\lambda_1 + \lambda_2 = \tau_E^{-1}(J_{EE}\alpha_E r_E^{\frac{\alpha_E-1}{\alpha_E}} - 1) - \tau_I^{-1}(1 + J_{II}\alpha_I r_I^{\frac{\alpha_I-1}{\alpha_I}}) < 0 \tag{11}$$

Notably, the stability conditions depend on the firing rate of the excitatory population $r_E$ and the inhibitory population $r_I$. Since firing rates are input-dependent, the stability of supralinear networks is input-dependent. In contrast, in linear networks in which $\alpha_E = \alpha_I = 1$, the conditions can be simplified to

$$\lambda_1\lambda_2 = -\tau_E^{-1}\tau_I^{-1}(J_{EE} - 1)(1 + J_{II}) + \tau_E^{-1}\tau_I^{-1}J_{EI}J_{IE} > 0 \tag{12}$$

$$\lambda_1 + \lambda_2 = \tau_E^{-1}(J_{EE} - 1) - \tau_I^{-1}(1 + J_{II}) < 0 \tag{13}$$

and are thus input-independent.

### ISN index for supralinear networks

If an ensemble is unstable without feedback inhibition, then the ensemble is an ISN (*Tsodyks et al., 1997*). To determine whether a given system is an ISN, we analyzed the stability of the E-E subnetwork, which is determined by the real part of the leading eigenvalue of the Jacobian of the E-E subnetwork. In the following, we call this leading eigenvalue the 'ISN index', which is defined as follows:

$$\text{ISN index} = \tau_E^{-1}(J_{EE}\alpha_E r_E^{\frac{\alpha_E-1}{\alpha_E}} - 1) \tag{14}$$

A positive ISN index indicates the system is an ISN. Otherwise, the system is non-ISN. For supralinear networks in which $\alpha_E > 1$, the ISN index depends on the firing rates, inputs can therefore switch the network from non-ISN to ISN. In contrast, $\alpha_E = 1$ for linear networks which renders the ISN index firing rate independent.

### Characteristic function

To investigate how network stability changes with input, we trace the steps of *Kraynyukova and Tchumatchenko, 2018* and define the characteristic function $F(z)$ as follows:

$$F(z) = J_{EE}\left[z\right]_+^{\alpha_E} - J_{EI}\left[\det(\mathbf{J}) \cdot J_{EI}^{-1}\left[z\right]_+^{\alpha_E} + J_{EI}^{-1}J_{II}z - J_{EI}^{-1}J_{II}g_E + g_I\right]_+^{\alpha_I} - z + g_E \tag{15}$$

where

$$z = J_{EE}r_E - J_{EI}r_I + g_E \tag{16}$$

is the current into the excitatory population. The characteristic function simplifies the original two-dimensional system to a one-dimensional system, and the zero crossings of $F(z)$ correspond to the fixed points of the original system. For $z \geq 0$, we note:

$$\frac{dF(z)}{dz} = J_{EE}\alpha_E r_E^{\frac{\alpha_E-1}{\alpha_E}} - J_{EI}\alpha_I \left( \det(\mathbf{J}) \cdot J_{EI}^{-1}\alpha_E r_E^{\frac{\alpha_E-1}{\alpha_E}} + J_{EI}^{-1}J_{II} \right) r_I^{\frac{\alpha_I-1}{\alpha_I}} - 1 = -\tau_E\tau_I\lambda_1\lambda_2 \tag{17}$$

Therefore, if the derivative of $F(z)$ evaluated at one of its roots is positive, the corresponding fixed point is a saddle point. Note that as $r_E$ and $r_I$ increase, the term in parenthesis becomes dominant. To ensure that $\lambda_1\lambda_2$ is negative also for large $r_E$ and $r_I$, the determinant of the weight matrix $\det(\mathbf{J})$ has to be positive. Therefore, $\det(\mathbf{J})$ has a decisive impact on the curvature of $F(z)$. In systems with negative determinant, $F(z)$ bends upwards for large $z$. In contrast, $F(z)$ asymptotically bends downwards in systems with positive determinant. Hence, the high-activity steady-state of systems with negative determinant is unstable. In addition, we can simplify the above condition to the determinant of the weight matrix which is a necessary condition for network stability at any firing rate:

$$\det(\mathbf{J}) = -J_{EE}J_{II} + J_{IE}J_{EI} > 0 \tag{18}$$

To investigate how the network stability changes with input $g_E$, we examined how $F(z)$ varies with changing input $g_E$ by calculating the derivative of $F(z)$ with respect to $g_E$,

$$\frac{dF(z)}{dg_E} = \alpha_I J_{II}\left[ \det(\mathbf{J}) \cdot J_{EI}^{-1}[z]_+^{\alpha_E} + J_{EI}^{-1}J_{II}z - J_{EI}^{-1}J_{II}g_E + g_I \right]_+^{\alpha_I-1} + 1 \tag{19}$$

Since $\frac{dF(z)}{dg_E}$ is positive, increasing $g_E$ always shifts $F(z)$ upwards, eventually leading to the vanishing of all roots and, thus, unstable dynamics in supralinear networks with negative $\det(\mathbf{J})$. In scenarios in which feedforward input to the inhibitory population also changes, we have

$$\begin{aligned}
\frac{dF(z)}{dt} &= \frac{\partial F(z)}{\partial g_E}\frac{dg_E}{dt} + \frac{\partial F(z)}{\partial g_I}\frac{dg_I}{dt} \\
&= \left( \alpha_I J_{II}\left[ \det(\mathbf{J}) \cdot J_{EI}^{-1}[z]_+^{\alpha_E} + J_{EI}^{-1}J_{II}z - J_{EI}^{-1}J_{II}g_E + g_I \right]_+^{\alpha_I-1} + 1 \right)\Delta g_E \\
&\quad - \alpha_I J_{EI}\left[ \det(\mathbf{J}) \cdot J_{EI}^{-1}[z]_+^{\alpha_E} + J_{EI}^{-1}J_{II}z - J_{EI}^{-1}J_{II}g_E + g_I \right]_+^{\alpha_I-1}\Delta g_I
\end{aligned} \tag{20}$$

When the change in stimulation strength into the excitatory ($\Delta g_E$) and the inhibitory population ($\Delta g_I$) are the same, $\frac{dF(z)}{dt}$ is always positive provided $J_{II}$ is greater than $J_{EI}$. Hence, depending on the value of $\frac{J_{II}}{J_{EI}}$, stimulation can lead to unstable network dynamics even when the input to the inhibitory population increases more than to the excitatory population.

## Spike-frequency adaptation (SFA)

We modeled SFA of excitatory neurons as an activity-dependent negative feedback current (*Benda and Herz, 2003*; *Brette and Gerstner, 2005*):

$$\tau_E\frac{dr_E}{dt} = -r_E + \left[ J_{EE}r_E - J_{EI}r_I + g_E \right]_+^{\alpha_E} - a \tag{21}$$

$$\tau_I\frac{dr_I}{dt} = -r_I + \left[ J_{IE}r_E - J_{II}r_I + g_I \right]_+^{\alpha_I} \tag{22}$$

$$\tau_a\frac{da}{dt} = -a + br_E \tag{23}$$

where $a$ is the adaptation variable, $\tau_a$ is the adaptation time constant, and $b$ is the adaptation strength.

### Stability conditions in networks with SFA

The Jacobian $\mathbf{M_{SFA}}$ of the system with SFA is given by

$$\mathbf{M_{SFA}} = \begin{bmatrix} \tau_E^{-1}(J_{EE}\alpha_E r_E^{\frac{\alpha_E-1}{\alpha_E}} - 1) & -\tau_E^{-1}J_{EI}\alpha_E r_E^{\frac{\alpha_E-1}{\alpha_E}} & -\tau_E^{-1} \\ \tau_I^{-1}J_{IE}\alpha_I r_I^{\frac{\alpha_I-1}{\alpha_I}} & -\tau_I^{-1}(1+J_{II}\alpha_I r_I^{\frac{\alpha_I-1}{\alpha_I}}) & 0 \\ \tau_a^{-1}b & 0 & -\tau_a^{-1} \end{bmatrix} \tag{24}$$

The characteristic polynomial of the system with SFA can be written as follows (*Horn and Johnson, 1985*):

$$\lambda^3 - \text{tr}(\mathbf{M_{SFA}})\lambda^2 + (A_{11} + A_{22} + A_{33})\lambda - \det(\mathbf{M_{SFA}}) = 0 \tag{25}$$

where $\text{tr}(\mathbf{M_{SFA}})$ and $\det(\mathbf{M_{SFA}})$ are the trace and the determinant of the Jacobian matrix $\mathbf{M_{SFA}}$, $A_{11}$, $A_{22}$, and $A_{33}$ are the matrix cofactors. More specifically,

$$\text{tr}(\mathbf{M_{SFA}}) = \tau_E^{-1}(J_{EE}\alpha_E r_E^{\frac{\alpha_E-1}{\alpha_E}} - 1) - \tau_I^{-1}(1+J_{II}\alpha_I r_I^{\frac{\alpha_I-1}{\alpha_I}}) - \tau_a^{-1} \tag{26}$$

$$A_{11} = \begin{vmatrix} -\tau_I^{-1}(1+J_{II}\alpha_I r_I^{\frac{\alpha_I-1}{\alpha_I}}) & 0 \\ 0 & -\tau_a^{-1} \end{vmatrix} = \tau_I^{-1}(1+J_{II}\alpha_I r_I^{\frac{\alpha_I-1}{\alpha_I}})\tau_a^{-1} \tag{27}$$

$$A_{22} = \begin{vmatrix} \tau_E^{-1}(J_{EE}\alpha_E r_E^{\frac{\alpha_E-1}{\alpha_E}} - 1) & -\tau_E^{-1} \\ \tau_a^{-1}b & -\tau_a^{-1} \end{vmatrix} = -\tau_E^{-1}(J_{EE}\alpha_E r_E^{\frac{\alpha_E-1}{\alpha_E}} - 1)\tau_a^{-1} + \tau_a^{-1}b\tau_E^{-1} \tag{28}$$

$$A_{33} = \begin{vmatrix} \tau_E^{-1}(J_{EE}\alpha_E r_E^{\frac{\alpha_E-1}{\alpha_E}} - 1) & -\tau_E^{-1}J_{EI}\alpha_E r_E^{\frac{\alpha_E-1}{\alpha_E}} \\ \tau_I^{-1}J_{IE}\alpha_I r_I^{\frac{\alpha_I-1}{\alpha_I}} & -\tau_I^{-1}(1+J_{II}\alpha_I r_I^{\frac{\alpha_I-1}{\alpha_I}}) \end{vmatrix}$$

$$= -\tau_E^{-1}(J_{EE}\alpha_E r_E^{\frac{\alpha_E-1}{\alpha_E}} - 1)\tau_I^{-1}(1+J_{II}\alpha_I r_I^{\frac{\alpha_I-1}{\alpha_I}}) + \tau_E^{-1}J_{EI}\alpha_E r_E^{\frac{\alpha_E-1}{\alpha_E}}\tau_I^{-1}J_{IE}\alpha_I r_I^{\frac{\alpha_I-1}{\alpha_I}} \tag{29}$$

$$A_{11} + A_{22} + A_{33} = \tau_I^{-1}(1+J_{II}\alpha_I r_I^{\frac{\alpha_I-1}{\alpha_I}})\tau_a^{-1} - \tau_E^{-1}(J_{EE}\alpha_E r_E^{\frac{\alpha_E-1}{\alpha_E}} - 1)\tau_a^{-1} + \tau_a^{-1}b\tau_E^{-1}$$

$$- \tau_E^{-1}(J_{EE}\alpha_E r_E^{\frac{\alpha_E-1}{\alpha_E}} - 1)\tau_I^{-1}(1+J_{II}\alpha_I r_I^{\frac{\alpha_I-1}{\alpha_I}}) + \tau_E^{-1}J_{EI}\alpha_E r_E^{\frac{\alpha_E-1}{\alpha_E}}\tau_I^{-1}J_{IE}\alpha_I r_I^{\frac{\alpha_I-1}{\alpha_I}} \tag{30}$$

$$\det(\mathbf{M_{SFA}}) = \tau_E^{-1}(J_{EE}\alpha_E r_E^{\frac{\alpha_E-1}{\alpha_E}} - 1)\tau_I^{-1}(1+J_{II}\alpha_I r_I^{\frac{\alpha_I-1}{\alpha_I}})\tau_a^{-1}$$

$$- \tau_E^{-1}J_{EI}\alpha_E r_E^{\frac{\alpha_E-1}{\alpha_E}}\tau_I^{-1}J_{IE}\alpha_I r_I^{\frac{\alpha_I-1}{\alpha_I}}\tau_a^{-1} - \tau_a^{-1}b\tau_E^{-1}\tau_I^{-1}(1+J_{II}\alpha_I r_I^{\frac{\alpha_I-1}{\alpha_I}}) \tag{31}$$

To ensure that the dynamics of the system are stable, the real parts of the eigenvalues of the Jacobian at the fixed point, and thus all roots of the characteristic polynomial have to be negative. Since the product of the roots is equal to $\det(\mathbf{M_{SFA}})$, $-\det(\mathbf{M_{SFA}})$ has to be positive. We then have

$$b > \frac{\alpha_E r_E^{\frac{\alpha_E-1}{\alpha_E}}(J_{EE} - \det(\mathbf{J}) \cdot \alpha_I r_I^{\frac{\alpha_I-1}{\alpha_I}})}{1+J_{II}\alpha_I r_I^{\frac{\alpha_I-1}{\alpha_I}}} - 1 \tag{32}$$

Since SFA does not modify the synaptic connections, the term $J_{EE} - \det(\mathbf{J}) \cdot \alpha_I r_I^{\frac{\alpha_I-1}{\alpha_I}}$ is positive for networks with $\det(\mathbf{J}) < 0$.

In the large $r_E$ limit, if $b$ is small such that the above condition cannot be fulfilled, $\det(\mathbf{M_{SFA}})$ is then positive, suggesting that the Jacobian of the system has always at least one positive eigenvalue. Therefore, the dynamics of the system cannot be stabilized in the presence of small $b$.

In addition, $A_{11} + A_{22} + A_{33}$ is equal to $\lambda_1\lambda_2 + \lambda_2\lambda_3 + \lambda_1\lambda_3$, with the roots of the characteristic polynomial $\lambda_1$, $\lambda_2$, and $\lambda_3$. If all roots are real and negative, $A_{11} + A_{22} + A_{33}$ has to be positive. If one root is real and negative and two other roots are complex conjugates, to ensure that all roots have negative real parts, one necessary condition is $A_{11} + A_{22} + A_{33} > 0$. From the $\text{tr}(\mathbf{M_{SFA}})$ and $\det(\mathbf{M_{SFA}})$ conditions, we have

$$A_{11} + A_{22} + A_{33} > \tau_a^{-1}(-\tau_a^{-1} + b\tau_E^{-1}) - b\tau_E^{-1}\tau_I^{-1}(1+J_{II}\alpha_I r_I^{\frac{\alpha_I-1}{\alpha_I}}) \tag{33}$$

As a result, if $\tau_a^{-1}(-\tau_a^{-1} + b\tau_E^{-1}) - b\tau_E^{-1}\tau_I^{-1}(1 + J_{II}\alpha_I r_I^{\frac{\alpha_I-1}{\alpha_I}}) > 0$, $A_{11} + A_{22} + A_{33}$ is guaranteed to be positive. We therefore have

$$b[\tau_a^{-1}\tau_E^{-1} - \tau_E^{-1}\tau_I^{-1}(1 + J_{II}\alpha_I r_I^{\frac{\alpha_I-1}{\alpha_I}})] > \tau_a^{-2} \tag{34}$$

Note that $\tau_a$ has to be small, in other words, SFA has to be fast, so that $\tau_a^{-1}\tau_E^{-1} - \tau_E^{-1}\tau_I^{-1}(1 + J_{II}\alpha_I r_I^{\frac{\alpha_I-1}{\alpha_I}})$ is positive for arbitrary $r_I$. For positive $\tau_a^{-1}\tau_E^{-1} - \tau_E^{-1}\tau_I^{-1}(1 + J_{II}\alpha_I r_I^{\frac{\alpha_I-1}{\alpha_I}})$, we have

$$b > \frac{\tau_a^{-2}}{\tau_a^{-1}\tau_E^{-1} - \tau_E^{-1}\tau_I^{-1}(1 + J_{II}\alpha_I r_I^{\frac{\alpha_I-1}{\alpha_I}})} \tag{35}$$

Since $\tau_a$ has to be small, the above condition cannot be satisfied for small $b$.

Next, we consider the system with large $b$. Suppose that the firing rate $r_E$ and $r_I$ in the initial network are of order 1, and $b$ is of order $K$, where $K$ is a large number. We therefore have $-\text{tr}(\mathbf{M_{SFA}}) \sim O(1)$, $A_{11} + A_{22} + A_{33} \sim O(K)$, and $-\det(\mathbf{M_{SFA}}) \sim O(K)$. The discriminant of the characteristic polynomial is

$$(-\text{tr}(\mathbf{M_{SFA}}))^2(A_{11} + A_{22} + A_{33})^2 - 4(A_{11} + A_{22} + A_{33})^3 - 4(-\text{tr}(\mathbf{M_{SFA}}))^3(-\det(\mathbf{M_{SFA}}))$$

$$-27(-\det(\mathbf{M_{SFA}}))^2 + 18(-\text{tr}(\mathbf{M_{SFA}}))(A_{11} + A_{22} + A_{33})(-\det(\mathbf{M_{SFA}}))$$

$$= (A_{11} + A_{22} + A_{33})^3 \left[ \frac{(-\text{tr}(\mathbf{M_{SFA}}))^2}{A_{11} + A_{22} + A_{33}} - 4 - \frac{4(-\text{tr}(\mathbf{M_{SFA}}))^3(-\det(\mathbf{M_{SFA}}))}{(A_{11} + A_{22} + A_{33})^3} \right. \tag{36}$$

$$\left. - \frac{27(-\det(\mathbf{M_{SFA}}))^2}{(A_{11} + A_{22} + A_{33})^3} + \frac{18(-\text{tr}(\mathbf{M_{SFA}}))(-\det(\mathbf{M_{SFA}}))}{(A_{11} + A_{22} + A_{33})^2} \right]$$

Clearly, in the large $b$ limit, the discriminant is negative, suggesting that the characteristic polynomial has one real root and two complex conjugate roots (*Irving, 2004*).

As the input $g_E$ increases, the complex conjugate eigenvalues cross the imaginary axis when $\text{tr}(\mathbf{M_{SFA}})(A_{11} + A_{22} + A_{33})$ equals $\det(\mathbf{M_{SFA}})$. As a result, the system undergoes a supercritical Hopf bifurcation. We numerically confirmed that the resulting limit cycle is stable (*Figure 2—figure supplement 1*), consistent with previous work (*van Vreeswijk and Hansel, 2001*). Thus, the system shows oscillatory behavior instead of stable steady state.

## Short-term plasticity (STP)

We modeled E-to-E STD following previous work (*Tsodyks and Markram, 1997*; *Varela et al., 1997*):

$$\tau_E \frac{dr_E}{dt} = -r_E + \left[ xJ_{EE}r_E - J_{EI}r_I + g_E \right]_+^{\alpha_E} \tag{37}$$

$$\tau_I \frac{dr_I}{dt} = -r_I + \left[ J_{IE}r_E - J_{II}r_I + g_I \right]_+^{\alpha_I} \tag{38}$$

$$\frac{dx}{dt} = \frac{1-x}{\tau_x} - U_d x r_E \tag{39}$$

where $x$ is the depression variable, which is limited to the interval $(0, 1)$, $\tau_x$ is the depression time constant, and $U_d$ is the depression rate. The steady-state solution $x^*$ is given by

$$x^* = \frac{1}{1 + U_d r_E \tau_x} \tag{40}$$

Similarly, we modeled E-to-I STF as

$$\tau_E \frac{dr_E}{dt} = -r_E + \left[ J_{EE}r_E - J_{EI}r_I + g_E \right]_+^{\alpha_E} \tag{41}$$

$$\tau_I \frac{dr_I}{dt} = -r_I + \left[ uJ_{IE}r_E - J_{II}r_I + g_I \right]_+^{\alpha_I} \tag{42}$$

$$\frac{du}{dt} = \frac{1-u}{\tau_u} + U_f(U_{max} - u)r_E \tag{43}$$

where $u$ is the facilitation variable constrained to the interval $(1, U_{max})$, $U_{max}$ is the maximal facilitation value, $\tau_u$ is the time constant of STF, and $U_f$ is the facilitation rate. The steady-state solution $u^*$ is given by

$$u^* = \frac{1 + U_f U_{max} r_E \tau_u}{1 + U_f r_E \tau_u} \tag{44}$$

## Stability conditions for networks with E-to-E STD

The Jacobian $\mathbf{M_{STD}}$ of the system with E-to-E STD is given by

$$\mathbf{M_{STD}} = \begin{bmatrix} \tau_E^{-1}(x J_{EE} \alpha_E r_E^{\frac{\alpha_E-1}{\alpha_E}} - 1) & -\tau_E^{-1} J_{EI} \alpha_E r_E^{\frac{\alpha_E-1}{\alpha_E}} & \tau_E^{-1} J_{EE} \alpha_E r_E^{\frac{2\alpha_E-1}{\alpha_E}} \\ \tau_I^{-1} J_{IE} \alpha_I r_I^{\frac{\alpha_I-1}{\alpha_I}} & -\tau_I^{-1}(1 + J_{II} \alpha_I r_I^{\frac{\alpha_I-1}{\alpha_I}}) & 0 \\ -U_d x & 0 & -\tau_x^{-1} - U_d r_E \end{bmatrix} \tag{45}$$

and the characteristic polynomial can be written as follows:

$$\lambda^3 - \text{tr}(\mathbf{M_{STD}})\lambda^2 + (A_{11} + A_{22} + A_{33})\lambda - \det(\mathbf{M_{STD}}) = 0 \tag{46}$$

where $\text{tr}(\mathbf{M_{STD}})$ and $\det(\mathbf{M_{STD}})$ are the trace and the determinant of the Jacobian matrix $\mathbf{M_{STD}}$, $A_{11}$, $A_{22}$, and $A_{33}$ are the matrix cofactors. More specifically,

$$\text{tr}(\mathbf{M_{STD}}) = \tau_E^{-1}(x J_{EE} \alpha_E r_E^{\frac{\alpha_E-1}{\alpha_E}} - 1) - \tau_I^{-1}(1 + J_{II} \alpha_I r_I^{\frac{\alpha_I-1}{\alpha_I}}) - \tau_x^{-1} - U_d r_E \tag{47}$$

In the case of unstable dynamics, $r_E$ goes to infinity due to run-away excitation. However, the depression variable $x$ approaches zero in this limit, as $\lim_{r_E \to \infty} x = \lim_{r_E \to \infty} \frac{1}{1 + U_d r_E \tau_x} = 0$. Therefore, in the large $r_E$ limit, $-\text{tr}(\mathbf{M_{STD}})$ is positive.

$$\begin{aligned} A_{11} + A_{22} + A_{33} = {} & \tau_I^{-1}(1 + J_{II} \alpha_I r_I^{\frac{\alpha_I-1}{\alpha_I}})(\tau_x^{-1} + U_d r_E) \\ & + \tau_E^{-1}(x J_{EE} \alpha_E r_E^{\frac{\alpha_E-1}{\alpha_E}} - 1)(-\tau_x^{-1} - U_d r_E) - \tau_E^{-1} J_{EE} \alpha_E r_E^{\frac{2\alpha_E-1}{\alpha_E}}(-U_d x) \\ & - \tau_E^{-1}(x J_{EE} \alpha_E r_E^{\frac{\alpha_E-1}{\alpha_E}} - 1)\tau_I^{-1}(1 + J_{II} \alpha_I r_I^{\frac{\alpha_I-1}{\alpha_I}}) + \tau_E^{-1} J_{EI} \alpha_E r_E^{\frac{\alpha_E-1}{\alpha_E}} \tau_I^{-1} J_{IE} \alpha_I r_I^{\frac{\alpha_I-1}{\alpha_I}} \end{aligned} \tag{48}$$

Similarly, in the large $r_E$ limit, $A_{11} + A_{22} + A_{33}$ is positive.

$$\begin{aligned} \det(\mathbf{M_{STD}}) = {} & \tau_E^{-1}(x J_{EE} \alpha_E r_E^{\frac{\alpha_E-1}{\alpha_E}} - 1)\tau_I^{-1}(1 + J_{II} \alpha_I r_I^{\frac{\alpha_I-1}{\alpha_I}})(\tau_x^{-1} + U_d r_E) \\ & - \tau_E^{-1} J_{EI} \alpha_E r_E^{\frac{\alpha_E-1}{\alpha_E}} \tau_I^{-1} J_{IE} \alpha_I r_I^{\frac{\alpha_I-1}{\alpha_I}}(\tau_x^{-1} + U_d r_E) - \tau_E^{-1} J_{EE} \alpha_E r_E^{\frac{2\alpha_E-1}{\alpha_E}} U_d x \tau_I^{-1}(1 + J_{II} \alpha_I r_I^{\frac{\alpha_I-1}{\alpha_I}}) \end{aligned} \tag{49}$$

Similarly, in the large $r_E$ limit, $-\det(\mathbf{M_{STD}})$ is positive.

According to the Descartes' rule of signs, the number of positive roots is at most the number of sign changes in the sequences of polynomial's coefficients. Therefore, there are no positive roots for the above characteristic polynomial and the network dynamics can be stabilized by E-to-E STD.

## Characteristic function approximation for networks with E-to-E STD

As demonstrated above, E-to-E STD is able to restabilize the system, there exists a stable steady state for which the STD variable $x$ is constant $x = x^*$. Because $x$ changes slowly compared to the neuronal dynamics, we can approximate it as constant which results in a natural reduction to a 2D system in which the weights with STD are modified. The stability of this 2D system can be readily characterized by the characteristic function $F(z)$ (*Kraynyukova and Tchumatchenko, 2018*), which depends on the previous steady state value of $x$. The characteristic function approximation with E-to-E STD can therefore be written as follows:

$$F(z) = x J_{EE}[z]_+^{\alpha_E} - J_{EI}\left[\det(\mathbf{J_{STD}}) \cdot J_{EI}^{-1}[z]_+^{\alpha_E} + J_{EI}^{-1} J_{II} z - J_{EI}^{-1} J_{II} g_E + g_I\right]_+^{\alpha_I} - z + g_E \tag{50}$$

where

$$\det(\mathbf{J_{STD}}) = \begin{vmatrix} xJ_{EE} & -J_{EI} \\ J_{IE} & -J_{II} \end{vmatrix} = -xJ_{EE}J_{II} + J_{IE}J_{EI} \tag{51}$$

Note that $\det(\mathbf{J_{STD}})$ can now change its sign due to E-to-E STD, the characteristic function can therefore change its bending shape. We used this relation to visualize how E-to-E STD effectively changes the network stability of the reduced system in **Figure 2D**.

## Conditions for ISN in networks with E-to-E STD

Here, we identify the condition of being in the ISN regime in supralinear networks with E-to-E STD. When the level of inhibition is frozen, the Jacobian of the system reduces to the following:

$$\mathbf{M_1} = \begin{bmatrix} \tau_E^{-1}(xJ_{EE}\alpha_E r_E^{\frac{\alpha_E-1}{\alpha_E}} - 1) & \tau_E^{-1}J_{EE}\alpha_E r_E^{\frac{2\alpha_E-1}{\alpha_E}} \\ -U_d x & -\tau_x^{-1} - U_d r_E \end{bmatrix} \tag{52}$$

For the system with frozen inhibition, the dynamics are stable if

$$\mathrm{tr}(\mathbf{M_1}) = \tau_E^{-1}(xJ_{EE}\alpha_E r_E^{\frac{\alpha_E-1}{\alpha_E}} - 1) - \tau_x^{-1} - U_d r_E < 0 \tag{53}$$

and

$$\det(\mathbf{M_1}) = \tau_E^{-1}(xJ_{EE}\alpha_E r_E^{\frac{\alpha_E-1}{\alpha_E}} - 1)(-\tau_x^{-1} - U_d r_E) + \tau_E^{-1}J_{EE}\alpha_E r_E^{\frac{2\alpha_E-1}{\alpha_E}}U_d x > 0 \tag{54}$$

Therefore, if the network is an ISN at the fixed point, the following condition has to be satisfied:

$$x > \min\left(\sqrt{\frac{1}{J_{EE}\alpha_E r_E^{\frac{\alpha_E-1}{\alpha_E}}}}, \frac{\tau_x+\tau_E+\tau_E\tau_x U_d r_E}{\tau_x J_{EE}\alpha_E r_E^{\frac{\alpha_E-1}{\alpha_E}}}\right) \tag{55}$$

Furthermore, we define the largest real part of the eigenvalues of $\mathbf{M_1}$ as the ISN index for networks with E-to-E STD. More specifically,

$$\text{ISN index} = \mathrm{Re}\left[\frac{\tau_E^{-1}(xJ_{EE}\alpha_E r_E^{\frac{\alpha_E-1}{\alpha_E}} - 1) - \tau_x^{-1} - U_d r_E}{2} + \sqrt{\frac{1}{4}(\tau_E^{-1}(xJ_{EE}\alpha_E r_E^{\frac{\alpha_E-1}{\alpha_E}} - 1) + \tau_x^{-1} + U_d r_E)^2 - \tau_E^{-1}J_{EE}\alpha_E r_E^{\frac{2\alpha_E-1}{\alpha_E}}U_d x}\right] \tag{56}$$

## Conditions for paradoxical response in networks with E-to-E STD

Next, we identify the condition of having the paradoxical effect in supralinear networks with E-to-E STD. To that end, we exploit a separation of timescales between the fast neural activity and the slow STP variable. Therefore, set the depression variable to its value at the fixed point corresponding to the fixed point value of $r_E$. The excitatory nullcline is defined as follows

$$\tau_E \frac{dr_E}{dt} = -r_E + \left[\frac{1}{1 + \tau_x U_d r_E}J_{EE}r_E - J_{EI}r_I + g_E\right]_+^{\alpha_E} = 0 \tag{57}$$

For $r_{E,I} > 0$, we have

$$r_I = \frac{\frac{1}{1+\tau_x U_d r_E}J_{EE}r_E - r_E^{\frac{1}{\alpha_E}} + g_E}{J_{EI}} \tag{58}$$

The slope of the excitatory nullcline in the $r_E/r_I$ plane where $x$ axis is $r_E$ and $y$ axis is $r_I$ can be written as follows

$$k_{STD}^E = \frac{1}{J_{EI}}\left(-\frac{J_{EE}}{(1 + \tau_x U_d r_E)^2}\tau_x U_d r_E + \frac{J_{EE}}{1 + \tau_x U_d r_E} - \frac{1}{\alpha_E}r_E^{\frac{1}{\alpha_E}-1}\right) \tag{59}$$

Note that the slope of the excitatory nullcline is nonlinear. To have paradoxical effect, the slope of the excitatory nullcline at the fixed point of the system has to be positive. Therefore, the STD variable $x$ at the fixed point has to satisfy the following condition

$$x > \sqrt{\frac{1}{J_{EE}\alpha_E r_E^{\frac{\alpha_E-1}{\alpha_E}}}} \tag{60}$$

The inhibitory nullcline can be written as follows

$$\tau_I \frac{dr_I}{dt} = -r_I + \left[ J_{IE}r_E - J_{II}r_I + g_I \right]_+^{\alpha_I} = 0 \tag{61}$$

In the region of rates $r_{E,I} > 0$, we have

$$r_I = \frac{J_{IE}r_E - r_I^{\frac{1}{\alpha_I}} + g_I}{J_{II}} \tag{62}$$

The slope of the inhibitory nullcline can be written as follows

$$k_{STD}^I = \frac{J_{IE}}{J_{II} + \frac{1}{\alpha_I}r_I^{\frac{1-\alpha_I}{\alpha_I}}} \tag{63}$$

In addition to the positive slope of the excitatory nullcline, the slope of the inhibitory nullcline at the fixed point of the system has to be larger than the slope of the excitatory nullcline. We therefore have

$$J_{EI}\alpha_E r_E^{\frac{\alpha_E-1}{\alpha_E}} J_{IE}\alpha_I r_I^{\frac{\alpha_I-1}{\alpha_I}} \left( \tau_x^{-1} + U_d r_E \right) > \left( 1 + J_{II}\alpha_I r_I^{\frac{\alpha_I-1}{\alpha_I}} \right) \left( -\frac{J_{EE}U_d r_E}{1 + \tau_x U_d r_E} \alpha_E r_E^{\frac{\alpha_E-1}{\alpha_E}} + \frac{J_{EE}}{1 + \tau_x U_d r_E} \alpha_E r_E^{\frac{\alpha_E-1}{\alpha_E}} (\tau_x^{-1} + U_d r_E) - (\tau_x^{-1} + U_d r_E) \right) \tag{64}$$

The above condition is the same as the stability condition of the determinant of the Jacobian of the system with E-to-E STD (**Eq. (49)**). Therefore, the condition is always satisfied when the system with E-to-E STD is stable.

Based on the condition of being ISN shown in **Eq. (55)** and the condition of having paradoxical effect shown in **Eq. (60)**, we therefore can conclude that in supralinear networks with E-to-E STD, the paradoxical effect implies inhibitory stabilization, whereas inhibitory stabilization does not necessarily imply paradoxical responses. This is consistent with recent work by **Sanzeni et al., 2020**, in which threshold-linear networks with STP have been studied. Here, we showed analytically that the conclusion holds for any rectified power-law activation function with positive $\alpha$.

To visualize the conditions in a two-dimensional plane, we reduced the conditions into a function of $J_{EE}$ and $x$. For **Figure 2G**, $r_E = 1$. In **Figure 2—figure supplement 5** and **Figure 2—figure supplement 8**, the depression variable thresholds above which the network exhibits the paradoxical effect were calculated based on **Eq. (60)**.

## Uni-stability conditions

The system is said to be 'uni-stable', when it has a single stable fixed point. We first identified the uni-stability condition for networks with global inhibition. To that end, we considered a general network with $N$ excitatory populations and $N$ inhibitory populations. To treat this problem analytically, we did not take STP into account in our analysis. The Jacobian matrix of networks with global inhibition $\mathbf{Q}$, can be written as follows,

$$\mathbf{Q} = \begin{bmatrix} \mathbf{J}_{E\leftarrow E} & \mathbf{J}_{E\leftarrow I} \\ \mathbf{J}_{I\leftarrow E} & \mathbf{J}_{I\leftarrow I} \end{bmatrix} \tag{65}$$

where $\mathbf{J}_{E\leftarrow E}$, $\mathbf{J}_{E\leftarrow I}$, $\mathbf{J}_{I\leftarrow E}$, and $\mathbf{J}_{I\leftarrow I}$ are $N$ by $N$ block matrices defined below.

$$\mathbf{J}_{E \leftarrow E} = \begin{bmatrix} a-e & ka & \cdots & ka \\ ka & a-e & \cdots & ka \\ \vdots & \vdots & \ddots & \vdots \\ ka & ka & \cdots & a-e \end{bmatrix} \tag{66}$$

$$\mathbf{J}_{E \leftarrow I} = -b\mathbf{J}_{N,N} \tag{67}$$

$$\mathbf{J}_{I \leftarrow E} = c\mathbf{J}_{N,N} \tag{68}$$

$$\mathbf{J}_{I \leftarrow I} = \begin{bmatrix} -d-f & -d & \cdots & -d \\ -d & -d-f & \cdots & -d \\ \vdots & \vdots & \ddots & \vdots \\ -d & -d & \cdots & -d-f \end{bmatrix} \tag{69}$$

where $a = \tau_E^{-1} J_{EE} \alpha_E [z_E]_+^{\alpha_E - 1}$, $b = \tau_E^{-1} J_{EI} \alpha_E [z_E]_+^{\alpha_E - 1}$, $c = \tau_I^{-1} J_{IE} \alpha_I [z_I]_+^{\alpha_I - 1}$, $d = \tau_I^{-1} J_{II} \alpha_I [z_I]_+^{\alpha_I - 1}$, $e = \tau_E^{-1}$, and $f = \tau_I^{-1}$. Here, $z_E$ and $z_I$ denote the total current into the excitatory and inhibitory population, respectively. Note that all these parameters are non-negative. Parameter $k$ controls the excitatory connection strength across different populations. $\mathbf{J}_{N,N}$ is a $N$ by $N$ matrix of ones.

The eigenvalues of the Jacobian $\mathbf{Q}$ are roots of its characteristic polynomial,

$$\det((\mathbf{J}_{E \leftarrow E} - \lambda \mathbb{1})(\mathbf{J}_{I \leftarrow I} - \lambda \mathbb{1}) - \mathbf{J}_{E \leftarrow I}\mathbf{J}_{I \leftarrow E}) = 0 \tag{70}$$

where $\mathbb{1}$ represents the identity matrix of size $N$. The characteristic polynomial can be expanded to:

$$\left[ (a-e-ka-\lambda)(-f-\lambda) \right]^{N-1} \left[ (a-e+(N-1)ka-\lambda)(-Nd-f-\lambda) + N^2 bc \right] = 0 \tag{71}$$

We therefore had four distinct eigenvalues:

$$\lambda_1 = a - e - ka \tag{72}$$

$$\lambda_2 = -f \tag{73}$$

and

$$\lambda_{3/4} = \frac{1}{2}\left[ (a-e-f-Nd+(N-1)ka) \pm \sqrt{(a-e-f-Nd+(N-1)ka)^2 - 4((-af+ef+kaf) - N(a-e)d - Nkaf - N(N-1)kad + N^2 bc)} \right] \tag{74}$$

Note that the eigenvalues $\lambda_1$ and $\lambda_2$ have an algebraic and geometric multiplicity of $(N-1)$, whereas the eigenvalues $\lambda_3$ and $\lambda_4$ have an algebraic and geometric multiplicity of 1.

In analogy to networks with global inhibition, the Jacobian matrix of networks with co-tuned inhibition $\mathbf{R}$, can be written as

$$\mathbf{R} = \begin{bmatrix} \mathbf{J}_{E \leftarrow E} & \mathbf{J}_{E \leftarrow I} \\ \mathbf{J}_{I \leftarrow E} & \mathbf{J}_{I \leftarrow I} \end{bmatrix} \tag{75}$$

where $\mathbf{J}_{E \leftarrow E}$, $\mathbf{J}_{E \leftarrow I}$, $\mathbf{J}_{I \leftarrow E}$, and $\mathbf{J}_{I \leftarrow I}$ are $N$ by $N$ block matrices defined as follows:

$$\mathbf{J}_{E \leftarrow E} = \begin{bmatrix} a-e & ka & \cdots & ka \\ ka & a-e & \cdots & ka \\ \vdots & \vdots & \ddots & \vdots \\ ka & ka & \cdots & a-e \end{bmatrix} \tag{76}$$

$$\mathbf{J}_{E\leftarrow I} = \begin{bmatrix} -Nb + (N-1)mb & -mb & \cdots & -mb \\ -mb & -Nb + (N-1)mb & \cdots & -mb \\ \vdots & \vdots & \ddots & \vdots \\ -mb & -mb & \cdots & -Nb + (N-1)mb \end{bmatrix} \quad (77)$$

$$\mathbf{J}_{I\leftarrow E} = \begin{bmatrix} Nc - (N-1)mc & mc & \cdots & mc \\ mc & Nc - (N-1)mc & \cdots & mc \\ \vdots & \vdots & \ddots & \vdots \\ mc & mc & \cdots & Nc - (N-1)mc \end{bmatrix} \quad (78)$$

$$\mathbf{J}_{I\leftarrow I} = \begin{bmatrix} -Nd + (N-1)md - f & -md & \cdots & -md \\ -md & -Nd + (N-1)md - f & \cdots & -md \\ \vdots & \vdots & \ddots & \vdots \\ -md & -md & \cdots & -Nd + (N-1)md - f \end{bmatrix} \quad (79)$$

where $m$ controls the degree of co-tuning in the network. If $m = 0$, the network decouples into $N$ independent ensembles and inhibition is perfectly co-tuned with excitation. In the case $m = 1$, inhibition is global and the block matrices become identical to the above case of global inhibition.

The eigenvalues of the matrix $\mathbf{R}$ are given as the roots of the characteristic polynomial defined by:

$$\det((\mathbf{J}_{E\leftarrow E} - \lambda\mathbb{1})(\mathbf{J}_{I\leftarrow I} - \lambda\mathbb{1}) - \mathbf{J}_{E\leftarrow I}\mathbf{J}_{I\leftarrow E}) = 0 \quad (80)$$

which yields the following expression:

$$\left[\lambda^2 - (a - e - ka - Nd + Nmd - f)\lambda - (a - e - ka)(Nd - Nmd - f) \right.$$
$$\left. + N^2 bc(1 - m)^2\right]^{N-1}\left[(a - e + (N-1)ka - \lambda)(-Nd - f - \lambda) + N^2 bc\right] = 0 \quad (81)$$

We therefore had four distinct eigenvalues:

$$\lambda'_{1/2} = \frac{1}{2}\left[(a - e - ka - Nd + Nmd - f) \pm \sqrt{(a - e - ka + Nd - Nmd + f)^2 - 4N^2 bc(1 - m)^2}\right] \quad (82)$$

$$\lambda'_{3/4} = \frac{1}{2}\left[(a - e - f - Nd + (N-1)ka) \pm \sqrt{(a - e - f - Nd + (N-1)ka)^2 - 4((-af + ef + kaf) - N(a - e)d - Nkaf - N(N-1)kad + N^2 bc)}\right] \quad (83)$$

The eigenvalues $\lambda'_1$ and $\lambda'_2$ have an algebraic and geometric multiplicity of $(N–1)$, whereas the eigenvalues $\lambda'_3$ and $\lambda'_4$ have an algebraic and geometric multiplicity of 1. We noted that $\lambda_3 = \lambda'_3$, $\lambda_4 = \lambda'_4$.

To compare under which conditions networks with different structures are uni-stable, we examined the different eigenvalues derived above. As $\lambda_2 < 0$, and $\lambda'_1 > \lambda'_2$, we only had to compare $\lambda'_1$ to $\lambda_1$. For networks with co-tuned inhibition, we have $m < 1$,

$$\lambda'_1 = \frac{1}{2}\left[(a - e - ka - Nd + Nmd - f) + \sqrt{(a - e - ka + Nd - Nmd + f)^2 - 4N^2 bc(1 - m)^2}\right]$$
$$< \frac{1}{2}\left[(a - e - ka - Nd + Nmd - f) + \sqrt{(a - e - ka + Nd - Nmd + f)^2}\right] = a - e - ka = \lambda_1 \quad (84)$$

The inequality, $\lambda'_1 < \lambda_1$, indicates that networks with co-tuned inhibition have a broad parameter regime in which they are uni-stable than networks with global inhibition. Note that in the absence of a saturating nonlinearity of the input-output function and in the absence of any additional stabilization mechanisms, systems with positive eigenvalues of the Jacobian are unstable. In this case, networks with co-tuned inhibition have a broad parameter regime of being stable than networks with global inhibition.

To visualize the conditions in a two-dimensional plane, we reduced the conditions into a function of $a$ and $d$. For **Figure 3C**, $k = 0.1$, $m = 0.5$ and $bc = 0.9ad$.

## Distance to the decision boundary

To calculate the distance to the decision boundary in **Figures 4 and 5**, **Figure 4—figure supplement 2** and **Figure 5—figure supplement 2**, we first projected the excitatory activity in Phase two onto a two-dimensional Cartesian coordinate system in which the horizontal axis is the activity of the first excitatory ensemble $r_{E1}$ and the vertical axis is the activity of the second excitatory ensemble $r_{E2}$. We denote the location of the projected data point in the Cartesian coordinate system by $(x, y)$, where $x$ and $y$ equal $r_{E1}$ and $r_{E2}$, respectively. The distance $L$ between the projected data and the decision boundary which corresponds to the diagonal line in the coordinate system can be expressed as follows:

$$L = \sqrt{x^2 + y^2}\sin(|45^o - \arcsin(\frac{x}{\sqrt{x^2+y^2}})|) \tag{85}$$

Note that the inverse trigonometric function arcsin gives the value of the angle in degrees.

## Inhibitory feedback pathways for suppressing unwanted neural activation

To identify the important neural pathways for the suppression of unwanted neural activation, we analyzed how the activity of the second excitatory ensemble $r_{E2}$ changes with the input to the first excitatory ensemble $g_{E1}$. To that end, we considered a general weight matrix for networks with two interacting ensembles

$$\mathbf{J} = \begin{bmatrix} J_{E1E1} & J_{E1E2} & -J_{E1I1} & -J_{E1I2} \\ J_{E2E1} & J_{E2E2} & -J_{E2I1} & -J_{E2I2} \\ J_{I1E1} & J_{I1E2} & -J_{I1I1} & -J_{I1I2} \\ J_{I2E1} & J_{I2E2} & -J_{I2I1} & -J_{I2I2} \end{bmatrix} \tag{86}$$

We can write the change in firing rate of the excitatory population in the second ensemble $\delta r_{E2}$ as a function of the change in the input to the other $\delta g_{E1}$:

$$\delta r_{E2} = \frac{1}{\det(\mathbb{1} - \mathbf{FJ})}\Big[ (-f'_{E2}J_{E2E1})f'_{I1}J_{I1I2}f'_{I2}J_{I2I1} + f'_{E2}J_{E2I1}(-f'_{I1}J_{I1E1})(1 + f'_{I2}J_{I2I2}) + f'_{E2}J_{E2I2}(1 + f'_{I1}J_{I1I1})(-f'_{I2}J_{I2E1}) \\ - (-f'_{E2}J_{E2E1})(1 + f'_{I1}J_{I1I1})(1 + f'_{I2}J_{I2I2}) - f'_{E2}J_{E2I1}f'_{I1}J_{I1I2}(-f'_{I2}J_{I2E1}) - f'_{E2}J_{E2I2}(-f'_{I1}J_{I1E1})f'_{I2}J_{I2I1} \Big] f'_{E1}\delta g_{E1} \tag{87}$$

where $\mathbb{1}$ is the identity matrix. And $\mathbf{F}$ is given by

$$\mathbf{F} = \begin{bmatrix} f'_{E1} & 0 & 0 & 0 \\ 0 & f'_{E2} & 0 & 0 \\ 0 & 0 & f'_{I1} & 0 \\ 0 & 0 & 0 & f'_{I2} \end{bmatrix} \tag{88}$$

where $f'_{E1}$, $f'_{E2}$, $f'_{I1}$ and $f'_{I2}$ are the derivatives of the input-output functions evaluated at the fixed point.

Assuming that $J_{E1E1} = J_{E2E2} = J_{EE}$, $J_{I1E1} = J_{I2E2} = J_{IE}$, $J_{E1I1} = J_{E2I2} = J_{EI}$, $J_{I1I1} = J_{I2I2} = J_{II}$, $J_{E1E2} = J_{E2E1} = J'_{EE}$, $J_{I1E2} = J_{I2E1} = J'_{IE}$, $J_{E1I2} = J_{E2I1} = J'_{EI}$ and $J_{I1I2} = J_{I2I1} = J'_{II}$, we find

$$\delta r_{E2} = \frac{1}{\det(\mathbb{1} - \mathbf{FJ})}\Big[ (-f'_{E2}J'_{EE})f'_{I1}J'_{II}f'_{I2}J'_{II} + f'_{E2}J'_{EI}(-f'_{I1}J_{IE})(1 + f'_{I2}J_{II}) + f'_{E2}J_{EI}(1 + f'_{I1}J_{II})(-f'_{I2}J'_{IE}) \\ - (-f'_{E2}J'_{EE})(1 + f'_{I1}J_{II})(1 + f'_{I2}J_{II}) - f'_{E2}J_{EI}f'_{I1}J'_{II}(-f'_{I2}J'_{IE}) - f'_{E2}J_{EI}(-f'_{I1}J_{IE})f'_{I2}J'_{II} \Big] f'_{E1}\delta g_{E1} \tag{89}$$

By further assuming that the weight strengths across ensembles are weak and ignoring the corresponding higher-order terms, we get

$$\delta r_{E2} \approx \frac{1}{\det(\mathbb{1} - \mathbf{FJ})} \left[ f'_{E2} J'_{EI} (-f'_{I1} J_{IE})(1 + f'_{I2} J_{II}) + f'_{E2} J'_{EI} (1 + f'_{I1} J_{II})(-f'_{I2} J'_{IE}) \right.$$

$$\left. - (-f'_{E2} J'_{EE})(1 + f'_{I1} J_{II})(1 + f'_{I2} J_{II}) - f'_{E2} J'_{EI} (-f'_{I1} J_{IE}) f'_{I2} J'_{II} \right] f'_{E1} \delta g_{E1}$$

$$= \frac{1}{\det(\mathbb{1} - \mathbf{FJ})} \left[ \left( \frac{J'_{II}}{J'_{EI}} f'_{I2} - (\frac{1}{J_{EI}} + f'_{I2} \frac{J_{II}}{J_{EI}}) \right) J'_{EI} J_{EI} J_{IE} f'_{E2} f'_{I1} \right.$$

$$\left. + \left( \frac{J'_{EE}}{J'_{IE}} (1 + J_{II} f'_{I2}) - J_{EI} f'_{I2} \right) J'_{IE} f'_{E2} (1 + f'_{I1} J_{II}) \right] f'_{E1} \delta g_{E1}$$
(90)

Note that $\frac{J'_{EE}}{J'_{IE}}$ and $\frac{J'_{II}}{J'_{EI}}$ are terms regulating the respective excitatory and inhibitory input from one ensemble to the excitatory and inhibitory population in another ensemble. The term $\det(\mathbb{1} - \mathbf{FJ})$ is positive to ensure the stability of the system.

To suppress the activity of the excitatory population in the second ensemble $r_{E2}$, in other words, to ensure that $\delta r_{E2} < 0$, $J'_{IE}$ or/and $J'_{EI}$ have to be large. Therefore, we identified $J'_{IE}$ and $J'_{EI}$ as important synaptic connections which lead to suppression of the unwanted neural activation, suggesting that inhibition can be provided via $J'_{IE}$ through the $E1$-$I2$-$E2$ pathway or via $J'_{EI}$ through the $E1$-$I1$-$E2$ pathway.

For *Figures 4 and 5*, the rate-based model consists of two ensembles, each of which is composed of 100 excitatory and 25 inhibitory neurons with all-to-all connectivity.

## Spiking neural network model

The spiking neural network model was composed of $N_E$ excitatory and $N_I$ inhibitory leaky integrate-and-fire neurons. Neurons were randomly connected with probability of 20%. The dynamics of membrane potential of neuron $i$, $U_i$, as defined by *Zenke et al., 2015*:

$$\tau^m \frac{dU_i}{dt} = (U^{\text{rest}} - U_i) + g_i^{\text{ext}}(t)(U^{\text{exc}} - U_i) + g_i^{\text{inh}}(t)(U^{\text{inh}} - U_i)$$
(91)

Here, $\tau^m$ is the membrane time constant and $U^{\text{rest}}$ is the resting potential. Spikes are triggered when the membrane potential reaches the spiking threshold $U^{\text{thr}}$. After a spike is emitted, the membrane potential is reset to $U^{\text{rest}}$ and the neuron enters a refractory period of $\tau^{\text{ref}}$. Inhibitory neurons obeyed the same integrate-and-fire formalism but with a shorter membrane time constant.

Excitatory synapses contain a fast AMPA component and a slow NMDA component. The dynamics of the excitatory conductance are described by:

$$\tau^{\text{ampa}} \frac{dg_i^{\text{ampa}}}{dt} = -g_i^{\text{ampa}} + \sum_{j \in \text{exc}} J_{ij} S_j(t)$$
(92)

$$\tau^{\text{nmda}} \frac{dg_i^{\text{nmda}}}{dt} = -g_i^{\text{nmda}} + g_i^{\text{ampa}}$$
(93)

$$g_i^{\text{exc}}(t) = \xi g_i^{\text{ampa}}(t) + (1 - \xi) g_i^{\text{nmda}}(t)$$
(94)

Here, $J_{ij}$ denotes the synaptic strength from neuron $j$ to neuron $i$. If the connection does not exist, $J_{ij}$ was set to 0. $S_j(t)$ is the spike train of neuron $j$, which is defined as $S_j(t) = \sum_k \delta(t - t_j^k)$, where $\delta$ is the Dirac delta function and $t_j^k$ the spikes times $k$ of neuron $j$. $\xi$ is a weighting parameter. The dynamics of inhibitory conductances are governed by:

$$\tau^{\text{gaba}} \frac{dg_i^{\text{inh}}}{dt} = -g_i^{\text{inh}} + \sum_{j \in \text{inh}} J_{ij} S_j(t)$$
(95)

In the spiking neural network models, SFA of excitatory neurons is modeled as follows,

$$\tau^m \frac{dU_i}{dt} = (U^{\text{rest}} - U_i) + g_i^{\text{ext}}(t)(U^{\text{exc}} - U_i) + (g_i^{\text{inh}}(t) + a_i(t))(U^{\text{inh}} - U_i)$$
(96)

$$\frac{da_i}{dt} = -\frac{a_i}{\tau_a} + b S_i(t)$$
(97)

where $i$ is the index of excitatory neurons.

The dynamics of E-to-E STD are given by

$$\frac{dx_{ij}}{dt} = \frac{1 - x_{ij}}{\tau_x} - U_d x_{ij} S_j(t) \tag{98}$$

$$\tau^{\text{ampa}} \frac{dg_i^{\text{ampa}}}{dt} = -g_i^{\text{ampa}} + \sum_{j \in \text{exc}} x_{ij} J_{ij} S_j(t) \tag{99}$$

where $i$ represents the index of excitatory neurons.

The dynamics of E-to-I STF are governed by

$$\frac{du_{ij}}{dt} = \frac{1 - u_{ij}}{\tau_u} + U_f(U_{max} - u_{ij}) S_j(t) \tag{100}$$

$$\tau^{\text{ampa}} \frac{dg_i^{\text{ampa}}}{dt} = -g_i^{\text{ampa}} + \sum_{j \in \text{exc}} u_{ij} J_{ij} S_j(t) \tag{101}$$

where $i$ denotes the index of inhibitory neurons.

For *Figure 6*, each excitatory and inhibitory neuron received external excitatory input from 300 neurons firing with Poisson statistics at an average firing rate of 0.1 Hz at baseline. During stimulation, the excitatory neurons corresponding to the activated ensemble received external excitatory input from 300 neurons firing with Poisson statistics at an average firing rate of 0.5 Hz. The ensemble activity is computed from the instantaneous firing rates of the respective ensembles with 10ms bin size. The difference in ensemble activity for the peak amplitude is calculated by subtracting the average maximal ensemble activity of the unstimulated ensembles from the maximal ensemble activity of the activated ensemble. Similarly, the difference in ensemble activity for the fixed point is calculated by subtracting the average ensemble activity of the unstimulated ensembles at the fixed point from the ensemble activity of the activated ensemble at the fixed point. Fixed point activity is computed by averaging the activity of the middle 1 second within the 2-second stimulation period.

For *Figure 2—figure supplement 10*, each excitatory and inhibitory neuron received external excitatory input from 300 neurons firing with Poisson statistics at an average firing rate of 0.1 Hz at

**Table 1.** Parameters for *Figure 1C–E*.

| Symbol | Value | Unit | Description |
|---|---|---|---|
| $J_{EE}$ | 1.8 | - | E-to-E connection strength |
| $J_{IE}$ | 1.0 | - | E-to-I connection strength |
| $J_{EI}$ | 1.0 | - | I-to-E connection strength |
| $J_{II}$ | 0.6 | - | I-to-I connection strength |
| $\alpha_E$ | 2 | - | Power of excitatory input-output function |
| $\alpha_I$ | 2 | - | Power of inhibitory input-output function |
| $\tau_E$ | 20 | ms | Time constant of excitatory firing dynamics |
| $\tau_I$ | 10 | ms | Time constant of inhibitory firing dynamics |
| $g_E^{bs}$ | 1.55 | - | Input to the $E$ population at baseline |
| $g_E^{stim}$ | 3.0 | - | Input to the $E$ population during stimulation |
| $g_I$ | 2.0 | - | Input to the $I$ population |
| Parameters for *Figure 1F* | | | |
| $J_{IE}$ | 0.45 | - | E-to-I connection strength |
| $J_{EI}$ | 1.0 | - | I-to-E connection strength |
| $J_{II}$ | 1.5 | - | I-to-I connection strength |

**Table 2.** Parameters for *Figure 2*.

| Symbol | Value | Unit | Description |
|---|---|---|---|
| $\tau_a$ | 200 | ms | Time constant of SFA |
| $b$ | 1.0 | - | Strength of SFA |
| $\tau_x$ | 200 | ms | Time constant of STD |
| $U_d$ | 1.0 | - | Depression rate |
| $\tau_u$ | 200 | ms | Time constant of STF |
| $U_f$ | 1.0 | - | Facilitation rate |
| $U_{max}$ | 6.0 | - | Maximal facilitation value |

Note that these values are also applied elsewhere unless mentioned otherwise.

the baseline. During stimulation, each excitatory neuron received external excitatory input from 300 neurons firing with Poisson statistics at an average firing rate of 0.3 Hz.

For *Figure 6—figure supplement 1*, the firing rates of 300 neurons are varying from 4/15 Hz to 7/15 Hz.

## Simulations

Simulations were performed in Python and Mathematica. All differential equations were implemented by Euler integration with a time step of 0.1 ms. All simulation parameters are listed in *Tables 1–5* and *Appendix 5—Tables 1–10*. The simulation source code to reproduce the figures is publicly available at https://github.com/fmi-basel/gzenke-nonlinear-transient-amplification (*Wu, 2021* copy archived at swh:1:rev:6ff6ff10b9f4994a0f948a987a66cc82f98451e1).

**Table 3.** Parameters for *Figure 3* bi/multi-stable example.

| Symbol | Value | Unit | Description |
|---|---|---|---|
| $J_{EE}$ | 1.4 | - | Within-ensemble E-to-E connection strength |
| $J_{IE}$ | 0.6 | - | Within-ensemble E-to-I connection strength |
| $J_{EI}$ | 1.0 | - | Within-ensemble I-to-E connection strength |
| $J_{II}$ | 0.6 | - | Within-ensemble I-to-I connection strength |
| $J'_{EE}$ | 0.14 | - | Inter-ensemble E-to-E connection strength |
| $J'_{IE}$ | 0.6 | - | Inter-ensemble E-to-I connection strength |
| $J'_{EI}$ | 1.0 | - | Inter-ensemble I-to-E connection strength |
| $J'_{II}$ | 0.6 | - | Inter-ensemble I-to-I connection strength |
| $g_{E1}^{bs}$ | 2.2 | - | Input to the $E1$ population at baseline |
| $g_{E1}^{stim}$ | 3.0 | - | Input to the $E1$ population during stimulation |
| $g_{E2}$ | 2.2 | - | Input to the $E2$ population |
| $g_I$ | 2.0 | - | Input to the $I$ population |
| Parameters for *Figure 3* uni-stable example | | | |
| $J_{EE}$ | 1.3 | - | Within-ensemble E-to-E connection strength |
| $J'_{EE}$ | 0.13 | - | Inter-ensemble E-to-E connection strength |

**Table 4.** Parameters for *Figures 4 and 5*.

| Symbol | Value | Unit | Description |
|---|---|---|---|
| $N_E$ | 200 | - | Number of excitatory neurons |
| $N_I$ | 50 | - | Number of inhibitory neurons |
| $N$ | 2 | - | Number of ensembles |
| $J_{EE}$ | $1.2/(N_E/2 - 1)$ | - | Within-ensemble E-to-E connection strength |
| $J_{IE}$ | $1.0/(N_E/2)$ | - | Within-ensemble E-to-I connection strength |
| $J_{EI}$ | $1.0/(N_I/2)$ | - | Within-ensemble I-to-E connection strength |
| $J_{II}$ | $1.0/(N_I/2 - 1)$ | - | Within-ensemble I-to-I connection strength |
| $J'_{EE}$ | $0.36/(N_E/2 - 1)$ | - | Inter-ensemble E-to-E connection strength |
| $J'_{IE}$ | $0.4/(N_E/2)$ | - | Inter-ensemble E-to-I connection strength |
| $J'_{EI}$ | $0.1/(N_I/2)$ | - | Inter-ensemble I-to-E connection strength |
| $J'_{II}$ | $0.1/(N_I/2)$ | - | Inter-ensemble I-to-I connection strength |
| $g_I$ | 2.0 | - | Input to the *I* population |
| Parameters for *Figure 4* | | | |
| $g_{E1}^{bs}$ | 1.35 | - | Input to the *E1* population |
| $g_{E1}^{stim}$ | 4.0 | - | Input to the *E1* population during stimulation |
| $g_{E2}$ | 1.35 | - | Input to the *E2* population |
| Parameters for *Figure 5* | | | |
| $g_{E1}^{bs}$ | 1.35 | - | Input to the *E1* population at baseline |
| $g_{E1}^{stim}$ | $1.35 + (4.0–1.35)(1\text{-}p)$ | - | Input to the *E1* population during stimulation |
| $g_{E2}^{bs}$ | 1.35 | - | Input to the *E2* population at baseline |
| $g_{E2}^{stim}$ | $1.35 + (4.0–1.35)p$ | - | Input to the *E2* population during stimulation |

Here, *p* is a parameter between 0 and 1 controlling the additional inputs to *E1* and *E2*.

**Table 5.** Parameters for *Figure 6*.

| Symbol | Value | Unit | Description |
|---|---|---|---|
| $N_E$ | 400 | - | Number of excitatory neurons |
| $N_I$ | 100 | - | Number of inhibitory neurons |
| $U^{\text{rest}}$ | –70 | mV | Resting membrane potential |
| $U^{\text{exc}}$ | 0 | mV | Excitatory reversal potential |
| $U^{\text{inh}}$ | –80 | mV | Inhibitory reversal potential |
| $\tau^{\text{ref}}$ | 3 | ms | Duration of refractory period |
| $\tau^m_{\text{exc}}$ | 20 | ms | Membrane time constant of excitatory neurons |
| $\tau^m_{\text{inh}}$ | 10 | ms | Membrane time constant of inhibitory neurons |
| $\tau^{\text{ampa}}$ | 5 | ms | Time constant of AMPA receptor |
| $\tau^{\text{gaba}}$ | 10 | ms | Time constant of GABA receptor |
| $\tau^{\text{nmda}}$ | 100 | ms | Time constant of NMDA receptor |
| $\xi$ | 0.5 | - | Receptor weighting factor |
| $J_{EE}$ | 0.19 | - | Within-ensemble E-to-E connection strength |
| $J_{IE}$ | 0.10 | - | Within-ensemble E-to-I connection strength |
| $J_{EI}$ | 0.10 | - | Within-ensemble I-to-E connection strength |
| $J_{II}$ | 0.06 | - | Within-ensemble I-to-I connection strength |
| $J'_{EE}$ | 0.019 | - | Inter-ensemble E-to-E connection strength |
| $J'_{IE}$ | 0.05 | - | Inter-ensemble E-to-I connection strength |
| $J'_{EI}$ | 0.04 | - | Inter-ensemble I-to-E connection strength |
| $J'_{II}$ | 0.006 | - | Inter-ensemble I-to-I connection strength |

# Acknowledgements

We thank Rainer W Friedrich, Claire Meissner-Bernard, William F Podlaski, and members of the Zenke Group for comments and discussions. This work was supported by the Novartis Research Foundation.

# Additional information

## Funding

| Funder | Grant reference number | Author |
|---|---|---|
| Novartis Foundation | | Yue Kris Wu<br>Friedemann Zenke |

The funders had no role in study design, data collection and interpretation, or the decision to submit the work for publication.

## Author contributions

Yue Kris Wu, Formal analysis, Investigation, Methodology, Software, Visualization, Writing – original draft, Writing – review and editing; Friedemann Zenke, Conceptualization, Funding acquisition, Methodology, Supervision, Writing – original draft, Writing – review and editing

**Author ORCIDs**
Yue Kris Wu http://orcid.org/0000-0002-9804-2537
Friedemann Zenke http://orcid.org/0000-0003-1883-644X

**Decision letter and Author response**
Decision letter https://doi.org/10.7554/eLife.71263.sa1
Author response https://doi.org/10.7554/eLife.71263.sa2

## Additional files

**Supplementary files**
• Transparent reporting form

**Data availability**

This project is a theory project without data. All simulation code has been deposited on GitHub under https://github.com/fmi-basel/gzenke-nonlinear-transient-amplification, (copy archived at swh:1:rev:6ff6ff10b9f4994a0f948a987a66cc82f98451e1).

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

## Appendix 1

## Stability conditions in networks with E-to-I STF

The dynamics of supralinear networks with E-to-I STF can be described as follows:

$$\tau_E \frac{dr_E}{dt} = -r_E + \left[ J_{EE}r_E - J_{EI}r_I + g_E \right]_+^{\alpha_E} \tag{102}$$

$$\tau_I \frac{dr_I}{dt} = -r_I + \left[ uJ_{IE}r_E - J_{II}r_I + g_I \right]_+^{\alpha_I} \tag{103}$$

$$\frac{du}{dt} = \frac{1-u}{\tau_u} + U_f(U_{max} - u)r_E \tag{104}$$

The Jacobian $\mathbf{M_{STF}}$ of the system with E-to-I STF is given by:

$$\mathbf{M_{STF}} = \begin{bmatrix} \tau_E^{-1}(J_{EE}\alpha_E r_E^{\frac{\alpha_E-1}{\alpha_E}} - 1) & -\tau_E^{-1}J_{EI}\alpha_E r_E^{\frac{\alpha_E-1}{\alpha_E}} & 0 \\ \tau_I^{-1}uJ_{IE}\alpha_I r_I^{\frac{\alpha_I-1}{\alpha_I}} & -\tau_I^{-1}(1 + J_{II}\alpha_I r_I^{\frac{\alpha_I-1}{\alpha_I}}) & \tau_I^{-1}J_{IE}r_E\alpha_I r_I^{\frac{\alpha_I-1}{\alpha_I}} \\ U_f(U_{max} - u) & 0 & -\tau_u^{-1} - U_f r_E \end{bmatrix} \tag{105}$$

The characteristic polynomial for the system with E-to-I STF can be written as follows:

$$\lambda^3 - \mathrm{tr}(\mathbf{M_{STF}})\lambda^2 + (A_{11} + A_{22} + A_{33})\lambda - \det(\mathbf{M_{STF}}) = 0 \tag{106}$$

where $\mathrm{tr}(\mathbf{M_{STF}})$ and $\det(\mathbf{M_{STF}})$ are the trace and the determinant of the Jacobian matrix $\mathbf{M_{SFA}}$, $A_{11}$, $A_{22}$, and $A_{33}$ are the matrix cofactors. More specifically,

$$\mathrm{tr}(\mathbf{M_{STF}}) = \tau_E^{-1}(J_{EE}\alpha_E r_E^{\frac{\alpha_E-1}{\alpha_E}} - 1) - \tau_I^{-1}(1 + J_{II}\alpha_I r_I^{\frac{\alpha_I-1}{\alpha_I}}) - \tau_u^{-1} - U_f r_E$$
$$\propto \tau_E^{-1}(J_{EE}\alpha_E r_E^{\frac{\alpha_E-1}{\alpha_E}} / r_I^{\frac{\alpha_I-1}{\alpha_I}} - r_I^{\frac{1-\alpha_I}{\alpha_I}}) - \tau_I^{-1}(r_I^{\frac{1-\alpha_I}{\alpha_I}} + J_{II}\alpha_I) - \tau_u^{-1}r_I^{\frac{1-\alpha_I}{\alpha_I}} - U_f r_E r_I^{\frac{1-\alpha_I}{\alpha_I}} \tag{107}$$

Assuming that $\alpha_E = \alpha_I = \alpha$, we then have

$$\mathrm{tr}(\mathbf{M_{STF}}) \propto \tau_E^{-1}\left[ J_{EE}\alpha\left(\frac{r_E}{r_I}\right)^{\frac{\alpha-1}{\alpha}} - r_I^{\frac{1-\alpha}{\alpha}} \right] - \tau_I^{-1}(r_I^{\frac{1-\alpha}{\alpha}} + J_{II}\alpha) - \tau_u^{-1}r_I^{\frac{1-\alpha}{\alpha}} - U_f r_E r_I^{\frac{1-\alpha}{\alpha}} \tag{108}$$

Substituting the firing rates with the current into excitatory population $z$, we then have

$$\mathrm{tr}(\mathbf{M_{STF}}) \propto \tau_E^{-1}\left[ J_{EE}\alpha\left( \frac{z}{\det(\mathbf{J_{STF}}) \cdot J_{EI}^{-1}[z]_+^\alpha + J_{EI}^{-1}J_{II}z - J_{EI}^{-1}J_{II}g_E + g_I} \right)^{\alpha-1} - r_I^{\frac{1-\alpha}{\alpha}} \right]$$
$$- \tau_I^{-1}(r_I^{\frac{1-\alpha}{\alpha}} + J_{II}\alpha) - \tau_u^{-1}r_I^{\frac{1-\alpha}{\alpha}} - U_f r_E r_I^{\frac{1-\alpha}{\alpha}} \tag{109}$$

$$\det(\mathbf{J_{STF}}) = \begin{vmatrix} J_{EE} & -J_{EI} \\ uJ_{IE} & -J_{II} \end{vmatrix} = -J_{EE}J_{II} + uJ_{IE}J_{EI} \tag{110}$$

In the large $r_E$ limit, $z$ is large, $\lim_{r_E \to \infty} u = \lim_{r_E \to \infty} \frac{1 + U_f U_{max} r_E \tau_u}{1 + U_f r_E \tau_u} \approx U_{max}$. Therefore, we can guarantee that $\det(\mathbf{J_{STF}})$ becomes positive for sufficiently large $U_{max}$. Since the denominator $\det(\mathbf{J_{STF}}) \cdot J_{EI}^{-1}[z]_+^\alpha + J_{EI}^{-1}J_{II}z - J_{EI}^{-1}J_{II}g_E + g_I$ grows faster than the numerator for $z \gg 1$, $\mathrm{tr}(\mathbf{M_{STF}})$ becomes negative for large $r_E$.

$$A_{11} + A_{22} + A_{33} = \tau_I^{-1}(1 + J_{II}\alpha_I r_I^{\frac{\alpha_I-1}{\alpha_I}})(\tau_u^{-1} + U_f r_E)$$
$$+ \tau_E^{-1}(J_{EE}\alpha_E r_E^{\frac{\alpha_E-1}{\alpha_E}} - 1)(-\tau_u^{-1} - U_f r_E)$$
$$- \tau_E^{-1}(J_{EE}\alpha_E r_E^{\frac{\alpha_E-1}{\alpha_E}} - 1)\tau_I^{-1}(1 + J_{II}\alpha_I r_I^{\frac{\alpha_I-1}{\alpha_I}}) + \tau_E^{-1}J_{EI}\alpha_E r_E^{\frac{\alpha_E-1}{\alpha_E}}\tau_I^{-1}uJ_{IE}\alpha_I r_I^{\frac{\alpha_I-1}{\alpha_I}} \tag{111}$$

Similarly, in the large $r_E$ limit, $A_{11} + A_{22} + A_{33}$ is positive.

$$\det(\mathbf{M_{STF}}) = \tau_E^{-1}(J_{EE}\alpha_E r_E^{\frac{\alpha_E-1}{\alpha_E}} - 1)\tau_I^{-1}(1 + J_{II}\alpha_I r_I^{\frac{\alpha_I-1}{\alpha_I}})(\tau_u^{-1} + U_f r_E)$$
$$+ \tau_E^{-1}J_{EI}\alpha_E r_E^{\frac{\alpha_E-1}{\alpha_E}}(\tau_I^{-1}uJ_{IE}\alpha_I r_I^{\frac{\alpha_I-1}{\alpha_I}}(-\tau_u^{-1} - U_f r_E) - \tau_I^{-1}J_{IE}r_E\alpha_I r_I^{\frac{\alpha_I-1}{\alpha_I}}U_f(U_{max}-u))$$

(112)

Similarly, in the large $r_E$ limit, $\det(\mathbf{M_{STF}})$ is negative.

Therefore, similar to E-to-E STD, networks dynamics can also be stabilized by E-to-I STF.

## Appendix 2

### Conditions for ISN in networks with E-to-I STF

Here, we identify the condition of being ISN in supralinear networks with E-to-I STF. If inhibition is frozen, in other words, if feedback inhibition is absent, the Jacobian of the system becomes as follows:

$$\mathbf{M_2} = \begin{bmatrix} \tau_E^{-1}(J_{EE}\alpha_E r_E^{\frac{\alpha_E-1}{\alpha_E}} - 1) & 0 \\ U_f(U_{max} - u) & -\tau_u^{-1} - U_f r_E \end{bmatrix} \tag{113}$$

For the system with frozen inhibition, the dynamics are stable if

$$\text{tr}(\mathbf{M_2}) = \tau_E^{-1}(J_{EE}\alpha_E r_E^{\frac{\alpha_E-1}{\alpha_E}} - 1) - \tau_u^{-1} - U_f r_E < 0 \tag{114}$$

and

$$\det(\mathbf{M_2}) = \tau_E^{-1}(J_{EE}\alpha_E r_E^{\frac{\alpha_E-1}{\alpha_E}} - 1)(-\tau_u^{-1} - U_f r_E) > 0 \tag{115}$$

Therefore, if the network is an ISN at the fixed point, the following condition has to be satisfied:

$$\tau_E^{-1}(J_{EE}\alpha_E r_E^{\frac{\alpha_E-1}{\alpha_E}} - 1) > 0 \tag{116}$$

Note that this condition is independent of the facilitation variable $u$ of E-to-I STF. We further define the ISN index for the system with E-to-I STF as follows:

$$\text{ISN index} = \tau_E^{-1}(J_{EE}\alpha_E r_E^{\frac{\alpha_E-1}{\alpha_E}} - 1) \tag{117}$$

## Appendix 3

### Conditions for paradoxical response in networks with E-to-I STF

Next, we identify the condition of having the paradoxical effect in supralinear networks with E-to-I STF. The excitatory nullcline is defined by

$$\tau_E \frac{dr_E}{dt} = -r_E + \left[ J_{EE}r_E - J_{EI}r_I + g_E \right]_+^{\alpha_E} = 0 \tag{118}$$

For $r_{E,I} > 0$, we have

$$r_I = \frac{J_{EE}r_E - r_E^{\frac{1}{\alpha_E}} + g_E}{J_{EI}} \tag{119}$$

The slope of the excitatory nullcline in the $r_E/r_I$ plane where $x$ axis is $r_E$ and $y$ axis is $r_I$ can be written as follows

$$k_{STF}^E = \frac{J_{EE} - \frac{1}{\alpha_E} r_E^{\frac{1}{\alpha_E} - 1}}{J_{EI}} \tag{120}$$

Note that the slope of the excitatory nullcline is nonlinear. To have paradoxical effect, the slope of the excitatory nullcline at the fixed point of the system has to be positive. We therefore have

$$J_{EE}\alpha_E r_E^{\frac{\alpha_E - 1}{\alpha_E}} - 1 > 0 \tag{121}$$

We exploit a separation of timescales between fast neural activity and slow short-term plasticity variable, we therefore set the facilitation variable to the value at its fixed point corresponding to the dynamical value of $r_E$. Then we can write the inhibitory nullcline as follows

$$\tau_I \frac{dr_I}{dt} = -r_I + \left[ \frac{1 + U_f U_{max} r_E \tau_u}{1 + U_f r_E \tau_u} J_{IE}r_E - J_{II}r_I + g_I \right]_+^{\alpha_I} = 0 \tag{122}$$

In the region of rates $r_{E,I} > 0$, we have

$$r_I = \frac{\frac{1 + U_f U_{max} r_E \tau_u}{1 + U_f r_E \tau_u} J_{IE}r_E - r_I^{\frac{1}{\alpha_I}} + g_I}{J_{II}} \tag{123}$$

The slope of the inhibitory nullcline can be written as follows

$$k_{STF}^I = \frac{\frac{1 + U_f U_{max} r_E \tau_u}{1 + U_f r_E \tau_u} J_{IE} + \frac{U_f U_{max} \tau_u - U_f \tau_u}{(1 + U_f r_E \tau_u)^2} J_{IE}r_E}{J_{II} + \frac{1}{\alpha_I} r_I^{\frac{1 - \alpha_I}{\alpha_I}}} \tag{124}$$

In addition to the positive slope of the excitatory nullcline, the slope of the inhibitory nullcline at the fixed point of the system has to be larger than the slope of the excitatory nullcline. We therefore have

$$-(J_{EE}\alpha_E r_E^{\frac{\alpha_E - 1}{\alpha_E}} - 1)(1 + J_{II}\alpha_I r_I^{\frac{\alpha_I - 1}{\alpha_I}}) + J_{IE}\alpha_E r_E^{\frac{\alpha_E - 1}{\alpha_E}} \frac{1 + U_f U_{max} r_E \tau_u}{1 + U_f r_E \tau_u} J_{EI}\alpha_I r_I^{\frac{\alpha_I - 1}{\alpha_I}}$$

$$+ J_{IE}\alpha_E r_E^{\frac{\alpha_E - 1}{\alpha_E}} \frac{U_f U_{max} \tau_u - U_f \tau_u}{(1 + U_f r_E \tau_u)^2} J_{EI}\alpha_I r_I^{\frac{\alpha_I - 1}{\alpha_I}} r_E > 0 \tag{125}$$

The above condition is the same as the stability condition of the determinant of the Jacobian of the system with E-to-I STF (*Eq. (112)*). Therefore, the condition is always satisfied when the system with E-to-I STF is stable.

Note that the condition of being ISN shown in *Eq. (116)* is identical to the condition of having paradoxical effect shown in *Eq. (121)*. Therefore, in networks with E-to-I STF alone, paradoxical

effect implies ISN and ISN implies paradoxical effect. We thus use paradoxical effect as a proxy for inhibitory stabilization.

## Appendix 4

### Change in steady-state activity of unstimulated co-tuned neurons

To analyze the pattern completion in supralinear networks, we considered a network with one excitatory population and one inhibitory population. Neurons in the excitatory population are co-tuned to the same stimulus feature and are separated into two subsets denoting by $E_{11}$ and $E_{12}$. The dynamics of the system can be described as follows:

$$\tau_E \frac{dr_{E_{11}}}{dt} = -r_{E_{11}} + \left[ J_{E_{11}E_{11}} r_{E_{11}} + J_{E_{11}E_{12}} r_{E_{12}} - J_{E_{11}I} r_I + g_{E_{11}} \right]_+^{\alpha_E} \tag{126}$$

$$\tau_E \frac{dr_{E_{12}}}{dt} = -r_{E_{12}} + \left[ J_{E_{12}E_{11}} r_{E_{11}} + J_{E_{12}E_{12}} r_{E_{12}} - J_{E_{12}I} r_I + g_{E_{12}} \right]_+^{\alpha_E} \tag{127}$$

$$\tau_I \frac{dr_I}{dt} = -r_I + \left[ J_{IE_{11}} r_{E_{11}} + J_{IE_{12}} r_{E_{12}} - J_{II} r_I + g_I \right]_+^{\alpha_I} \tag{128}$$

The change in the firing rate of the Subset 2 in the excitatory population $\delta r_{E_{12}}$ can be written as a function of the change in the input to the Subset 1 $\delta g_{E_{11}}$:

$$\begin{aligned}
\delta r_{E_{12}} &= \frac{1}{\det(\mathbb{1} - \mathbf{FJ})} [-f'_{E_{12}} J_{E_{12}I} f'_I J_{IE_{11}} - (-f'_{E_{12}} J_{E_{12}E_{11}})(1 + f'_I J_{II})] f'_{E_{11}} \delta g_{E_{11}} \\
&= \frac{1}{\det(\mathbb{1} - \mathbf{FJ})} [J_{E_{12}E_{11}} + J_{E_{12}E_{11}} J_{II} f'_I - J_{E_{12}I} J_{IE_{11}} f'_I] f'_{E_{11}} f'_{E_{12}} \delta g_{E_{11}}
\end{aligned} \tag{129}$$

where $\mathbb{1}$ is the identity matrix. And $\mathbf{F}$ is given by

$$\mathbf{F} = \begin{bmatrix} f'_{E_{11}} & 0 & 0 \\ 0 & f'_{E_{12}} & 0 \\ 0 & 0 & f'_I \end{bmatrix} \tag{130}$$

where $f'_{E_{11}}$, $f'_{E_{12}}$, and $f'_I$ are the derivatives of the input-output functions evaluated at the fixed point. The term $\det(\mathbb{1} - \mathbf{FJ})$ is positive to ensure the stability of the system.

Clearly, if the term $J_{E_{12}E_{11}} + J_{E_{12}E_{11}} J_{II} f'_I - J_{E_{12}I} J_{IE_{11}} f'_I$ is positive (negative), increasing the input to the Subset 1 leads to an increase (a decrease) in the activity of neurons in the Subset 2. As the input to the Subset 1 increases, the firing rate of the inhibitory population $r_I$ and also $f'_I$ will increase. In the presence of E-to-E STD or E-to-I STF, $J_{E_{12}E_{11}}$ or $J_{IE_{11}}$ will decrease or increase with the input to the Subset 1. As a result, the sign of $J_{E_{12}E_{11}} + J_{E_{12}E_{11}} J_{II} f'_I - J_{E_{12}I} J_{IE_{11}} f'_I$ can switch from positive to negative as the input to the Subset 1 increases, indicating that the effect on the activity of the co-tuned unstimulated neurons in the same ensemble can switch from potentiation to suppression. Note that this behavior is different from linear networks in which the change is independent of the input or firing rates.

# Appendix 5

**Appendix 5—table 1.** Parameters for *Figure 1—figure supplement 1*.

| Symbol | Value | Unit | Description |
|---|---|---|---|
| $J_{EE}$ | 0.5 | - | E-to-E connection strength |
| $J_{IE}$ | 0.45 | - | E-to-I connection strength |
| $J_{EI}$ | 1.0 | - | I-to-E connection strength |
| $J_{II}$ | 1.5 | - | I-to-I connection strength |
| $g_E^{bs}$ | 0.5 | - | Input to the E population at baseline |
| $g_I^{bs}$ | 1.5 | - | Input to the I population at baseline |

**Appendix 5—table 2.** Parameters for *Figure 2—figure supplement 2*.

| Symbol | Value | Unit | Description |
|---|---|---|---|
| $g_E^{stim}$ | 2.0 | - | Input to the E population during stimulation |

Note that values of the unlisted parameters are the same as *Tables 1–2*.

**Appendix 5—table 3.** Parameters for *Figure 2—figure supplement 3* SSN example.

| Symbol | Value | Unit | Description |
|---|---|---|---|
| $J_{EE}$ | 1.8 | - | E-to-E connection strength |
| $J_{IE}$ | 2.0 | - | E-to-I connection strength |
| $J_{EI}$ | 1.0 | - | I-to-E connection strength |
| $J_{II}$ | 1.0 | - | I-to-I connection strength |

**Appendix 5—table 4.** Parameters for *Figure 2—figure supplement 5*.

| Symbol | Value | Unit | Description |
|---|---|---|---|
| $g_E^{bs}$ | 1.8 | - | Input to the E population at baseline |

Note that values of the unlisted parameters are the same as *Tables 1–2*.

**Appendix 5—table 5.** Parameters for *Figure 2—figure supplement 10*.

| Symbol | Value | Unit | Description |
|---|---|---|---|
| $N_E$ | 400 | - | Number of excitatory neurons |
| $N_I$ | 100 | - | Number of inhibitory neurons |
| $J_{EE}$ | 0.05 | - | E-to-E connection strength |
| $J_{IE}$ | 0.02 | - | E-to-I connection strength |
| $J_{EI}$ | 0.05 | - | I-to-E connection strength |
| $J_{II}$ | 0.03 | - | I-to-I connection strength |

**Appendix 5—table 6.** Parameters for *Figure 2—figure supplement 11*.

| Symbol | Value | Unit | Description |
|---|---|---|---|
| $J_{EE}$ | 0.9 | - | E-to-E connection strength |
| $J_{IE}$ | 1.2 | - | E-to-I connection strength |

*Appendix 5—table 6 Continued on next page*

*Appendix 5—table 6 Continued*

| Symbol | Value | Unit | Description |
|---|---|---|---|
| $J_{EI}$ | 0.5 | - | I-to-E connection strength |
| $J_{II}$ | 0.5 | - | I-to-I connection strength |
| $\tau_E$ | 20 | ms | Time constant of excitatory firing dynamics |
| $\tau_I$ | 60 | ms | Time constant of inhibitory firing dynamics |
| $g_E^{bs}$ | 1.0 | - | Input to the *E* population at baseline |
| $g_E^{stim}$ | 2.0 | - | Input to the *E* population during stimulation |
| $g_I$ | 2.0 | - | Input to the *I* population |

Note that values of the unlisted parameters are the same as *Tables 1–2*.

**Appendix 5—table 7.** Parameters for *Figure 2—figure supplement 12*.

| Symbol | Value | Unit | Description |
|---|---|---|---|
| $J_{EE}$ | 1.0 | - | E-to-E connection strength |
| $J_{IE}$ | 1.2 | - | E-to-I connection strength |
| $J_{EI}$ | 0.5 | - | I-to-E connection strength |
| $J_{II}$ | 1.0 | - | I-to-I connection strength |
| $g_E^{bs}$ | 0.5 | - | Input to the *E* population at baseline |
| $g_I^{bs}$ | 1.0 | - | Input to the *I* population at baseline |

**Appendix 5—table 8.** Parameters for *Figure 3—figure supplement 1* global inhibition example.

| Symbol | Value | Unit | Description |
|---|---|---|---|
| $J_{EE}$ | 1.6 | - | Within-ensemble E-to-E connection strength |
| $J_{IE}$ | 1.0 | - | Within-ensemble E-to-I connection strength |
| $J_{EI}$ | 1.0 | - | Within-ensemble I-to-E connection strength |
| $J_{II}$ | 1.2 | - | Within-ensemble I-to-I connection strength |
| $J_{EE}'$ | 0.16 | - | Inter-ensemble E-to-E connection strength |
| $J_{IE}'$ | 1.0 | - | Inter-ensemble E-to-I connection strength |
| $J_{EI}'$ | 1.0 | - | Inter-ensemble I-to-E connection strength |
| $J_{II}'$ | 1.2 | - | Inter-ensemble I-to-I connection strength |
| $g_{E1}^{bs}$ | 1.5 | - | Input to the *E*1 population at baseline |
| $g_{E2}$ | 1.5 | - | Input to the *E*2 population |
| $g_{I1}$ | 2.5 | - | Input to the *I*1 population |
| $g_{I2}$ | 2.5 | - | Input to the *I*2 population |

Parameters for *Figure 3—figure supplement 1* co-tuned example

| Symbol | Value | Unit | Description |
|---|---|---|---|
| $J_{IE}$ | 1.0 * (4/3) | - | Within-ensemble E-to-I connection strength |
| $J_{EI}$ | 1.0 * (4/3) | - | Within-ensemble I-to-E connection strength |
| $J_{II}$ | 1.2 * (4/3) | - | Within-ensemble I-to-I connection strength |

*Appendix 5—table 8 Continued on next page*

*Appendix 5—table 8 Continued*

| Symbol | Value | Unit | Description |
|---|---|---|---|
| $J'_{IE}$ | 1.0 * (2/3) | - | Inter-ensemble E-to-I connection strength |
| $J'_{EI}$ | 1.0 * (2/3) | - | Inter-ensemble I-to-E connection strength |
| $J'_{II}$ | 1.2 * (2/3) | - | Inter-ensemble I-to-I connection strength |

**Appendix 5—table 9.** Parameters for *Figure 4—figure supplement 1*.

| Symbol | Value | Unit | Description |
|---|---|---|---|
| $J_{EE}$ | $1.5/(N_E/2 - 1)$ | - | Within-ensemble E-to-E connection strength |
| $J_{IE}$ | $1.0/(N_E/2)$ | - | Within-ensemble E-to-I connection strength |
| $J_{EI}$ | $1.0/(N_I/2)$ | - | Within-ensemble I-to-E connection strength |
| $J_{II}$ | $1.0/(N_I/2 - 1)$ | - | Within-ensemble I-to-I connection strength |
| $J'_{EE}$ | $0.1/(N_E/2 - 1)$ | - | Inter-ensemble E-to-E connection strength |
| $J'_{IE}$ | $0.3/(N_E/2)$ | - | Inter-ensemble E-to-I connection strength |
| $J'_{EI}$ | $0.3/(N_I/2)$ | - | Inter-ensemble I-to-E connection strength |
| $J'_{II}$ | $0.1/(N_I/2)$ | - | Inter-ensemble I-to-I connection strength |
| $g_{E1}^{bs}$ | 1.5 | - | Input to the *E1* population at baseline |
| $g_{E2}$ | 1.5 | - | Input to the *E2* population |
| $g_I$ | 2.0 | - | Input to the *I* population |

**Appendix 5—table 10.** Parameters for *Figure 6—figure supplement 1*.

| Symbol | Value | Unit | Description |
|---|---|---|---|
| $J_{EE}$ | 0.20 | - | Within-ensemble E-to-E connection strength |
| $J_{IE}$ | 0.09 | - | Within-ensemble E-to-I connection strength |
| $J_{EI}$ | 0.10 | - | Within-ensemble I-to-E connection strength |
| $J_{II}$ | 0.10 | - | Within-ensemble I-to-I connection strength |
| $J'_{EE}$ | 0.02 | - | Inter-ensemble E-to-E connection strength |
| $J'_{IE}$ | 0.054 | - | Inter-ensemble E-to-I connection strength |
| $J'_{EI}$ | 0.07 | - | Inter-ensemble I-to-E connection strength |
| $J'_{II}$ | 0.01 | - | Inter-ensemble I-to-I connection strength |

Note that values of the unlisted parameters are the same as *Table 5*.

