## [Editor Report]

Many brain circuits, particularly those found in mammalian sensory cortices, need to respond rapidly to stimuli while at the same time avoiding pathological, runaway excitation. Over several years, many theoretical studies have attempted to explain how cortical circuits achieve these goals through interactions between inhibitory and excitatory cells. This study adds to this literature by showing how synaptic short-term depression can stabilise strong positive feedback in a circuit under a variety of plausible scenarios, allowing strong, rapid and stimulus-specific responses.

---

## [Decision Letter]

**Decision letter after peer review:**

Thank you for submitting your article "Nonlinear transient amplification in recurrent neural networks with short-term plasticity" for consideration by *eLife*. Your article has been reviewed by 2 peer reviewers, and the evaluation has been overseen by a Reviewing Editor and Ronald Calabrese as the Senior Editor. The reviewers have opted to remain anonymous.

Essential revisions:

1) The findings in this work need to be better contextualised with existing literature. Some of the claims and results are presaged in other work and the reviewers agreed that the manuscript would benefit from better acknowledgement and discussion of prior work – see reviewer comments for details.

2) There are concerns that the behaviour of the model violates experimental observations (Reviewer 1, comment 1). The authors should address this concern, with additional results or sound arguments that reconcile these observations with the model.

3) Greater depth in the intuition and analysis of the stabilisation mechanism in this network is needed to justify all of the conclusions (Reviewer 1, comments 3-4; Reviewer 2, comments 4-6). This will also elevate the contribution of the manuscript and allow new findings to be better appreciated in the context of previous work.

4) The reviewers provide detailed comments below for improving the clarity and depth of the presentation. The authors should consider and respond to these suggestions.

*Reviewer #1 (Recommendations for the authors):*

– Line 51. Results from [Bolding and Franks, Science 2018] have been explained in [Stern et al., *eLife* 2018], without the need of adaptation. This work should be cited here, as it provides an explanation of transient response that is alternative to the mechanism proposed by the authors.

– Line 55. Isn't coupling strength, rather than inhibition, the key factor that linearizes network responses (e.g. see [van Vreeswijk and Sompolinsky, Neural computation 1998]).

– Above line 101. I do not understand why strong E-E connectivity is consistent only with detJ<0. Strong E-E connectivity typically also requires strong E-I and I-E connectivity [Tsodyks et al., J neuroscience 1997]. The assumption of detJ<0 seems important for the behavior of the characteristic function F(z). How are the results of the paper modified in the case in which det J>0?

– Line 124-126. I am confused about the conclusion of this section. Results discussed in Pages 5 and 6 are relative to the stability of the network as a function of input strength; why is this relevant for the nonlinear amplification of inputs? Nonlinear amplification of inputs is a general property of the SSN, which emerges also for parameters that lead to stable dynamics [Ahmadian et al., Neural computation 2013]. The authors should also explain why they select parameters that lead to unstable responses for large inputs. On a related note, the fact that the supralinear input-output function leads to activity dependent coupling and, for certain parameters, to unstable dynamics has been shown in [Ahmadian et al., Neural computation 2013]; the authors should acknowledge this fact.

– Line 188. There are many more papers related to ISN in cortex to be cited; see [Sadeh and Clopath, Nat Rev Neurosci 2021] for a summary.

– Line 273-274. It would be useful to include citations of experiments supporting this claim.

– Line 407-409. What is the relation of this work with network criticality?

– The stability of a system with excitatory-inhibitory neurons and STD depends on the eigenvalues of a three dimensional matrix, why does the condition described after Equation (32) do not include the component given by the dynamics of x?

*Reviewer #2 (Recommendations for the authors):*

First a note about the model of Loebel et al.,: note that while single neurons in their models have a ReLU nonlinearity, they receive different external baseline inputs, and therefore the population has an effective input-output nonlinearity that is convex and supralinear as in the current manuscript.

Specific/detailed comments:

1 – In the abstract, intro and discussion it is claimed that nonnormal connectivity is in clash with symmetric connectivity the E-subnetwork (and thus with the Hebbian learning/doctrine). But non-normal transient amplification happens in E-I networks in which the E-E part of connectivity matrix is indeed symmetric, while the full matrix is necessarily asymmetric and nonnormal (due to Dale's law). So the symmetry of the E-E connectivity (the part that is affected by Hebbian plasticity in most models) is in fact consistent with non-normal TA.

2 – Similarly in lines 56-65 of Intro (and 438-454 of discussion) it is claimed that balanced amplification (due to feedback inhibition and recurrent excitation) cannot generate strong TA, because "several ensembles need to be chained together into a hidden feedforward structure" which the authors say is "at odds with the often observed symmetric excitatory connectivity". But again symmetric E-E connectivity is consistent with nonnormality of the full (E-I) connectivity matrix, and moreover the chains are hidden between patterns, and not anatomical between neuronal ensembles. Also long hidden chains are not necessary, and when recurrent E and I connections are strong, short chains (e.g. in the SSN model) can also yield strong enough TA to be consistent with those observed in sensory cortex – though probably not as strong as in NTA. This parts should thus be more accurately written and if the claim is maintained a more thorough comparison (e.g. in a supplementary figure) with a recurrent network with neural nonlinearity (e.g. the supralinear powerlaw as in the current model) but without STP (but possibly with different connectivity so that the instability in Figure 1 does not happen) should be provided.

3 – Given that property C is a main selling point, they should characterize it better. The only result plot about this is currently Figure 1F, in which activity actually diverges post-stimulus. The nonlinear threshold (for stimulus strength) should be quantified and shown in some plot (e.g. to be added to Figure 2) for the case in which STP re-stabilizes activity. In particular the dependence of NTA (e.g. as measured by the ratio of transient peak to steady-state response) on parameters such as stimulus strength (c.f. Figure 4B of Loebel and Tsodyks 2002) or J_EE_, can be characterised/plotted.

4 – The effect of co-tuning of E-I (feature G) is characterised in abstract and the corresponding sub-section (its title on p. 11 and Figure 3 title) as broadening the parameter regime of NTA. But Figure 3 in fact shows that NTA happens even in the bistable case and with global inhibition. So a more accurate characterization his that E/I co-tuning broadens the parameter regime in which activity is unistable and non-persistent (which is suitable for sensory processing), without adversely affecting NTA. Also given that NTA is weaker -though still strong- with co-tuned inhibition, perhaps the strength of NTA can be quantatively compared between the two cases of co-tuned vs global inhibition.

5 – In the parts where intuitive explanation of re-stabilization (or lack thereof) due to SFA or STP are given on pp. 7-8, I would first of all provide Equations. 20-26 right there, or at least refer to them. More importantly when referring to F(z) they should mention that they are employing a fast-slow approximation using the separation of timescales between fast neural activity and adaptation variables (a in SFA, x in STD, etc) are therefore fixing the slow variables to their value at the previous/baseline fixed point. At least I believe this is what they are doing and showing in Figure 2D, but again I'm suggesting they should write this more clearly. In particular, it seems to me that while Equation 30 (with x assumed to be fixed at its value at the preceding fixed point) is consistent with the above interpretation (which I believe is the right thing to do), Equation 28 is not, as it is obtained by setting a equal to its value at its fixed point corresponding to the dynamical value of r_E_, and this is the right thing to do if "a" was a fast variable, not a slow one (as it presumably is). So, I believe, the correct F(z) for the SFA model should be rederived by assuming treating "a" in equation 20 as a constant independent of r_E_ (rather solving the steady state from 21 and plugging it in Equation 20).

6 – Switching to ISN with stimulus from a non-ISN baseline is presumably parameter dependent, and this dependence should be characterized better (at least give examples of networks in which this is not the case, in a supplementary figure, and correspondingly weaken the absolute statement in lines 192-194 so that it says this transition "can" happen, which would avoid implying that this is necessarily the case, assuming it's not). Also in the ISN index (Equation 13) shouldn't J_EE_ be multiplied by x (depression variable) at the fixed point?

7 – I found the part about pattern separation (p. 14) not clearly written and motivated. In fact I think the term "pattern separation" should be replaced with "stimulus selectivity". Pattern separation is the appropriate term when a networks output patterns are more separable than the patterns at the input-level. However, in the current case in which the external input is given only to E1 (only to one of the two E populations), the input is fully separable, so the network is potentially only capable of reducing separation. At any rate the pattern separation would be meaningful if they had compared separation at input-level (distance from decision boundary in gE1 – gE2 plane) with separation at output-level (distance from decision boundary in rE1 – rE2 plane). I think what is currently shown and charcterized is stimulus selectivity and the result is that transient onset responses are more selective than fixed point responses. So this section (and corresponding parts in intro/discussion) should be rewritten to convey this.

7a – On the other hand, this finding (stronger selectivity at onset, vs steady-state or sustained response) is presented as a desired outcome. This may be so, but it seems inconsistent with empirical findings on orientation selectivity in (mouse) visual cortex in which orientation selectivity is much stronger for sustained responses compared to onset transients. This should be mentioned (e.g. in lines 290, 326, and 418-420).

7b – The distance to decision boundary used to characterize "pattern separation" is not clearly described in lines 280-282. The description in the Methods section was also somewhat ambiguous, but giving a formula there would solve this issue. What I understood is that the measure is basically rE1 – rE2 where rE1 and rE2 are population mean activities of the E neurons in ensembles 1 and 2. If so, why not use the normalized measure (rE1 – rE2)/(rE1 + rE2)?

8 – The property B is depicted in various places (e.g. line 410) by saying NTA requires or relies on *symmetric* E-E connections. While NTA clearly relies on the destabilizing effect of recurrent EE connections, it is not clear that the symmetry (at least exact symmetry) of these connections is necessary. Indeed their LIF spiking net has random connectivity, so the EE part of connectivity at the level of single neurons in that network is actually not symmetric. These sentences could thus be more accurately written to clarify this point.

9 – In the current study, external stimulus was only given to excitatory neurons, with authors saying the study of feedforward input to inhibition is beyond the scope of this work, yet they also say (lines 429-432) that "we are confident that our main findings remain unaffected in the presence of substantial feedforward inhibition", and cite supplementary figure 1, which however was for the case without STP. I think it would be easy to provide a supplementary figure characterizing NTA in a network with STP in which gI is nonzero for stimulus (e.g. gI = gE).

---

## [Author Response]

Essential revisions:1) The findings in this work need to be better contextualised with existing literature. Some of the claims and results are presaged in other work and the reviewers agreed that the manuscript would benefit from better acknowledgement and discussion of prior work – see reviewer comments for details.

Thanks for the constructive feedback. The revised version now positions itself more clearly relative to the existing literature. Specifically, we highlight the close relation to (*Loebel and Tsodyks, 2002*; *Loebel et al., 2007*) and how we extend this work (see our reply to Point P 2.1). Moreover, we formulated the manuscript such that NTA is a possible alternative mechanism to balanced amplification (*Murphy and Miller, 2009*) and that the two are not mutually exclusive in biological networks (see our reply to Points P 1.3 and P 2.1). Finally, we elucidate the relation to the spiking network models without STP studied by *Stern et al.,* (*2018*) (see our reply to Point P 1.6).

2) There are concerns that the behaviour of the model violates experimental observations (Reviewer 1, comment 1). The authors should address this concern, with additional results or sound arguments that reconcile these observations with the model.

We were very grateful for these comments because they prompted us to perform additional analyses that led to new insights. Importantly, we were able to show that NTA can also occur in networks that are an ISN already at baseline. This effect depends on the network parameters. These additional results reconcile the model with recent experimental findings suggesting that the visual cortex is an ISN at baseline (*Sanzeni et al., 2020*). However, because the effect is parameter dependent, it leaves the possibility open that other sensory areas may be a non-ISN at baseline, e.g., olfactory areas (see our reply to Point P 1.1). Moreover, we also show that the firing rates of unstimulated ensemble neurons are not necessarily suppressed when stimulating the rest of the ensemble which renders our model compatible with observations on recent pattern completion experiments (*Carrillo-Reid et al., 2016*; *Marshel et al., 2019*). Again, this effect is parameter-dependent and our analysis suggests that it also changes with the stimulus intensity, which also makes a prediction that could be tested experimentally (see our reply to Point P 1.2).

3) Greater depth in the intuition and analysis of the stabilisation mechanism in this network is needed to justify all of the conclusions (Reviewer 1, comments 3-4; Reviewer 2, comments 4-6). This will also elevate the contribution of the manuscript and allow new findings to be better appreciated in the context of previous work.

Again thanks to this feedback, we deepened our understanding of the mechanisms through additional analyses to further justify our conclusions. This work led to the inclusion of multiple additional figure supplements and Methods sections. We also submit an Appendix with this revised manuscript. We addressed the reviewer concerns, specifically to Points P 1.4, P 1.5, P 2.5, P 2.6, and P 2.7, below.

4) The reviewers provide detailed comments below for improving the clarity and depth of the presentation. The authors should consider and respond to these suggestions.

Many thanks for the thoughtful feedback. Please find our responses to these points below.

Reviewer #1 (Recommendations for the authors):– Line 51. Results from [Bolding and Franks, Science 2018] have been explained in [Stern et al., eLife 2018], without the need of adaptation. This work should be cited here, as it provides an explanation of transient response that is alternative to the mechanism proposed by the authors.

Thanks for the pointer. We now added *Stern et al.,* (*2018*) in the Discussion. The main difference in their work is that the networks they study have feedback inhibition that is strong enough to stabilize dynamics already in the absence of STP. We now cite it as follows:

“In addition, in spiking neural networks, strong input can induce synchronous firing at the population level which is subsequently stabilized by strong feedback inhibition without the requirement for STP mechanisms (*Stern et al., 2018*)”

– Line 55. Isn’t coupling strength, rather than inhibition, the key factor that linearizes network responses (e.g. see [van Vreeswijk and Sompolinsky, Neural computation 1998]).

Thanks for raising this point. It is both the negative feedback from inhibition and strong coupling strength. To clarify this point, we highlight the feedback inhibition is powerful and strong in the introduction which now reads:

“Preventing run-away excitation and multi-stable attractor dynamics in recurrent networks requires powerful and often finely tuned feedback inhibition resulting in EI balance (Amit and Brunel, 1997; Compte et al., 2000; Litwin-Kumar and Doiron, 2012; Ponce-Alvarez et al., 2013; Mazzucato et al., 2019). However, strong feedback inhibition tends to linearize steady-state activity (Van Vreeswijk and Sompolinsky, 1996; Baker et al., 2020).”

– Above line 101. I do not understand why strong E-E connectivity is consistent only with detJ<0. Strong E-E connectivity typically also requires strong E-I and I-E connectivity [Tsodyks et al., J neuroscience 1997]. The assumption of detJ<0 seems important for the behavior of the characteristic function F(z). How are the results of the paper modified in the case in which det J>0?

Thanks for asking this question. We realized that we were not clear about this point in our introduction. We are, specifically, interested in networks with det*J <* 0 and supralinear activation functions *because* they have the ability to generate run-away dynamics for large enough initial conditions (*Ahmadian et al., 2013*) (and in the absence of STP). This is a key ingredient of NTA and it underlies its threshold dependence and high amplification gain, but it requires an additional “reset” or re-stabilization mechanism which is implemented by STP. Previous work has largely focused on the case of det*J >* 0 because this ensures network stability for *any* input provided that inhibition is fast enough, in particular, when it is infinitely fast (*_I_*∕*_E_* → 0).

To make this point clear we rewrote the corresponding Results section following Equation (3). It now reads as follows:

“We were specifically interested in networks with strong recurrent excitation that can generate positive feedback dynamics in response to external inputs *g_E_*. Therefore, we studied networks with

det(**J**) = −*J_EE_J_II_* + *J_IE_J_EI_ <* 0 *.* (1)

In contrast, networks in which recurrent excitation is met by strong feedback inhibition such that det(**J**) *>* 0 are unable to generate positive feedback dynamics provided that inhibition is fast enough (*Ahmadian et al., 2013*).”

It is worth noting, however, that positive runaway dynamics can also occur in networks with det*J >* 0 if inhibition is not fast enough. This, for instance, is shown in the example below with *E* = 20ms and *_I_* = 60ms, in which additional stimulation can still lead to unstable network dynamics in networks with det(**J**) *>* 0. The conclusions about the stabilization mechanisms, however, still seem to hold as we now show in a new figure supplement with simulations (*Figure 2*—figure supplement *11*). We now clearly explain this point and refer to the new figure supplement in the Discussion section, where we write:

“NTA’s requirement to generate positive feedback dynamics through recurrent excitation, motivated our focus on networks with det(**J**) *<* 0. As demonstrated in previous work (*Ahmadian et al., 2013*), supralinear networks with det(**J**) *>* 0 and instantaneous inhibition (_I_∕*_E_*→ 0) are always stable for any given input, they are thus unable to generate positive feedback dynamics. In addition, networks with det(**J**) *>* 0 can exhibit a range of interesting behaviors, e.g., oscillatory dynamics and persistent activity (*Kraynyukova and Tchumatchenko, 2018*). It is worth noting, however, that for delayed or slow inhibition, stimulation can still lead to unstable network dynamics in networks with det(**J**) *>* 0. Nevertheless, our simulations suggest that our main conclusions about the stabilization mechanisms still hold (*Figure 2*—figure supplement *11*).”

– Line 124-126. I am confused about the conclusion of this section. Results discussed in Pages 5 and 6 are relative to the stability of the network as a function of input strength; why is this relevant for the nonlinear amplification of inputs? Nonlinear amplification of inputs is a general property of the SSN, which emerges also for parameters that lead to stable dynamics [Ahmadian et al., Neural computation 2013]. The authors should also explain why they select parameters that lead to unstable responses for large inputs. On a related note, the fact that the supralinear input-output function leads to activity dependent coupling and, for certain parameters, to unstable dynamics has been shown in [Ahmadian et al., Neural computation 2013]; the authors should acknowledge this fact.

Our apologies, this was not explained clearly in our previous submission. We agree that some form of nonlinear amplification can also occur in SSNs, which are stabilized through inhibition by design, as also shown in (*Echeveste et al., 2020*). However, in the present manuscript we consider networks which can transiently destabilize due to the temporary lack of inhibition, which is essentially the definition of NTA. NTA relies on the runway dynamics for amplification, but renders them transient through the addition of STP. Essentially the same mechanism underlies neuronal action potential generation whereby depolarization of the membrane potential leads to the opening of sodium channels, which leads to further depolarization, which leads to further opening … If it wasn’t for the inactivation of sodium channels and the simultaneous opening of potassium channels which induce the reset, the spike would never “finish”. We fully agree with the reviewer, that supralinear input-output functions lead to activity-dependent coupling and associated input-dependent instabilities was shown in (*Ahmadian et al., 2013*). To clarify this point and to also explain why we chose parameters that lead to unstable responses of large inputs we now clearly state these points in the beginning of our Results section just below Equation (3):

“We were specifically interested in networks with strong recurrent excitation that can generate positive feedback dynamics in response to external inputs . Therefore, we studied networks with

det(**J**) = −*J_EE_J_II_* + *J_IE_J_EI_ <* 0 *.* (2)

In contrast, networks in which recurrent excitation is met by strong feedback inhibition such that det(**J**) *>* 0 are unable to generate positive feedback dynamics provided that inhibition is fast enough (*Ahmadian et al., 2013*).

– Line 188. There are many more papers related to ISN in cortex to be cited; see [Sadeh and Clopath, Nat Rev Neurosci 2021] for a summary.

Thank you for this comment. We have now included additional references in the sentence in line 193-195.

“Recent studies suggest that cortical networks operate as inhibition-stabilized networks (ISNs) (*Sanzeni et al., 2020*; *Sadeh and Clopath, 2021*), in which the excitatory network is unstable in the absence of feedback inhibition (*Tsodyks et al., 1997*; *Ozeki et al., 2009*).”

– Line 273-274. It would be useful to include citations of experiments supporting this claim.

Thank you for this comment. We have now added the corresponding references supporting the claim in line 310-312.

“Neural circuits are capable of generating stereotypical activity patterns in response to partial cues and forming distinct representations in response to different stimuli (*Carrillo-Reid et al., 2016*; *Marshel et al., 2019*; *Bolding et al., 2020*; *Vinje and Gallant, 2000*; *Cayco-Gajic and Silver, 2019*).”

– Line 407-409. What is the relation of this work with network criticality?

That’s an interesting question. There are some commonalities, but also some differences. In network criticality, and specifically in self-organized criticality the network is at a critical point at which a phase transition occurs, which leads the system displays hallmarks of spatial and temporal scale-invariance, e.g., as stimulus responses on a diversity of time-scales and power-law neuronal avalanche distributions. To that end, the network model is typically kept at the edge of chaos and driven by external noise. In contrast, in this article we studied networks which are for most part kept away from any critical point, i.e., from the transition between stable and transiently unstable dynamics (unless when the stimulus is turned on, but rather away). It is, however, conceivable that for suitable input statistics the boundary between the two becomes blurred. To elucidate this point, we added three sentences in the Discussion section which read as follows:

“NTA shares some properties with the notion of network criticality in the brain, like synchronous activation of cell ensembles (*Plenz and Thiagarajan, 2007*) and STP which can tune networks to a critical state (*Levina et al., 2007*). However, in contrast to most models of criticality, in NTA an ensemble briefly transitions to supercritical dynamics in a controlled, stimulus-dependent manner rather than spontaneously. Yet, how the two paradigms are connected at a more fundamental level, is an intriguing question left for future work.”

– The stability of a system with excitatory-inhibitory neurons and STD depends on the eigenvalues of a three dimensional matrix, why does the condition described after Equation (32) do not include the component given by the dynamics of x?

We thank the reviewer for this comment. The relevant equations now include. Specifically, we have redone substantial parts of our analysis in response to Point P 1.5. In particular, we now added a detailed analysis for the 3D system which incorporates the components corresponding to the dynamics of the STP variables. To that end, we added several new method sections in which we performed linear stability analysis on the full 3D systems. The derivations are laid out in the following Methods and Appendix sections:

– Stability conditions in networks with SFA (Methods)

– Stability conditions in networks with E-to-E STD (Methods)

– Stability conditions in networks with E-to-I STF (Appendix 1)

Reviewer #2 (Recommendations for the authors):First a note about the model of Loebel et al.,: note that while single neurons in their models have a ReLU nonlinearity, they receive different external baseline inputs, and therefore the population has an effective input-output nonlinearity that is convex and supralinear as in the current manuscript.Specific/detailed comments:1 – In the abstract, intro and discussion it is claimed that nonnormal connectivity is in clash with symmetric connectivity the E-subnetwork (and thus with the Hebbian learning/doctrine). But non-normal transient amplification happens in E-I networks in which the E-E part of connectivity matrix is indeed symmetric, while the full matrix is necessarily asymmetric and nonnormal (due to Dale's law). So the symmetry of the E-E connectivity (the part that is affected by Hebbian plasticity in most models) is in fact consistent with non-normal TA.

We fully agree with the reviewer that in both cases, NTA and balanced amplification, the connectivity matrices are non-normal and that in our original submission this was not described accurately. To clarify this point, we extensively revised the manuscript in several places and, specifically, in the passages where we refer to balanced amplification (see also our reply to Point P 1.3).

First, we revised the Abstract and removed the sentence claiming a “clash” between nonnormal connectivity and symmetric connectivity in the E-subnetwork:

“To rapidly process information, neural circuits have to amplify specific activity patterns transiently. How the brain performs this nonlinear operation remains elusive. Hebbian assemblies are one possibility whereby strong recurrent excitatory connections boost neuronal activity. However, such Hebbian amplification is often associated with dynamical slowing of network dynamics, non-transient attractor states, and pathological run-away activity. Feedback inhibition can alleviate these effects but typically linearizes responses and reduces amplification gain. Here we study NTA, a plausible alternative mechanism that reconciles strong recurrent excitation with rapid amplification while avoiding the above issues. NTA has two distinct temporal phases. Initially, positive feedback excitation selectively amplifies inputs that exceed a critical threshold. Subsequently, short-term plasticity quenches the run-away dynamics into an inhibition-stabilized network state. By characterizing NTA in supralinear network models, we establish that the resulting onset transients are stimulus selective and well-suited for speedy information processing. Further, we find that excitatory-inhibitory co-tuning widens the parameter regime in which NTA is possible in the absence of persistent activity. In summary, NTA provides a parsimonious explanation for how excitatory-inhibitory co-tuning and short-term plasticity collaborate in recurrent networks to achieve transient amplification.”

In the revised manuscript, we write in the Introduction:

“Preventing run-away excitation and multi-stable attractor dynamics in recurrent networks requires powerful and often finely tuned feedback inhibition resulting in EI balance (Amit and Brunel, 1997; Compte et al., 2000; Litwin-Kumar and Doiron, 2012; Ponce-Alvarez et al., 2013; Mazzucato et al., 2019), which tends to linearize steady-state activity (Van Vreeswijk and Sompolinsky, 1996; Baker et al., 2020). Murphy and Miller (2009) proposed balanced amplification which reconciles transient amplification with strong recurrent excitation by tightly balancing recurrent excitation with strong feedback inhibition (Goldman, 2009; Hennequin et al., 2012, 2014; Bondanelli and Ostojic, 2020; Gillett et al., 2020). Importantly, balanced amplification was formulated for linear network models of excitatory and inhibitory neurons. Due to linearity, it intrinsically lacks the ability to nonlinearly amplify stimuli which limits its capabilities for pattern completion and pattern separation. Further, how balanced amplification relates to nonlinear neuronal activation functions and nonlinear synaptic transmission as, for instance, mediated by STP (Tsodyks and Markram, 1997; Markram et al., 1998; Zucker and Regehr, 2002; Pala and Petersen, 2015), remains elusive. This begs the question of whether there are alternative nonlinear amplification mechanisms and how they relate to existing theories of recurrent neural network processing.”

Finally, in the Discussion where we compare to previous work, we write:

Several theoretical studies approached the problem of transient amplification in recurrent neural network models. Loebel and Tsodyks (2002) have described an NTA-like mechanism as a driver for powerful ensemble synchronization in rate-based networks and in spiking neural network models of auditory cortex (Loebel et al., 2007). Here we generalized this work to both E-to-E STD and E-to-I STF and provide an in-depth characterization of its amplification capabilities, pattern completion properties, and the resulting network states with regard to their inhibition-stabilization properties. Moreover, we showed that SFA cannot provide similar network stabilization and explored how EI co-tuning interacts with NTA. Finally, we contrasted NTA to alternative transient amplification mechanisms. Balanced amplification is a particularly well-studied transient amplification mechanism (Murphy and Miller, 2009; Goldman, 2009; Hennequin et al., 2014; Bondanelli and Ostojic, 2020; Gillett et al., 2020; Christodoulou et al., 2021) that relies on non-normality of the connectivity matrix to selectively and rapidly amplify stimuli. Importantly, balanced amplification occurs in networks in which strong recurrent excitation is appropriately balanced by strong recurrent inhibition. It is capable of generating rich transient activity in linear network models (Hennequin et al., 2014), and selectively amplifies specific activity patterns, but without a specific activation threshold. In addition, in spiking neural networks, strong input can induce synchronous firing at the population level which is subsequently stabilized by strong feedback inhibition without the requirement for STP mechanisms (Stern et al., 2018). These properties contrast with NTA,

which has a nonlinear activation threshold and intrinsically relies on STP to stabilize otherwise unstable run-away dynamics. Due to the switch of the network’s dynamical state, NTA’s amplification can be orders of magnitudes larger than balanced amplification (Figure 2—figure supplement 3). Interestingly, after the transient amplification phase, ensemble dynamics settle in an inhibitory-stabilized state, which renders NTA compatible with previous work on SSNs but in the presence of STP. Finally, although NTA and balanced amplification rely on different amplification mechanisms, they are not mutually exclusive and could, in principle, co-exist in biological networks.”

2 – Similarly in lines 56-65 of Intro (and 438-454 of discussion) it is claimed that balanced amplification (due to feedback inhibition and recurrent excitation) cannot generate strong TA, because "several ensembles need to be chained together into a hidden feedforward structure" which the authors say is "at odds with the often observed symmetric excitatory connectivity". But again symmetric E-E connectivity is consistent with nonnormality of the full (E-I) connectivity matrix, and moreover the chains are hidden between patterns, and not anatomical between neuronal ensembles. Also long hidden chains are not necessary, and when recurrent E and I connections are strong, short chains (e.g. in the SSN model) can also yield strong enough TA to be consistent with those observed in sensory cortex – though probably not as strong as in NTA. This parts should thus be more accurately written and if the claim is maintained a more thorough comparison (e.g. in a supplementary figure) with a recurrent network with neural nonlinearity (e.g. the supralinear powerlaw as in the current model) but without STP (but possibly with different connectivity so that the instability in Figure 1 does not happen) should be provided.

We agree with the reviewer, we now removed the misleading explanation pertaining to the “chains” in the Introduction and rewrote the corresponding section more accurately. The relevant text passages are bold faced in our reply to the previous point (Point P 2.2). Moreover, to characterize the difference in transient amplification between the different mechanisms, we added *Figure 2*—figure supplement *3* which shows the amplification index in linear networks and SNNs in comparison to networks exhibiting NTA. Of course, a direct comparison is difficult because the mechanisms are parameter dependent. In particular, in *Figure 2*—figure supplement *3* we simulated a linear network with the same connection strengths as in the NTA network, while only changing the nonlinearity. Similarly, for the SSN network, we simply removed STP and increased *J_IE_* and *J_II_* by the minimal amount that would ensure stability. Even though, we only show this difference numerically, we believe that the overall results are general and the same qualitative behavior will also arise in other parameter regimes.

3 – Given that property C is a main selling point, they should characterize it better. The only result plot about this is currently Figure 1F, in which activity actually diverges post-stimulus. The nonlinear threshold (for stimulus strength) should be quantified and shown in some plot (e.g. to be added to Figure 2) for the case in which STP re-stabilizes activity. In particular the dependence of NTA (e.g. as measured by the ratio of transient peak to steady-state response) on parameters such as stimulus strength (c.f. Figure 4B of Loebel and Tsodyks 2002) or J_EE_, can be characterised/plotted.

This point is well taken. We performed additional simulations to characterize the nonlinear threshold behavior of NTA and its dependence on input strength*_E_* as well as the recurrent excitatory weight *J_EE_*. These results are now included in the new *Figure 2*—figure supplement *2*. We now refer to these new results in the Results section where we write:

“Crucially, transient amplification in supralinear networks with STP occurs above a critical threshold (*Figure 2*—figure supplement *2*A, B), and requires recurrent excitation *J*_EE_ to be sufficiently strong (*Figure 2*—figure supplement *2*C, D). To quantify the amplification ability of these networks, we calculated the ratio of the evoked peak firing rate to the input strength, henceforth called the “Amplification index”.”

4 – The effect of co-tuning of E-I (feature G) is characterised in abstract and the corresponding sub-section (its title on p. 11 and Figure 3 title) as broadening the parameter regime of NTA. But Figure 3 in fact shows that NTA happens even in the bistable case and with global inhibition. So a more accurate characterization his that E/I co-tuning broadens the parameter regime in which activity is unistable and non-persistent (which is suitable for sensory processing), without adversely affecting NTA. Also given that NTA is weaker -though still strong- with co-tuned inhibition, perhaps the strength of NTA can be quantatively compared between the two cases of co-tuned vs global inhibition.

This is a good point and we thank the reviewer for the comment. To clarify this in the revised manuscript, we updated the relevant sentence in the Abstract, renamed the corresponding section to “Co-tuned inhibition broadens the parameter regime of NTA in the absence of persistent activity,” and performed additional simulations to quantify the difference of amplification ability between networks with global inhibition and networks with co-tuned inhibition. To that end, we computed the amplification index for both architectures while keeping the total connection strength between the two the same. Networks with co-tuned inhibition are constructed such that E-to-I, I-to-E, and I-to-I connection strengths within the ensemble are two times as strong as the ones between ensembles. In contrast to networks with global inhibition, these connection strengths are the same for within the ensemble and across ensembles. This shows, as the reviewer suspected, that NTA is weaker — but still strong — in networks with co-tuned inhibition.

Finally, we now conclude the corresponding section on co-tuning as follows:

“In comparison to the ensemble with global inhibition, the ensemble with co-tuned inhibition exhibits weaker – but still strong – NTA (*Figure 3*—figure supplement *1*). Thus, co-tuned inhibition broadens the parameter regime in which NTA is possible while simultaneously avoiding persistent attractor dynamics.”

5 – In the parts where intuitive explanation of re-stabilization (or lack thereof) due to SFA or STP are given on pp. 7-8, I would first of all provide Equations. 20-26 right there, or at least refer to them. More importantly when referring to F(z) they should mention that they are employing a fast-slow approximation using the separation of timescales between fast neural activity and adaptation variables (a in SFA, x in STD, etc) are therefore fixing the slow variables to their value at the previous/baseline fixed point. At least I believe this is what they are doing and showing in Figure 2D, but again I'm suggesting they should write this more clearly. In particular, it seems to me that while Equation 30 (with x assumed to be fixed at its value at the preceding fixed point) is consistent with the above interpretation (which I believe is the right thing to do), Equation. 28 is not, as it is obtained by setting a equal to its value at its fixed point corresponding to the dynamical value of r_E_, and this is the right thing to do if "a" was a fast variable, not a slow one (as it presumably is). So, I believe, the correct F(z) for the SFA model should be rederived by assuming treating "a" in equation 20 as a constant independent of r_E_ (rather solving the steady state from 21 and plugging it in Equation 20).

Thanks for the comment. We agree with the reviewer on the point of the fast slow approximation. However, while responding to a point raised by Reviewer 1 (Point P 1.5), we realized that while the characteristic function gives good intuitions about why SFA cannot stabilize the system, the particular approximation is not fully justified. To address this point, we therefore analyzed the full 3D system instead and, consequently, the corresponding analysis has changed substantially in the revised manuscript. Specifically, we now provide the asymptotic stability analysis for the three-dimensional dynamical system for the cases of SFA, E-to-E STD, and E-to-I STF in the new method sections with the titles:

– Stability conditions in networks with SFA (Methods)

– Stability conditions in networks with E-to-E STD (Methods)

– Stability conditions in networks with E-to-I STF (*Appendix 1*)

In the case of STP, the reduction to the characteristic function *F*(*z*) is still valid and illuminating, because a stable high-activity fixed point exists. Due to a separation of timescales between the STP variable and the rate dynamics, we can approximate with its fixed point value ≈^∗^ or the corresponding previous fixed value in the case of input changes (*Figure 2*). This allows us to visualize the change of network stability via the characteristic function. This is now stated clearly in a dedicated Methods section with the title “Characteristic function approximation for networks with E-to-E STD.”

6 – Switching to ISN with stimulus from a non-ISN baseline is presumably parameter dependent, and this dependence should be characterized better (at least give examples of networks in which this is not the case, in a supplementary figure, and correspondingly weaken the absolute statement in lines 192-194 so that it says this transition "can" happen, which would avoid implying that this is necessarily the case, assuming it's not).

Thanks for raising this valid point. Indeed, the switch from ISN to non-ISN is parameter dependent and in the revised manuscript we now state it as such and characterize this transition extensively. First, we edited and added the following sentences making the point to the relevant part of the Results section:

“We found that in networks with STP the ISN index can switch sign from negative to positive during external stimulation, indicating that the ensemble can transition from a non-ISN to an ISN (*Figure 2*F). Notably, this behavior is distinct from linear network models in which the network operating regime is independent of the input (Methods). Whether this switch between non-ISN to ISN occurred, however, was parameter dependent and we also found network configurations that were already in the ISN regime at baseline and remained ISNs during stimulation (*Figure 2*—figure supplement *5*). Thus, re-stabilization was largely unaffected by the network state and consistent with experimentally observed ISN states.”

To make the conditions more concise, we added the mathematical definition of ISN index in networks with STP to the new Methods section “Conditions for ISN in networks with E-to-E STD” and the new Appendix section “Conditions for ISN in networks with E-to-I STF” and added *Figure 2*—figure supplement *5* showing an example network which is already in ISN at baseline. Finally, *Figure 2*—figure supplement *7*A illustrates by plotting the ISN index as a function of input*_E_* that the inhibition stabilization property of the network depends on the input strength*_E_*.

Also in the ISN index (Equation 13) shouldn't J_EE_ be multiplied by x (depression variable) at the fixed point?

This is now fixed. We added the Method section “Conditions for ISN in networks with E-to-E STD” and the variable appears in the relevant expression for the ISN index (Equation (56)).

7 – I found the part about pattern separation (p. 14) not clearly written and motivated. In fact I think the term "pattern separation" should be replaced with "stimulus selectivity". Pattern separation is the appropriate term when a networks output patterns are more separable than the patterns at the input-level. However, in the current case in which the external input is given only to E1 (only to one of the two E populations), the input is fully separable, so the network is potentially only capable of reducing separation. At any rate the pattern separation would be meaningful if they had compared separation at input-level (distance from decision boundary in gE1 – gE2 plane) with separation at output-level (distance from decision boundary in rE1 – rE2 plane). I think what is currently shown and charcterized is stimulus selectivity and the result is that transient onset responses are more selective than fixed point responses. So this section (and corresponding parts in intro/discussion) should be rewritten to convey this.

We agree with the reviewer that the setup in *Figure 4* really addresses questions pertaining to pattern selectivity rather than pattern separation, which is rather the topic of *Figure 5*.

To address this point, we changed the figure and text in the section “NTA provides better pattern completion than fixed points while retaining stimulus selectivity” to refer to selectivity rather than pattern separation and we discuss pattern separation in the next section “NTA provides higher amplification and pattern separation in morphing experiments.”

Specifically, we expanded the relevant paragraph at the end of this section. It now reads:

“Further, we examined the impact of competition through lateral inhibition as a function of the E-to-I inter-ensemble strength *J’_IE_* (Methods). As above, we quantified its impact by measuring the representational distance to the decision boundary for the transient onset responses and fixed point activity. We found that regardless of the specific STP mechanism, the distance was larger for the onset responses than for the fixed point activity, consistent with the notion that the onset dynamics separate stimulus identity reliably (*Figure 5*D, E). Since the absolute activity levels between onset and fixed point differed substantially, we further computed the relative pattern Separation index (*r_E_*_2_ − *r_E_*_1_)∕(*r_E_*_1_ + *r_E_*_2_) and found that the onset transient provides better pattern separation ability for ambiguous stimuli with close to 0.5 (*Figure 5*—figure supplement *1*) provided that the E-to-I connection strength across ensembles *J’_IE_*^′^ is strong enough. All the while separability for the onset transient was slightly decreased for distinct inputs with ∈ {0*,*1} in comparison to the fixed point. In contrast, fixed points clearly separated such pure stimuli while providing weaker pattern separation for ambiguous input combinations. Importantly, these findings qualitatively held for networks with NTA mediated by E-to-I STF (*Figure 5*—figure supplement *2*). Thus, NTA provides stronger amplification and pattern separation than fixed point activity in response to ambiguous stimuli.”

7a – On the other hand, this finding (stronger selectivity at onset, vs steady-state or sustained response) is presented as a desired outcome. This may be so, but it seems inconsistent with empirical findings on orientation selectivity in (mouse) visual cortex in which orientation selectivity is much stronger for sustained responses compared to onset transients. This should be mentioned (e.g. in lines 290, 326, and 418-420).

Thanks for raising this interesting point. Unfortunately, we were unable to find a clear line of literature arguing either for or against this statement. On the one hand, we found sources that might suggest an increase of selectivity at steady state (*Ringach et al., 1997*; *Miller et al., 2001*), while other studies (in primates) seem to suggest the opposite (*Celebrini et al., 1993*). Importantly, the comparison is hampered, because in many studies the input stimulus does not seem to be on for long enough to allow for the system to settle in a steady state (*Ringach et al., 1997*). Finally, the fact we did not quantify the width of tuning curves of continuous stimuli in the present study further complicates the direct comparisons between our model and the above studies.

To, nevertheless, address this point, we rephrased our conclusions more carefully and added the following sentence to the Discussion (formerly 418-420):

“Yet, it remains to be seen whether these findings are also coherent with data on the temporal evolution in other sensory systems.”

7b – The distance to decision boundary used to characterize "pattern separation" is not clearly described in lines 280-282. The description in the Methods section was also somewhat ambiguous, but giving a formula there would solve this issue. What I understood is that the measure is basically rE1 – rE2 where rE1 and rE2 are population mean activities of the E neurons in ensembles 1 and 2. If so, why not use the normalized measure (rE1 – rE2)/(rE1 + rE2)?

Thanks for making this point. We used an absolute distance measure to capture that the difference in absolute firing rate, which is what a downstream neuron can directly read out, is much larger during the onset transient. In contrast, the relative distance measure always goes to ±1 if one of the populations has zero activity, which happens in our example at the fixed point at which the overall activity might still be very low and hence hard to read out by downstream brain areas. Thus, pattern selectivity at the fixed point is high, but it is not clear whether it could be read-out if the overall activity is too low. We now add both the absolute distance measure and a relative measure in our analysis of the morphing experiments (see below).

In any case, we now provide the formula for the distance to the decision boundary in the Methods section “Distance to the decision boundary.” Moreover, we added an inset to *Figure 4*A which illustrates this measure.

8 – The property B is depicted in various places (e.g. line 410) by saying NTA requires or relies on symmetric E-E connections. While NTA clearly relies on the destabilizing effect of recurrent EE connections, it is not clear that the symmetry (at least exact symmetry) of these connections is necessary. Indeed their LIF spiking net has random connectivity, so the EE part of connectivity at the level of single neurons in that network is actually not symmetric. These sentences could thus be more accurately written to clarify this point.

We agree, this has now been amended. We extensively revised our Abstract, Introduction and Discussion and now speak of recurrent excitation rather than (exact) symmetry. Please see our response to Points P 1.3 and P 2.2.

9 – In the current study, external stimulus was only given to excitatory neurons, with authors saying the study of feedforward input to inhibition is beyond the scope of this work, yet they also say (lines 429-432) that "we are confident that our main findings remain unaffected in the presence of substantial feedforward inhibition", and cite supplementary figure 1, which however was for the case without STP. I think it would be easy to provide a supplementary figure characterizing NTA in a network with STP in which gI is nonzero for stimulus (e.g. gI = gE).

We fully agree. We added *Figure 2*—figure supplement *12* to address this point. The figure characterizes the amplification in networks with E-to-E STD in the presence of additional feedforward inhibitory inputs. It shows that NTA can still emerge even when stimulus causes a change in feedforward inhibition*_I_* that is by far greater than the change in feedforward excitation_E_.

In the Discussion we now refer to this figure as follows:

“While an in-depth comparison for different origins of inhibition was beyond the scope of the present study, we found that increasing the inputs to the excitatory population and inhibitory population by the same amount can still lead to NTA (*Figure 1*—figure supplement *1*; *Figure 2*—figure supplement *12*; Methods), suggesting that our main findings can remain unaffected in the presence of substantial feedforward inhibition.”

References

Ahmadian Y, Rubin DB, Miller KD. Analysis of the stabilized supralinear network. Neural Computation 2013;25(8):1994–2037. doi: https://doi.org/10.1162/NECO_a_00472.

Amit DJ, Brunel N. Model of global spontaneous activity and local structured activity during delay periods in the cerebral cortex. Cerebral Cortex 1997;7(3):237–252. doi: https://doi.org/10.1093/cercor/7.3.237.

Baker C, Zhu V, Rosenbaum R. Nonlinear stimulus representations in neural circuits with approximate excitatoryinhibitory balance. PLoS Computational Biology 2020;16(9):1–30. doi: https://doi.org/10.1371/journal.pcbi. 1008192.

Bolding KA, Nagappan S, Han BX, Wang F, Franks KM. Recurrent circuitry is required to stabilize piriform cortex odor representations across brain states. *eLife* 2020;9:1–23. doi: https://doi.org/10.7554/*eLife*.53125.

Bondanelli G, Ostojic S. Coding with transient trajectories in recurrent neural networks. PLoS Computational Biology 2020;16(2):1–36. doi: https://doi.org/10.1371/journal.pcbi.1007655.

Carrillo-Reid L, Yang W, Bando Y, Peterka DS, Yuste R. Imprinting and recalling cortical ensembles. Science 2016;353(6300):691–694. doi: https://doi.org/10.1126/science.aaf7560.

Cayco-Gajic NA, Silver RA. Re-evaluating Circuit Mechanisms Underlying Pattern Separation. Neuron 2019;101(4):584–602. doi: https://doi.org/10.1016/j.neuron.2019.01.044.

Celebrini S, Thorpe S, Trotter Y, Imbert M. Dynamics of orientation coding in area VI of the awake primate. Visual Neuroscience 1993;10(5):811–825. doi: https://doi.org/10.1017/S0952523800006052.

Christodoulou G, Vogels TP, Agnes EJ. Regimes and mechanisms of transient amplification in abstract and biological networks. bioRxiv 2021; doi: https://doi.org/10.1101/2021.04.01.437964.

Compte A, Brunel N, Goldman-Rakic PS, Wang XJ. Synaptic mechanisms and network dynamics underlying spatial working memory in a cortical network model. Cerebral Cortex 2000;10(9):910–923. doi: https://doi.org/10.1093/ cercor/10.9.910.

DeWeese MR, Wehr M, Zador AM. Binary spiking in auditory cortex. Journal of Neuroscience 2003;23(21):7940– 7949. doi: https://doi.org/10.1523/JNEUROSCI.23-21-07940.2003.

Echeveste R, Aitchison L, Hennequin G, Lengyel M. Cortical-like dynamics in recurrent circuits optimized for sampling-based probabilistic inference. Nature Neuroscience 2020;23(9):1138–1149. doi: http://dx.doi.org/10.1038/s41593-020-0671-1.

Gillett M, Pereira U, Brunel N. Characteristics of sequential activity in networks with temporally asymmetric Hebbian learning. Proceedings of the National Academy of Sciences of the United States of America 2020;117(47):29948–29958. doi: https://doi.org/10.1073/pnas.1918674117.

Goldman MS. Memory without feedback in a neural network. Neuron 2009;61(4):621–634. doi: https://doi.org/10. 1016/j.neuron.2008.12.012.

Hennequin G, Ahmadian Y, Rubin DB, Lengyel M, Miller KD. The dynamical regime of sensory cortex: Stable dynamics around a single stimulus-tuned attractor account for patterns of noise variability. Neuron 2018;98(4):846– 860.e5. doi: https://doi.org/10.1016/j.neuron.2018.04.017.

Hennequin G, Vogels TP, Gerstner W. Non-normal amplification in random balanced neuronal networks. Physical Review E – Statistical, Nonlinear, and Soft Matter Physics 2012;86(1):1–12. doi: https://doi.org/10.1103/PhysRevE.86.011909.

Hennequin G, Vogels TP, Gerstner W. Optimal control of transient dynamics in balanced networks supports generation of complex movements. Neuron 2014;82(6):1394–1406. doi: https://doi.org/10.1016/j.neuron.2014.04.045.

Kraynyukova N, Tchumatchenko T. Stabilized supralinear network can give rise to bistable, oscillatory, and persistent activity. Proceedings of the National Academy of Sciences of the United States of America 2018;115(13):3464–3469. doi: https://doi.org/10.1073/pnas.1700080115.

Levina A, Herrmann JM, Geisel T. Dynamical synapses causing self-organized criticality in neural networks. Nature Physics 2007;3(12):857–860. doi: https://doi.org/10.1038/nphys758.

Litwin-Kumar A, Doiron B. Slow dynamics and high variability in balanced cortical networks with clustered connections. Nature Neuroscience 2012;15(11):1498–1505. doi: https://doi.org/10.1038/nn.3220.

Loebel A, Nelken I, Tsodyks M. Processing of sounds by population spikes in a model of primary auditory cortex. Frontiers in Neuroscience 2007;1:15. doi: https://doi.org/10.3389/neuro.01.1.1.015.2007.

Loebel A, Tsodyks M. Computation by Ensemble Synchronization in Recurrent Networks with Synaptic Depression. J Comput Neurosci 2002 Sep;13(2):111–124. doi: https://doi.org/10.1023/A:1020110223441.

Markram H, Wang Y, Tsodyks M. Differential signaling via the same axon of neocortical pyramidal neurons. Proceedings of the National Academy of Sciences of the United States of America 1998;95(9):5323–5328. doi: https://doi.org/10.1073/pnas.95.9.5323.

Marshel JH, Kim YS, Machado TA, Quirin S, Benson B, Kadmon J, et al. Cortical layer-specific critical dynamics triggering perception. Science 2019;365(6453). doi: https://doi.org/10.1126/science.aaw5202.

Mazzucato L, La Camera G, Fontanini A. Expectation-induced modulation of metastable activity underlies faster coding of sensory stimuli. Nature Neuroscience 2019;22(5):787–796. doi: https://doi.org/10.1038/ s41593-019-0364-9.

Miller KD, Pinto DJ, Simons DJ. Processing in layer 4 of the neocortical circuit: New insights from visual and somatosensory cortex. Current Opinion in Neurobiology 2001;11(4):488–497. doi: https://doi.org/10.1016/ S0959-4388(00)00239-7.

Murphy BK, Miller KD. Balanced amplification: A new mechanism of selective amplification of neural activity patterns. Neuron 2009;61(4):635–648. doi: https://doi.org/10.1016/j.neuron.2009.02.005.

Ozeki H, Finn IM, Schaffer ES, Miller KD, Ferster D. Inhibitory Stabilization of the Cortical Network Underlies Visual Surround Suppression. Neuron 2009;62(4):578–592. doi: http://doi.org/10.1016/j.neuron.2009.03.028.

Pala A, Petersen CCH. in vivo measurement of cell-type-specific synaptic connectivity and synaptic transmission in layer 2/3 mouse barrel cortex. Neuron 2015;85(1):68–75. doi: https://doi.org/10.1016/j.neuron.2014.11.025.

Plenz D, Thiagarajan TC. The organizing principles of neuronal avalanches: cell assemblies in the cortex? Trends in Neurosciences 2007 Mar;30(3):101–110. doi: https://doi.org/10.1016/j.tins.2007.01.005.

Ponce-Alvarez A, Thiele A, Albright TD, Stoner GR, Deco G. Stimulus-dependent variability and noise correlations in cortical MT neurons. Proceedings of the National Academy of Sciences of the United States of America 2013;110(32):13162–13167. doi: https://doi.org/10.1073/pnas.1300098110.

Ringach DL, Hawken MJ, Shapley R. The dynamics of orientation tuning in macaque primary visual cortex. Investigative Ophthalmology and Visual Science 1997;38(4):281–284. doi: https://doi.org/10.1038/387281a0.

Rubin DB, VanHooser SD, Miller KD. The stabilized supralinear network: A unifying circuit motif underlying multiinput integration in sensory cortex. Neuron 2015;85(2):402–417. doi: https://doi.org/10.1016/j.neuron.2014.12. 026.

Rupprecht P, Friedrich RW. Precise synaptic balance in the zebrafish homolog of olfactory cortex. Neuron 2018;100(3):669–683.e5. doi: https://doi.org/10.1016/j.neuron.2018.09.013.

Sadeh S, Clopath C. Inhibitory stabilization and cortical computation. Nature Reviews Neuroscience 2021;22(1):21–37. doi: http://doi.org/10.1038/s41583-020-00390-z.

Sanzeni A, Akitake B, Goldbach HC, Leedy CE, Brunel N, Histed MH. Inhibition stabilization is a widespread property of cortical networks. *eLife* 2020;9:1–39. doi: https://doi.org/10.7554/*eLife*.54875.

Stern M, Bolding KA, Abbott LF, Franks KM. A transformation from temporal to ensemble coding in a model of piriform cortex. *eLife* 2018;7:1–26. doi: https://doi.org/10.7554/*eLife*.34831.

Tsodyks MV, Markram H. The neural code between neocortical pyramidal neurons depends on neurotransmitter release probability. Proceedings of the National Academy of Sciences of the United States of America 1997;94(2):719–723. doi: https://doi.org/10.1073/pnas.94.2.719.

Tsodyks MV, Skaggs WE, Sejnowski TJ, McNaughton BL. Paradoxical effects of external modulation of inhibitory interneurons. Journal of Neuroscience 1997;17(11):4382–4388. doi: https://doi.org/10.1523/jneurosci.17-11-04382.1997.

Van Vreeswijk C, Hansel D. Patterns of synchrony in neural networks with spike adaptation. Neural Computation 2001;13(5):959–992. doi: https://doi.org/10.1162/08997660151134280.

Van Vreeswijk C, Sompolinsky H. Chaos in neuronal networks with balanced excitatory and inhibitory activity. Science 1996;274(5293):1724–1726. doi: https://doi.org/10.1126/science.274.5293.1724.

Vinje WE, Gallant JL. Sparse coding and decorrelation in primary visual cortex during natural vision. Science 2000;287(5456):1273–1276. doi: http://doi.org/10.1126/science.287.5456.1273.

Zucker RS, Regehr WG. Short-term synaptic plasticity. Annual Review of Physiology 2002;64:355–405. doi: https://doi.org/10.1146/annurev.physiol.64.092501.114547.